# MOGANET: MULTI-ORDER GATED AGGREGATION NETWORK

**Siyuan Li**[1,2][*] **Zedong Wang**[1][*] **Zicheng Liu**[1,2] **Cheng Tan**[1,2] **Haitao Lin**[1,2] **Di Wu**[1,2]
**Zhiyuan Chen**[1,2] **Jiangbin Zheng**[1,2] **Stan Z. Li**[1][†]
[1]AI Lab, Research Center for Industries of the Future, Westlake University, Hangzhou, China
[2]Zhejiang University, College of Computer Science and Technology, Hangzhou, China

## ABSTRACT

By contextualizing the kernel as global as possible, Modern ConvNets have shown great potential in computer vision tasks. However, recent progress on *multi-order game-theoretic interaction* within deep neural networks (DNNs) reveals the representation bottleneck of modern ConvNets, where the expressive interactions have not been effectively encoded with the increased kernel size. To tackle this challenge, we propose a new family of modern ConvNets, dubbed MogaNet, for discriminative visual representation learning in pure ConvNet-based models with favorable complexity-performance trade-offs. MogaNet encapsulates conceptually simple yet effective convolutions and gated aggregation into a compact module, where discriminative features are efficiently gathered and contextualized adaptively. MogaNet exhibits great scalability, impressive efficiency of parameters, and competitive performance compared to state-of-the-art ViTs and ConvNets on ImageNet and various downstream vision benchmarks, including COCO object detection, ADE20K semantic segmentation, 2D&3D human pose estimation, and video prediction. Notably, MogaNet hits 80.0% and 87.8% accuracy with 5.2M and 181M parameters on ImageNet-1K, outperforming ParC-Net and ConvNeXt-L, while saving 59% FLOPs and 17M parameters, respectively. The source code is available at https://github.com/Westlake-AI/MogaNet.

## 1 INTRODUCTION

By relaxing local inductive bias, Vision Transformers (ViTs) (Dosovitskiy et al., 2021; Liu et al., 2021) have rapidly challenged the long dominance of Convolutional Neural Networks (ConvNets) (Ren et al., 2015; He et al., 2016; Kirillov et al., 2019) for visual recognition. It is commonly conjectured that such superiority of ViT stems from its self-attention operation (Bahdanau et al., 2015; Vaswani et al., 2017), which facilitates the global-range feature interaction. From a practical standpoint, however, the quadratic complexity within self-attention prohibitively restricts its computational efficiency (Wang et al., 2021a; Hua et al., 2022) and applications to high-resolution fine-grained scenarios (Zhu et al., 2021; Jiang et al., 2021a; Liu et al., 2022a). Additionally, the dearth of local bias induces the detriment of neighborhood correlations (Pinto et al., 2022).

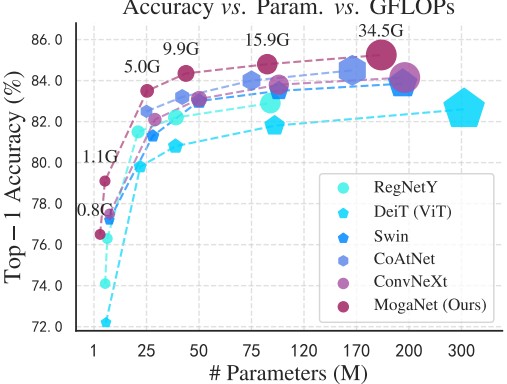

Figure 1: **Performance on ImageNet-1K validation set at $224^2$ resolutions.** MogaNet outperforms Transformers (DeiT(Touvron et al., 2021a) and Swin (Liu et al., 2021)), ConvNets (RegNetY (Radosavovic et al., 2020) and ConvNeXt (Liu et al., 2022b)), and hybrid models (CoAtNet (Dai et al., 2021)) across all scales.

To resolve this problem, endeavors have been made by reintroducing locality priors (Wu et al., 2021a; Dai et al., 2021; Han et al., 2021a; Li et al., 2022a; Chen et al., 2022) and pyramid-like hier-

---

[*]First two authors contribute equally. [†]Corrsponding author (stan.zq.li@westlake.edu.cn).

archical layouts (Liu et al., 2021; Fan et al., 2021; Wang et al., 2021b) to ViTs, albeit at the expense of model generalizability and expressivity. Meanwhile, further explorations toward ViTs (Tolstikhin et al., 2021; Raghu et al., 2021; Yu et al., 2022) have triggered the resurgence of modern ConvNets (Liu et al., 2022b; Ding et al., 2022b). With advanced training setup and ViT-style framework design, ConvNets can readily deliver competitive performance *w.r.t.* well-tuned ViTs across a wide range of vision benchmarks (Wightman et al., 2021; Pinto et al., 2022). Essentially, most of the modern ConvNets aim to perform feature extraction in a *local-global blended fashion* by contextualizing the convolutional kernel or the perception module as *global* as possible.

Despite their superior performance, recent progress on *multi-order game-theoretic interaction* within DNNs (Ancona et al., 2019b; Zhang et al., 2020; Cheng et al., 2021) unravels that the representation capacity of modern ConvNets has not been exploited well. Holistically, low-order interactions tend to model relatively simple and common local visual concepts, which are of poor expressivity and are incapable of capturing high-level semantic patterns. In comparison, the high-order ones represent the complex concepts of absolute global scope yet are vulnerable to attacks and with poor generalizability. Deng et al. (2022) first shows that modern networks are implicitly prone to encoding extremely low- or high-order interactions rather than the empirically proved more discriminative middle ones. Attempts have been made to tackle this issue from the perspective of loss function (Deng et al., 2022) and modeling contextual relations (Wu et al., 2022a; Li et al., 2023a). This unveils the serious challenge but also the great potential for modern ConvNet architecture design.

To this end, we present a new ConvNet architecture named **M**ulti-**o**rder **g**ated **a**ggregation Network (MogaNet) to achieve *adaptive* context extraction and further pursue more discriminative and efficient visual representation learning first under the guidance of interaction within modern ConvNets. In MogaNet, we encapsulate both locality perception and gated context aggregation into a compact spatial aggregation block, where features encoded by the inherent overlooked interactions are forced to congregated and contextualized efficiently in parallel. From the channel perspective, as existing methods are prone to huge channel-wise information redundancy (Raghu et al., 2021; Hua et al., 2022), we design a conceptually simple yet effective channel aggregation block to adaptively force the network to encode expressive interactions that would have originally been ignored. Intuitively, it performs channel-wise reallocation to the input, which outperforms prevalent counterparts (*e.g.*, SE (Hu et al., 2018), RepMLP (Ding et al., 2022a)) with more favorable computational overhead.

Extensive experiments demonstrate the consistent efficiency of model parameters and competitive performance of MogaNet at different model scales on various vision tasks, including image classification, object detection, semantic segmentation, instance segmentation, pose estimation, *etc.* As shown in Fig. 1, MogaNet achieves 83.4% and 87.8% top-1 accuracy with 25M and 181M parameters, which exhibits favorable computational overhead compared with existing lightweight models. MogaNet-T attains 80.0% accuracy on ImageNet-1K, outperforming the state-of-the-art ParC-Net-S (Zhang et al., 2022b) by 1.0% with 2.04G lower FLOPs. MogaNet also shows great performance gain on various downstream tasks, *e.g.,* surpassing Swin-L (Liu et al., 2021) by 2.3% $AP^b$ on COCO detection with fewer parameters and computational budget. It is surprising that the parameter efficiency of MogaNet exceeds our expectations. This is probably owing to the network encodes more discriminative middle-order interactions, which maximizes the usage of model parameters.

## 2 RELATED WORK

### 2.1 VISION TRANSFORMERS

Since the success of Transformer (Vaswani et al., 2017) in natural language processing (Devlin et al., 2018), ViT has been proposed (Dosovitskiy et al., 2021) and attained impressive results on ImageNet (Deng et al., 2009). Yet, compared to ConvNets, ViTs are over-parameterized and rely on large-scale pre-training (Bao et al., 2022; He et al., 2022; Li et al., 2023a). Targeting this problem, one branch of researchers presents lightweight ViTs (Xiao et al., 2021; Mehta & Rastegari, 2022; Li et al., 2022c; Chen et al., 2023) with efficient attentions (Wang et al., 2021a). Meanwhile, the incorporation of self-attention and convolution as a hybrid backbone has been studied (Guo et al., 2022; Wu et al., 2021a; Dai et al., 2021; d'Ascoli et al., 2021; Li et al., 2022a; Pan et al., 2022b; Si et al., 2022) for imparting locality priors to ViTs. By introducing local inductive bias (Zhu et al., 2021; Chen et al., 2021; Jiang et al., 2021a; Arnab et al., 2021), advanced training strategies (Touvron et al., 2021a; Yuan et al., 2021a; Touvron et al., 2022) or extra knowledge (Jiang et al., 2021b;

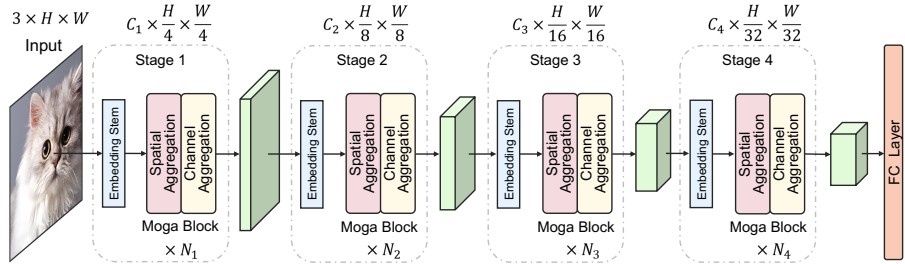

Figure 2: **MogaNet architecture with four stages.** Similar to (Liu et al., 2021; 2022b), MogaNet uses hierarchical architecture of 4 stages. Each stage $i$ consists of an embedding stem and $N_i$ Moga Blocks, which contain spatial aggregation blocks and channel aggregation blocks.

Lin et al., 2022; Wu et al., 2022c), ViTs can achieve superior performance and have been extended to various vision areas. MetaFormer (Yu et al., 2022) considerably influenced the roadmap of deep architecture design, where all ViTs (Trockman & Kolter, 2022; Wang et al., 2022a) can be classified by the token-mixing strategy, such as relative position encoding (Wu et al., 2021b), local window shifting (Liu et al., 2021) and MLP layer (Tolstikhin et al., 2021), *etc.*

## 2.2 POST-ViT MODERN CONVNETS

Taking the merits of ViT-style framework design (Yu et al., 2022), modern ConvNets (Liu et al., 2022b; 2023; Rao et al., 2022; Yang et al., 2022) show superior performance with large kernel depth-wise convolutions (Han et al., 2021b) for global perception (view Appendix E for detail backgrounds). It primarily comprises three components: (i) embedding stem, (ii) spatial mixing block, and (iii) channel mixing block. Embedding stem downsamples the input to reduce redundancies and computational overload. We assume the input feature $X$ is in the shape $\mathbb{R}^{C \times H \times W}$, we have:

$$Z = \text{Stem}(X), \tag{1}$$

where $Z$ is downsampled features, *e.g.,*. Then, the feature flows to a stack of residual blocks. In each stage, the network modules can be decoupled into two separate functional components, $\text{SMixer}(\cdot)$ and $\text{CMixer}(\cdot)$ for spatial-wise and channel-wise information propagation,

$$Y = X + \text{SMixer}\big(\text{Norm}(X)\big), \tag{2}$$
$$Z = Y + \text{CMixer}\big(\text{Norm}(Y)\big), \tag{3}$$

where $\text{Norm}(\cdot)$ denotes a normalization layer, *e.g.,* BatchNorm (Ioffe & Szegedy, 2015a) (BN). $\text{SMixer}(\cdot)$ can be various spatial operations (*e.g.,* self-attention, convolution), while $\text{CMixer}(\cdot)$ is usually achieved by channel MLP with inverted bottleneck (Sandler et al., 2018) and expand ratio $r$. Notably, we abstract *context aggregation* in modern ConvNets as a series of operations that can *adaptively* aggregate contextual information while suppressing trivial redundancies in spatial mixing block $\text{SMixer}(\cdot)$ between two embedded features:

$$O = \mathcal{S}\big(\mathcal{F}_\phi(X), \mathcal{G}_\psi(X)\big), \tag{4}$$

where $\mathcal{F}_\phi(\cdot)$ and $\mathcal{G}_\psi(\cdot)$ are the aggregation and context branches with parameters $\phi$ and $\psi$. Context aggregation models the importance of each position on $X$ by the aggregation branch $\mathcal{F}_\phi(X)$ and reweights the embedded feature from the context branch $\mathcal{G}_\psi(X)$ by operation $\mathcal{S}(\cdot, \cdot)$.

# 3 MULTI-ORDER GAME-THEORETIC INTERACTION FOR DEEP ARCHITECTURE DESIGN

**Representation Bottleneck of DNNs**   Recent studies toward the generalizability (Geirhos et al., 2019; Ancona et al., 2019a; Tuli et al., 2021; Geirhos et al., 2021) and robustness (Naseer et al., 2021; Zhou et al., 2022; Park & Kim, 2022) of DNNs delivers a new perspective to improve deep architectures. Apart from them, the investigation of multi-order game-theoretic interaction unveils the representation bottleneck of DNNs. Methodologically, multi-order interactions between two input variables represent marginal contribution brought by collaborations among these two and other involved contextual variables, where the order indicates the number of contextual variables within the collaboration. Formally, it can be explained by $m$-th order game-theoretic interaction $I^{(m)}(i,j)$ and $m$-order interaction strength $J^{(m)}$, as defined in (Zhang et al., 2020; Deng et al., 2022). Considering the image with $n$ patches in total, $I^{(m)}(i,j)$ measures the average interaction complexity between the patch pair $i,j$ over all contexts consisting of $m$ patches, where $0 \le m \le n-2$ and the

order $m$ reflects the scale of the context involved in the game-theoretic interactions between pixels $i$ and $j$. Normalized by the average of interaction strength, the relative interaction strength $J^{(m)}$ with $m \in (0,1)$ measures the complexity of interactions encoded in DNNs. Notably, low-order interactions tend to encode **common** or **widely-shared local texture**, and the high-order ones are inclined to forcibly memorize the pattern of **rare outliers** (Deng et al., 2022; Cheng et al., 2021). As shown in Fig. 3, existing DNNs are implicitly prone to excessively low- or high-order interactions while suppressing the most expressive and versatile middle-order ones (Deng et al., 2022; Cheng et al., 2021). Refer to Appendix B.1 for definitions and more details.

**Multi-order Interaction for Architecture Design.**
Existing deep architecture design is usually derived from intuitive insights, lacking hierarchical theoretic guidance. Multi-order interaction can serve as a reference that fits well with the already gained insights on computer vision and further guides the ongoing quest. For instance, the extremely high-order interactions encoded in ViTs (*e.g.*, DeiT in Fig. 3) may stem from its adaptive global-range self-attention mechanism. Its superior robustness can be attributed to its excessive low-order interactions, representing common and widely shared local patterns. However, the absence of locality priors still leaves ViTs lacking middle-order interactions, which cannot be replaced by the low-order ones. As for modern ConvNets (*e.g.*, SLaK in Fig. 3), despite the $51 \times 51$ kernel size, it still fails to encode enough expressive interactions (view more results in Appendix B.1). Likewise, we argue that such a dilemma may be attributed to the inappropriate com-

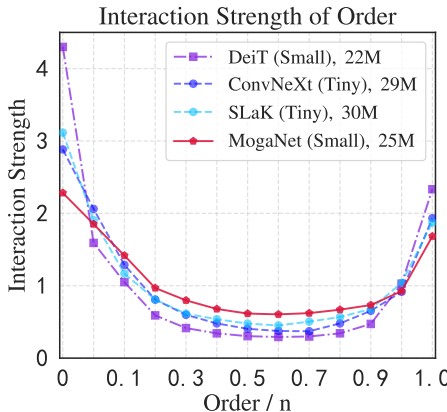

Figure 3: **Distributions of the interaction strength** $J^{(m)}$ for Transformers and ConvNets on ImageNet-1K with $224^2$ resolutions and $n = 14 \times 14$.

position of convolutional locality priors and global context injections (Treisman & Gelade, 1980; Tuli et al., 2021; Li et al., 2023a). A naive combination of self-attention or convolutions can be intrinsically prone to the strong bias of global shape (Geirhos et al., 2021; Ding et al., 2022b) or local texture (Hermann et al., 2020), infusing extreme-order interaction preference to models. In MogaNet, we aim to provide an architecture that can *adaptively force the network to encode expressive interactions that would have otherwise been ignored inherently*.

# 4 METHODOLOGY

## 4.1 OVERVIEW OF MOGANET

Built upon modern ConvNets, we design a four-stage MogaNet architecture as illustrated in Fig. 2. For stage $i$, the input image or feature is first fed into an embedding stem to regulate the resolutions and embed into $C_i$ dimensions. Assuming the input image in $H \times W$ resolutions, features of the four stages are in $\frac{H}{4} \times \frac{W}{4}$, $\frac{H}{8} \times \frac{W}{8}$, $\frac{H}{16} \times \frac{W}{16}$, and $\frac{H}{32} \times \frac{W}{32}$ resolutions respectively. Then, the embedded

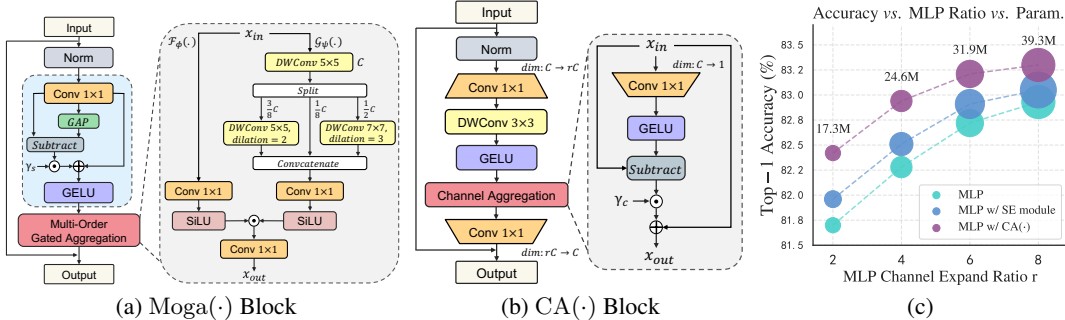

(a) Moga($\cdot$) Block        (b) CA($\cdot$) Block        (c)

Figure 4: **(a) Structure of spatial aggregation block** Moga($\cdot$)**. (b) Structure of channel aggregation block. (c) Analysis of channel MLP and the channel aggregation module.** Based on MogaNet-S, performances and model sizes of the raw channel MLP, MLP with SE block, and the channel aggregation is compared with the MLP ratio of $\{2, 4, 6, 8\}$ on ImageNet-1K.

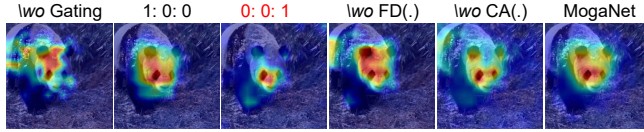

| \wo Gating | 1: 0: 0 | 0: 0: 1 | \wo FD(.) | \wo CA(.) | MogaNet |

**Figure 5: Grad-CAM visualization of ablations.** 1: 0: 0 and 0: 0: 1 denote only using $C_l$ or $C_h$ for Multi-order DWConv Layers in SA block. The models encoded extremely low- ($C_l$) or high- ($C_h$) order interactions are sensitive to similar regional textures (1: 0: 0) or excessive discriminative parts (0: 0: 1), not localizing precise semantic parts. Gating effectively eliminates the disturbing contextual noise (\wo Gating).

| Modules | | Top-1 Acc (%) | Params. (M) | FLOPs (G) |
|---|---|---|---|---|
| Baseline | | 76.6 | 4.75 | 1.01 |
| SMixer | +Gating branch | 77.3 | 5.09 | 1.07 |
| | +DW$_{7\times7}$ | 77.5 | 5.14 | 1.09 |
| | +Multi-order DW($\cdot$) | 78.0 | 5.17 | 1.10 |
| | +FD($\cdot$) | 78.3 | 5.18 | 1.10 |
| CMixer | +SE module | 78.6 | 5.29 | 1.14 |
| | +CA($\cdot$) | **79.0** | 5.20 | 1.10 |

Table 1: **Ablation of designed modules on ImageNet-1K.** The baseline uses the non-linear projection and DW$_{5\times5}$ as SMixer($\cdot$) and the vanilla MLP as CMixer($\cdot$).

feature flows into $N_i$ Moga Blocks, consisting of spatial and channel aggregation blocks (in Sec. 4.2 and 4.3), for further context aggregation. After the final output, GAP and a linear layer are added for classification tasks. As for dense prediction tasks (He et al., 2017; Xiao et al., 2018b), the output from four stages can be used through neck modules (Lin et al., 2017a; Kirillov et al., 2019).

### 4.2 Multi-order Spatial Gated Aggregation

As discussed in Sec. 3, DNNs with the incompatible composition of locality perception and context aggregation can be implicitly prone to extreme-order game-theoretic interaction strengths while suppressing the more robust and expressive middle-order ones (Li et al., 2022a; Pinto et al., 2022; Deng et al., 2022). As shown in Fig. 5, the primary obstacle pertains to how to **force** the network to encode the originally ignored expressive interactions and informative features. We first suppose that the essential *adaptive* nature of attention in ViTs has not been well leveraged and grafted into ConvNets. Thus, we propose spatial aggregation (SA) block as an instantiation of SMixer($\cdot$) to learn representations of multi-order interactions in a unified design, as shown in Fig. 4a, consisting of two cascaded components. We instantiate Eq. (2) as:

$$Z = X + \text{Moga}\Big(\text{FD}\big(\text{Norm}(X)\big)\Big), \tag{5}$$

where FD($\cdot$) indicates a feature decomposition module (FD) and Moga($\cdot$) denotes a multi-order gated aggregation module comprising the gating $\mathcal{F}_\phi(\cdot)$ and context branch $\mathcal{G}_\psi(\cdot)$.

**Context Extraction.** As a pure ConvNet structure, we extract multi-order features with both *static* and *adaptive* locality perceptions. There are two complementary counterparts, fine-grained local texture (low-order) and complex global shape (middle-order), which are instantiated by $\text{Conv}_{1\times1}(\cdot)$ and GAP($\cdot$) respectively. To **force** the network against its implicitly inclined interaction strengths, we design FD($\cdot$) to adaptively exclude the trivial (overlooked) interactions, defined as:

$$Y = \text{Conv}_{1\times1}(X), \tag{6}$$

$$Z = \text{GELU}\Big(Y + \gamma_s \odot \big(Y - \text{GAP}(Y)\big)\Big), \tag{7}$$

where $\gamma_s \in \mathbb{R}^{C\times1}$ denotes a scaling factor initialized as zeros. By re-weighting the complementary interaction component $Y - \text{GAP}(Y)$, FD($\cdot$) also increases spatial feature diversities (Park & Kim, 2022; Wang et al., 2022b). Then, we ensemble depth-wise convolutions (DWConv) to encode multi-order features in the context branch of Moga($\cdot$). Unlike previous works that simply combine DWConv with self-attentions to model local and global interactions (Zhang et al., 2022b; Pan et al., 2022a; Si et al., 2022; Rao et al., 2022) , we employ three different DWConv layers with dilation ratios $d \in \{1, 2, 3\}$ in parallel to capture low, middle, and high-order interactions: given the input feature $X \in \mathbb{R}^{C\times HW}$, $\text{DW}_{5\times5,d=1}$ is first applied for low-order features; then, the output is factorized into $X_l \in \mathbb{R}^{C_l\times HW}$, $X_m \in \mathbb{R}^{C_m\times HW}$, and $X_h \in \mathbb{R}^{C_h\times HW}$ along the channel dimension, where $C_l + C_m + C_h = C$; afterward, $X_m$ and $X_h$ are assigned to $\text{DW}_{5\times5,d=2}$ and $\text{DW}_{7\times7,d=3}$, respectively, while $X_l$ serves as identical mapping; finally, the output of $X_l$, $X_m$, and $X_h$ are concatenated to form multi-order contexts, $Y_C = \text{Concat}(Y_{l,1:C_l}, Y_m, Y_h)$. Notice that the proposed FD($\cdot$) and multi-order DWConv layers only require a little extra computational overhead and parameters in comparison to $\text{DW}_{7\times7}$ used in ConvNeXt (Liu et al., 2022b), *e.g.,* +multi-order and +FD($\cdot$) increase 0.04M parameters and 0.01G FLOPS over $\text{DW}_{7\times7}$ as shown in Table 1.

**Gated Aggregation.** To *adaptively* aggregate the extracted feature from the context branch, we employ SiLU (Elfwing et al., 2018) activation in the gating branch, *i.e.,* $x \cdot \text{Sigmoid}(x)$, which has been well-acknowledged as an advanced version of Sigmoid activation. As illustrated in Appendix C.1, we empirically show that SiLU in MogaNet exhibits both the gating effects as Sigmoid and the stable training property. Taking the output from FD($\cdot$) as the input, we instantiate Eq. (4):

$$Z = \underbrace{\text{SiLU}\big(\text{Conv}_{1\times1}(X)\big)}_{\mathcal{F}_\phi} \odot \underbrace{\text{SiLU}\big(\text{Conv}_{1\times1}(Y_C)\big)}_{\mathcal{G}_\psi}, \tag{8}$$

With the proposed SA blocks, MogaNet captures more middle-order interactions, as validated in Fig. 3. The SA block produces discriminative multi-order representations with similar parameters and FLOPs as $\text{DW}_{7\times7}$ in ConvNeXt, which is well beyond the reach of existing methods without the cost-consuming self-attentions.

### 4.3 MULTI-ORDER CHANNEL REALLOCATION

Prevalent architectures, as illustrated in Sec. 2, perform channel-mixing $\text{CMixer}(\cdot)$ mainly by two linear projections, *e.g.,* 2-layer channel-wise MLP (Dosovitskiy et al., 2021; Liu et al., 2021; Tolstikhin et al., 2021) with a expand ratio $r$ or the MLP with a $3 \times 3$ DWConv in between (Wang et al., 2022c; Pan et al., 2022b;a). Due to the information redundancy cross channels (Woo et al., 2018; Cao et al., 2019; Tan & Le, 2019; Wang et al., 2020), vanilla MLP requires a number of parameters ($r$ default to 4 or 8) to achieve expected performance, showing low computational efficiency as plotted in Fig. 4c. To address this issue, most current methods directly insert a channel enhancement module, *e.g.,* SE module (Hu et al., 2018), into MLP. Unlike these designs requiring additional MLP bottleneck, motivated by $\text{FD}(\cdot)$, we introduce a lightweight channel aggregation module $\text{CA}(\cdot)$ to adaptive reallocate channel-wise features in

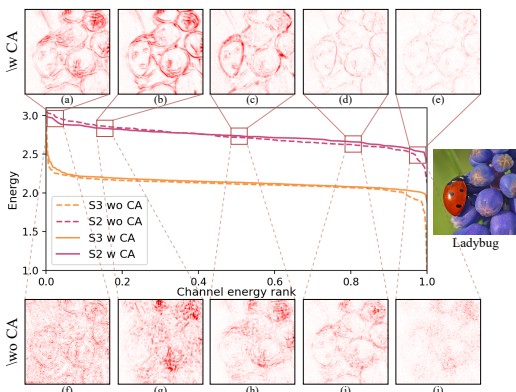

Figure 6: **Channel energy ranks and channel saliency maps (CSM)** (Kong et al., 2022) with or without our CA block based on MogaNet-S. The energy reflects the importance of the channel, while the highlighted regions of CSMs are the activated spatial features of each channel.

high-dimensional hidden spaces and further extend it to a channel aggregation (CA) block. As shown in Fig. 4b, we rewrite Eq. (3) for our CA block as:

$$Y = \text{GELU}\Big(\text{DW}_{3\times3}\big(\text{Conv}_{1\times1}(\text{Norm}(X))\big)\Big),$$
$$Z = \text{Conv}_{1\times1}\big(\text{CA}(Y)\big) + X. \tag{9}$$

Concretely, $\text{CA}(\cdot)$ is implemented by a channel-reducing projection $W_r : \mathbb{R}^{C \times HW} \to \mathbb{R}^{1 \times HW}$ and GELU to gather and reallocate channel-wise information:

$$\text{CA}(X) = X + \gamma_c \odot \big(X - \text{GELU}(XW_r)\big), \tag{10}$$

where $\gamma_c$ is the channel-wise scaling factor initialized as zeros. It reallocates the channel-wise feature with the complementary interactions $(X - \text{GELU}(XW_r))$. As shown in Fig. 7, $\text{CA}(\cdot)$ enhances originally overlooked game-theoretic interactions. Fig. 4c and Fig. 6 verify the effectiveness of $\text{CA}(\cdot)$ compared with vanilla MLP and MLP with SE module in channel-wise effiency and representation ability. Despite some improvements to the baseline, the MLP $w/$ SE module still requires large MLP ratios (*e.g., $r = 6$*) to achieve expected performance while bringing extra parameters and overhead. Yet, our $\text{CA}(\cdot)$ with $r = 4$ brings 0.6% gain over the baseline at a small extra cost (0.04M extra parameters & 0.01G FLOPs) while achieving the same performance as the baseline with $r = 8$.

### 4.4 IMPLEMENTATION DETAILS

Following the network design style of ConvNets (Liu et al., 2022b), we scale up MogaNet for six model sizes (X-Tiny, Tiny, Small, Base, Large, and X-Large) via stacking the different number of spatial and channel aggregation blocks at each stage, which has similar numbers of parameters as RegNet (Radosavovic et al., 2020) variants. Network configurations and hyper-parameters are detailed in Table A1. FLOPs and throughputs are analyzed in Appendix C.3. We set the channels of the multi-order DWConv layers to $C_l : C_m : C_h$ = 1:3:4 (see Appendix C.2). Similar to (Touvron et al., 2021c; Li et al., 2022a;c), the first embedding stem in MogaNet is designed as two stacked $3\times3$ convolution layers with the stride of 2 while adopting the single-layer version for embedding stems in other three stages. We select GELU (Hendrycks & Gimpel, 2016) as the common activation function and only use SiLU in the Moga module as Eq. (8).

## 5 EXPERIMENTS

To impartially evaluate and compare MogaNet with the leading network architectures, we conduct extensive experiments across various popular vision tasks, including image classification, object detection, instance and semantic segmentation, 2D and 3D pose estimation, and video prediction. The experiments are implemented with PyTorch and run on NVIDIA A100 GPUs.

### 5.1 IMAGENET CLASSIFICATION

**Settings.** For classification experiments on ImageNet (Deng et al., 2009), we train our MogaNet following the standard procedure (Touvron et al., 2021a; Liu et al., 2021) on ImageNet-1K (IN-1K) for a fair comparison, training 300 epochs with AdamW (Loshchilov & Hutter, 2019) optimizer, a basic learning rate of $1 \times 10^{-3}$, and a cosine scheduler (Loshchilov & Hutter, 2016). To explore the large model capacities, we pre-trained MogaNet-XL on ImageNet-21K (IN-21K) for 90 epochs and then fine-tuned 30 epochs on IN-1K following (Liu et al., 2022b). Appendix A.2 and D.1 provide implementation details and more results. We compare three classical architectures: **Pure ConvNets** (C), **Transformers** (T), and **Hybrid model** (H) with both self-attention and convolution operations.

**Results.** With regard to the lightweight models, Table 2 shows that MogaNet-XT/T significantly outperforms existing lightweight architectures with a more efficient usage of parameters and FLOPs. MogaNet-T achieves 79.0% top-1 accuracy, which improves models with $\sim$5M parameters by at least 1.1 at $224^2$ resolutions. Using $256^2$ resolutions, MogaNet-T outperforms the current SOTA ParC-Net-S by 1.0 while achieving 80.0% top-1 accuracy with the refined settings. Even with only 3M parameters, MogaNet-XT still surpasses models with around 4M parameters, *e.g.,* +4.6 over T2T-ViT-7. Particularly, MogaNet-T$^\S$ achieves 80.0% top-1 accuracy using $256^2$ resolutions and the refined training settings (detailed in Appendix C.5). As for scaling up models in Table 3, MogaNet shows superior or comparable performances to SOTA architectures with similar parameters and computational costs. For example, MogaNet-S achieves 83.4% top-1 accuracy, outperforming Swin-T and ConvNeXt-T with a clear margin of 2.1 and 1.2. MogaNet-B/L also improves recently proposed ConvNets with fewer parameters, *e.g.,* +0.3/0.4 and +0.5/0.7 points over HorNet-S/B and SLaK-S/B. When pre-trained on IN-21K, MogaNet-XL is boosted to 87.8% top-1 accuracy with 181M parameters, saving 169M compared to ConvNeXt-XL. Noticeably, MogaNet-XL can achieve 85.1% at $224^2$ resolutions without pre-training and improves ConvNeXt-L by 0.8, indicating MogaNets are easier to converge than existing models (also verified in Appendix D.1).

### 5.2 DENSE PREDICTION TASKS

**Object detection and segmentation on COCO.** We evaluate MogaNet for object detection and instance segmentation tasks on COCO (Lin et al., 2014) with RetinaNet (Lin et al., 2017b), Mask-RCNN (He et al., 2017), and Cascade Mask R-CNN (Cai & Vasconcelos, 2019) as detectors. Following the training and evaluation settings in (Liu et al., 2021; 2022b), we fine-tune the models by the AdamW optimizer for $1\times$ and $3\times$ training schedule on COCO *train2017* and evaluate on COCO *val2017*, implemented on MMDetection (Chen et al., 2019) codebase. The box mAP (AP$^b$) and mask mAP (AP$^m$) are adopted as metrics. Refer Appendix A.3 and D.2 for detailed settings and full results. Table 4 shows that detectors with MogaNet variants significantly outperform previous backbones. It is worth noticing that Mask R-CNN with MogaNet-T achieves 42.6 AP$^b$, outperforming Swin-T by 0.4 with 48% and 27% fewer parameters and FLOPs. Using advanced training setting and IN-21K pre-trained weights, Cascade Mask R-CNN with MogaNet-XL achieves 56.2 AP$^b$, +1.4 and +2.3 over ConvNeXt-L and RepLKNet-31L.

**Semantic segmentation on ADE20K.** We also evaluate MogaNet for semantic segmentation tasks on ADE20K (Zhou et al., 2018) with Semantic FPN (Kirillov et al., 2019) and UperNet (Xiao et al., 2018b) following (Liu et al., 2021; Yu et al., 2022), implemented on MMSegmentation (Contributors, 2020b) codebase. The performance is measured by single-scale mIoU. Initialized by IN-1K or IN-21K pre-trained weights, Semantic FPN and UperNet are fine-tuned for 80K and 160K iterations by the AdamW optimizer. See Appendix A.4 and D.3 for detailed settings and full results. In Table 5, Semantic FPN with MogaNet-S consistently outperforms Swin-T and Uniformer-S by 6.2 and 1.1 points; UperNet with MogaNet-S/B/L improves ConvNeXt-T/S/B by 2.5/1.4/1.8 points. Using higher resolutions and IN-21K pre-training, MogaNet-XL achieves 54.0 SS mIoU, surpassing ConvNeXt-L and RepLKNet-31L by 0.3 and 1.6.

| Architecture | Date | Type | Image Size | Param. (M) | FLOPs (G) | Top-1 Acc (%) |
|---|---|---|---|---|---|---|
| ResNet-18 | CVPR'2016 | C | $224^2$ | 11.7 | 1.80 | 71.5 |
| ShuffleNetV2 2× | ECCV'2018 | C | $224^2$ | 5.5 | 0.60 | 75.4 |
| EfficientNet-B0 | ICML'2019 | C | $224^2$ | 5.3 | 0.39 | 77.1 |
| RegNetY-800MF | CVPR'2020 | C | $224^2$ | 6.3 | 0.80 | 76.3 |
| DeiT-T$^†$ | ICML'2021 | T | $224^2$ | 5.7 | 1.08 | 74.1 |
| PVT-T | ICCV'2021 | T | $224^2$ | 13.2 | 1.60 | 75.1 |
| T2T-ViT-7 | ICCV'2021 | T | $224^2$ | 4.3 | 1.20 | 71.7 |
| ViT-C | NIPS'2021 | T | $224^2$ | 4.6 | 1.10 | 75.3 |
| SReT-T$_{Distill}$ | ECCV'2022 | T | $224^2$ | 4.8 | 1.10 | 77.6 |
| PiT-Ti | ICCV'2021 | H | $224^2$ | 4.9 | 0.70 | 74.6 |
| LeViT-S | ICCV'2021 | H | $224^2$ | 7.8 | 0.31 | 76.6 |
| CoaT-Lite-T | ICCV'2021 | H | $224^2$ | 5.7 | 1.60 | 77.5 |
| Swin-1G | ICCV'2021 | H | $224^2$ | 7.3 | 1.00 | 77.3 |
| MobileViT-S | ICLR'2022 | H | $256^2$ | 5.6 | 4.02 | 78.4 |
| MobileFormer-294M | CVPR'2022 | H | $224^2$ | 11.4 | 0.59 | 77.9 |
| ConvNext-XT | CVPR'2022 | C | $224^2$ | 7.4 | 0.60 | 77.5 |
| VAN-B0 | CVMJ'2023 | C | $224^2$ | 4.1 | 0.88 | 75.4 |
| ParC-Net-S | ECCV'2022 | C | $256^2$ | 5.0 | 3.48 | 78.6 |
| **MogaNet-XT** | Ours | C | $256^2$ | 3.0 | 1.04 | 77.2 |
| **MogaNet-T** | Ours | C | $224^2$ | 5.2 | 1.10 | 79.0 |
| **MogaNet-T$^§$** | Ours | C | $256^2$ | 5.2 | 1.44 | **80.0** |

Table 2: **IN-1K classification** with lightweight models. § denotes the refined training scheme.

| Architecture | Date | Type | Image Size | Param. (M) | FLOPs (G) | Top-1 Acc (%) |
|---|---|---|---|---|---|---|
| Deit-S | ICML'2021 | T | $224^2$ | 22 | 4.6 | 79.8 |
| Swin-T | ICCV'2021 | T | $224^2$ | 28 | 4.5 | 81.3 |
| CSWin-T | CVPR'2022 | T | $224^2$ | 23 | 4.3 | 82.8 |
| LITV2-S | NIPS'2022 | T | $224^2$ | 28 | 3.7 | 82.0 |
| CoaT-S | ICCV'2021 | H | $224^2$ | 22 | 12.6 | 82.1 |
| CoAtNet-0 | NIPS'2021 | H | $224^2$ | 25 | 4.2 | 82.7 |
| UniFormer-S | ICLR'2022 | H | $224^2$ | 22 | 3.6 | 82.9 |
| RegNetY-4GF$^†$ | CVPR'2020 | C | $224^2$ | 21 | 4.0 | 81.5 |
| ConvNeXt-T | CVPR'2022 | C | $224^2$ | 29 | 4.5 | 82.1 |
| SLaK-T | ICLR'2023 | C | $224^2$ | 30 | 5.0 | 82.5 |
| HorNet-T$_{7×7}$ | NIPS'2022 | C | $224^2$ | 22 | 4.0 | 82.8 |
| **MogaNet-S** | Ours | C | $224^2$ | 25 | 5.0 | **83.4** |
| Swin-S | ICCV'2021 | T | $224^2$ | 50 | 8.7 | 83.0 |
| Focal-S | NIPS'2021 | T | $224^2$ | 51 | 9.1 | 83.6 |
| CSWin-S | CVPR'2022 | T | $224^2$ | 35 | 6.9 | 83.6 |
| LITV2-M | NIPS'2022 | T | $224^2$ | 49 | 7.5 | 83.3 |
| CoaT-M | ICCV'2021 | H | $224^2$ | 45 | 9.8 | 83.6 |
| CoAtNet-1 | NIPS'2021 | H | $224^2$ | 42 | 8.4 | 83.3 |
| UniFormer-B | ICLR'2022 | H | $224^2$ | 50 | 8.3 | 83.9 |
| FAN-B-Hybrid | ICML'2022 | H | $224^2$ | 50 | 11.3 | 83.9 |
| EfficientNet-B6 | ICML'2019 | C | $528^2$ | 43 | 19.0 | 84.0 |
| RegNetY-8GF$^†$ | CVPR'2020 | C | $224^2$ | 39 | 8.1 | 82.2 |
| ConvNeXt-S | CVPR'2022 | C | $224^2$ | 50 | 8.7 | 83.1 |
| FocalNet-S (LRF) | NIPS'2022 | C | $224^2$ | 50 | 8.7 | 83.5 |
| HorNet-S$_{7×7}$ | NIPS'2022 | C | $224^2$ | 50 | 8.8 | 84.0 |
| SLaK-S | ICLR'2023 | C | $224^2$ | 55 | 9.8 | 83.8 |
| **MogaNet-B** | Ours | C | $224^2$ | 44 | 9.9 | **84.3** |
| DeiT-B | ICML'2021 | T | $224^2$ | 86 | 17.5 | 81.8 |
| Swin-B | ICCV'2021 | T | $224^2$ | 89 | 15.4 | 83.5 |
| Focal-B | NIPS'2021 | T | $224^2$ | 90 | 16.4 | 84.0 |
| CSWin-B | CVPR'2022 | T | $224^2$ | 78 | 15.0 | 84.2 |
| DeiT III-B | ECCV'2022 | T | $224^2$ | 87 | 18.0 | 83.8 |
| BoTNet-T7 | CVPR'2021 | H | $256^2$ | 79 | 19.3 | 84.2 |
| CoAtNet-2 | NIPS'2021 | H | $224^2$ | 75 | 15.7 | 84.1 |
| FAN-B-Hybrid | ICML'2022 | H | $224^2$ | 77 | 16.9 | 84.3 |
| RegNetY-16GF | CVPR'2020 | C | $224^2$ | 84 | 16.0 | 82.9 |
| ConvNeXt-B | CVPR'2022 | C | $224^2$ | 89 | 15.4 | 83.8 |
| RepLKNet-31B | CVPR'2022 | C | $224^2$ | 79 | 15.3 | 83.5 |
| FocalNet-B (LRF) | NIPS'2022 | C | $224^2$ | 89 | 15.4 | 83.9 |
| HorNet-B$_{7×7}$ | NIPS'2022 | C | $224^2$ | 87 | 15.6 | 84.3 |
| SLaK-B | ICLR'2023 | C | $224^2$ | 95 | 17.1 | 84.0 |
| **MogaNet-L** | Ours | C | $224^2$ | 83 | 15.9 | **84.7** |
| Swin-L$^‡$ | ICCV'2021 | T | $384^2$ | 197 | 104 | 87.3 |
| DeiT III-L$^‡$ | ECCV'2022 | T | $384^2$ | 304 | 191 | 87.7 |
| CoAtNet-3$^‡$ | NIPS'2021 | H | $384^2$ | 168 | 107 | 87.6 |
| RepLKNet-31L$^‡$ | CVPR'2022 | C | $384^2$ | 172 | 96 | 86.6 |
| ConvNeXt-L | CVPR'2022 | C | $224^2$ | 198 | 34.4 | 84.3 |
| ConvNeXt-L$^‡$ | CVPR'2022 | C | $384^2$ | 198 | 101 | 87.5 |
| ConvNeXt-XL$^‡$ | CVPR'2022 | C | $384^2$ | 350 | 179 | 87.8 |
| HorNet-L$^‡$ | NIPS'2022 | C | $384^2$ | 202 | 102 | 87.7 |
| **MogaNet-XL** | Ours | C | $224^2$ | 181 | 34.5 | 85.1 |
| **MogaNet-XL$^‡$** | Ours | C | $384^2$ | 181 | 102 | **87.8** |

Table 3: **IN-1K classification** performance with scaling-up models. ‡ denotes the model is pre-trained on IN-21K and fine-tuned on IN-1K.

| Architecture | Data | Method | Param. (M) | FLOPs (G) | AP$^b$ (%) | AP$^m$ (%) |
|---|---|---|---|---|---|---|
| ResNet-101 | CVPR'2016 | RetinaNet | 57 | 315 | 38.5 | - |
| PVT-S | ICCV'2021 | RetinaNet | 34 | 226 | 40.4 | - |
| CMT-S | CVPR'2022 | RetinaNet | 45 | 231 | 44.3 | - |
| **MogaNet-S** | Ours | RetinaNet | 35 | 253 | **45.8** | - |
| RegNet-1.6G | CVPR'2020 | Mask R-CNN | 29 | 204 | 38.9 | 35.7 |
| PVT-T | ICCV'2021 | Mask R-CNN | 33 | 208 | 36.7 | 35.1 |
| **MogaNet-T** | Ours | Mask R-CNN | 25 | 192 | **42.6** | **39.1** |
| Swin-T | ICCV'2021 | Mask R-CNN | 48 | 264 | 42.2 | 39.1 |
| Uniformer-S | ICLR'2022 | Mask R-CNN | 41 | 269 | 45.6 | 41.6 |
| ConvNeXt-T | CVPR'2022 | Mask R-CNN | 48 | 262 | 44.2 | 40.1 |
| PVTV2-B2 | CVMJ'2022 | Mask R-CNN | 45 | 309 | 45.3 | 41.2 |
| LITV2-S | NIPS'2022 | Mask R-CNN | 47 | 261 | 44.9 | 40.8 |
| FocalNet-T | NIPS'2022 | Mask R-CNN | 49 | 267 | 45.9 | 41.3 |
| **MogaNet-S** | Ours | Mask R-CNN | 45 | 272 | **46.7** | **42.2** |
| Swin-S | ICCV'2021 | Mask R-CNN | 69 | 354 | 44.8 | 40.9 |
| Focal-S | NIPS'2021 | Mask R-CNN | 71 | 401 | 47.4 | 42.8 |
| ConvNeXt-S | CVPR'2022 | Mask R-CNN | 70 | 348 | 45.4 | 41.8 |
| HorNet-B$_{7×7}$ | NIPS'2022 | Mask R-CNN | 68 | 322 | 47.4 | 42.3 |
| **MogaNet-B** | Ours | Mask R-CNN | 63 | 373 | **47.9** | **43.2** |
| Swin-L$^‡$ | ICCV'2021 | Cascade Mask | 253 | 1382 | 53.9 | 46.7 |
| ConvNeXt-L$^‡$ | CVPR'2022 | Cascade Mask | 255 | 1354 | 54.8 | 47.6 |
| RepLKNet-31L$^‡$ | CVPR'2022 | Cascade Mask | 229 | 1321 | 53.9 | 46.5 |
| HorNet-L$^‡$ | NIPS'2022 | Cascade Mask | 259 | 1399 | 56.0 | 48.6 |
| **MogaNet-XL$^‡$** | Ours | Cascade Mask | 238 | 1355 | **56.2** | **48.8** |

Table 4: **COCO object detection and instance segmentation** with RetinaNet (1×), Mask R-CNN (1×), and Cascade Mask R-CNN (multi-scale 3×). ‡ indicates IN-21K pre-trained models. The FLOPs are measured at $800 × 1280$.

| Method | Architecture | Date | Crop size | Param. (M) | FLOPs (G) | mIoU$^{ss}$ (%) |
|---|---|---|---|---|---|---|
| Semantic FPN (80K) | PVT-S | ICCV'2021 | $512^2$ | 28 | 161 | 39.8 |
| | Twins-S | NIPS'2021 | $512^2$ | 28 | 162 | 44.3 |
| | Swin-T | ICCV'2021 | $512^2$ | 32 | 182 | 41.5 |
| | Uniformer-S | ICLR'2022 | $512^2$ | 25 | 247 | 46.6 |
| | LITV2-S | NIPS'2022 | $512^2$ | 31 | 179 | 44.3 |
| | VAN-B2 | CVMJ'2023 | $512^2$ | 30 | 164 | 46.7 |
| | **MogaNet-S** | Ours | $512^2$ | 29 | 189 | **47.7** |
| UperNet (160K) | DeiT-S | ICML'2021 | $512^2$ | 52 | 1099 | 44.0 |
| | Swin-T | ICCV'2021 | $512^2$ | 60 | 945 | 46.1 |
| | ConvNeXt-T | CVPR'2022 | $512^2$ | 60 | 939 | 46.7 |
| | UniFormer-S | ICLR'2022 | $512^2$ | 52 | 1008 | 47.6 |
| | HorNet-T$_{7×7}$ | NIPS'2022 | $512^2$ | 52 | 926 | 48.1 |
| | **MogaNet-S** | Ours | $512^2$ | 55 | 946 | **49.2** |
| | Swin-S | ICCV'2021 | $512^2$ | 81 | 1038 | 48.1 |
| | ConvNeXt-S | CVPR'2022 | $512^2$ | 82 | 1027 | 48.7 |
| | SLaK-S | ICLR'2023 | $512^2$ | 91 | 1028 | 49.4 |
| | **MogaNet-B** | Ours | $512^2$ | 74 | 1050 | **50.1** |
| | Swin-B | ICCV'2021 | $512^2$ | 121 | 1188 | 49.7 |
| | ConvNeXt-B | CVPR'2022 | $512^2$ | 122 | 1170 | 49.1 |
| | RepLKNet-31B | CVPR'2022 | $512^2$ | 112 | 1170 | 49.9 |
| | SLaK-B | ICLR'2023 | $512^2$ | 135 | 1185 | 50.2 |
| | **MogaNet-L** | Ours | $512^2$ | 113 | 1176 | **50.9** |
| | Swin-L$^‡$ | ICCV'2021 | $640^2$ | 234 | 2468 | 52.1 |
| | ConvNeXt-L$^‡$ | CVPR'2022 | $640^2$ | 245 | 2458 | 53.7 |
| | RepLKNet-31L$^‡$ | CVPR'2022 | $640^2$ | 207 | 2404 | 52.4 |
| | **MogaNet-XL$^‡$** | Ours | $640^2$ | 214 | 2451 | **54.0** |

Table 5: **ADE20K semantic segmentation** with semantic FPN (80K) and UperNet (160K). ‡ indicates using IN-21K pre-trained models. The FLOPs are measured at $512×2048$ or $640×2560$.

| Architecture | Date | Crop size | Param. (M) | FLOPs (G) | AP (%) | AP$^{50}$ (%) | AP$^{75}$ (%) | AR (%) |
|---|---|---|---|---|---|---|---|---|
| RSN-18 | ECCV'2020 | 256 × 192 | 9.1 | 2.3 | 70.4 | 88.7 | 77.9 | 77.1 |
| **MogaNet-T** | Ours | 256 × 192 | 8.1 | 2.2 | **73.2** | **90.1** | **81.0** | **78.8** |
| HRNet-W32 | CVPR'2019 | 256 × 192 | 28.5 | 7.1 | 74.4 | 90.5 | 81.9 | 78.9 |
| Swin-T | ICCV'2021 | 256 × 192 | 32.8 | 6.1 | 72.4 | 90.1 | 80.6 | 78.2 |
| PVTV2-B2 | CVML'2022 | 256 × 192 | 29.1 | 4.3 | 73.7 | 90.5 | 81.2 | 79.1 |
| Uniformer-S | ICLR'2022 | 256 × 192 | 25.2 | 4.7 | 74.0 | 90.3 | 82.2 | 79.5 |
| ConvNeXt-T | CVPR'2022 | 256 × 192 | 33.1 | 5.5 | 73.2 | 90.0 | 80.9 | 78.8 |
| **MogaNet-S** | Ours | 256 × 192 | 29.0 | 6.0 | **74.9** | **90.7** | **82.8** | **80.1** |
| Uniformer-S | ICLR'2022 | 384 × 288 | 25.2 | 11.1 | 75.9 | 90.6 | 83.4 | 81.4 |
| ConvNeXt-T | CVPR'2022 | 384 × 288 | 33.1 | 33.1 | 75.3 | 90.4 | 82.1 | 80.5 |
| **MogaNet-S** | Ours | 384 × 288 | 29.0 | 13.5 | **76.4** | **91.0** | **83.3** | **81.4** |
| HRNet-W48 | CVPR'2019 | 384 × 288 | 63.6 | 32.9 | 76.3 | 90.8 | 82.0 | 81.2 |
| Swin-L | ICCV'2021 | 384 × 288 | 203.4 | 86.9 | 76.3 | 91.2 | 83.0 | 814 |
| Uniformer-B | ICLR'2022 | 384 × 288 | 53.5 | 14.8 | 76.7 | 90.8 | 84.0 | 81.4 |
| **MogaNet-B** | Ours | 384 × 288 | 47.4 | 24.4 | **77.3** | **91.4** | **84.0** | **82.2** |

Table 6: **COCO 2D human pose estimation** with Top-Down SimpleBaseline. The FLOPs are measured at $256 × 192$ or $384 × 288$.

| Modules | Top-1 Acc (%) |
|---|---|
| ConvNeXt-T | 82.1 |
| Baseline | 82.2 |
| Moga Block | **83.4** |
| $-$FD$(\cdot)$ | 83.2 |
| $-$Multi-DW$(\cdot)$ | 83.1 |
| $-$Moga$(\cdot)$ | 82.7 |
| $-$CA$(\cdot)$ | 82.9 |

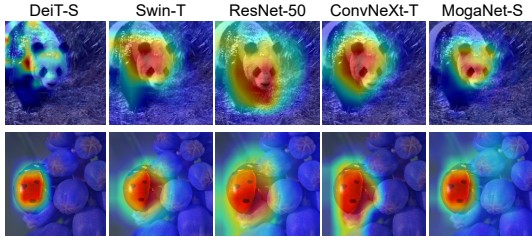

Figure 7: **Ablation of proposed modules** on IN-1K **Left**: the table ablates MogaNet modules by removing each of them based on the baseline of MogaNet-S. **Right**: the figure plots distributions of interaction strength $J^{(m)}$, which verifies that Moga$(\cdot)$ and CA$(\cdot)$ both contributes to learning multi-order interactions and better performance.

Figure 8: **Grad-CAM activation maps** on IN-1K. MogaNet exhibits similar activation maps as attention architectures (Swin), which are located on the semantic targets. Unlike previous ConvNets that might activate some irrelevant regions, activation maps of MogaNet are more semantically gathered. See more results in Appendix B.2.

**2D and 3D Human Pose Estimation.** We evaluate MogaNet on 2D and 3D human pose estimation tasks. As for 2D key points estimation on COCO, we conduct evaluations with SimpleBaseline (Xiao et al., 2018a) following (Wang et al., 2021b; Li et al., 2022a), which fine-tunes the model for 210 epoch by Adam optimizer (Kingma & Ba, 2014). Table 6 shows that MogaNet variants yield at least 0.9 AP improvements for $256 \times 192$ input, *e.g.,* +2.5 and +1.2 over Swin-T and PVTV2-B2 by MogaNet-S. Using $384 \times 288$ input, MogaNet-B outperforms Swin-L and Uniformer-B by 1.0 and 0.6 AP with fewer parameters. As for 3D face/hand surface reconstruction tasks on Stirling/ESRC 3D (Feng et al., 2018) and FreiHAND (Zimmermann et al., 2019) datasets, we benchmark backbones with ExPose (Choutas et al., 2020), which fine-tunes the model for 100 epoch by Adam optimizer. 3DRMSE and Mean Per-Joint Position Error (PA-MPJPE) are the metrics. In Table 7, MogaNet-S shows the lowest errors compared to Transformers and ConvNets. We provide detailed implementations and results for 2D and 3D pose estimation tasks in Appendix D.4 and D.5.

**Video Prediction.** We further objectively evaluate MogaNet for unsupervised video prediction tasks with SimVP (Gao et al., 2022) on MMNIST (Srivastava et al., 2015), where the model predicts the successive 10 frames with the given 10 frames as the input. We train the model for 200 epochs from scratch by the Adam optimizer and are evaluated by MSE and Structural Similarity Index (SSIM). Table 7 shows that SimVP with MogaNet blocks improves the baseline by 6.58 MSE and outperforms ConvNeXt and HorNet by 1.37 and 4.07 MSE. Appendix A.7 and D.6 show more experiment settings and results.

| Architecture | 3D Face | | | 3D Hand | | | Video Prediction | | | |
|---|---|---|---|---|---|---|---|---|---|---|
| | #P. (M) | FLOPs (G) | 3DRMSE $\downarrow$ | #P. (M) | FLOPs (G) | PA-MPJPE (mm)$\downarrow$ | #P. (M) | FLOPs (G) | MSE $\downarrow$ | SSIM (%)$\uparrow$ |
| DeiT-S | 25 | 6.6 | 2.52 | 25 | 4.8 | 7.86 | 46 | 16.9 | 35.2 | 91.4 |
| Swin-T | 30 | 6.1 | 2.45 | 30 | 4.6 | 6.97 | 46 | 16.4 | 29.7 | 93.3 |
| ConvNeXt-T | 30 | 5.8 | 2.34 | 30 | 4.5 | 6.46 | 37 | 14.1 | 26.9 | 94.0 |
| HorNet-T | 25 | 5.6 | 2.39 | 25 | 4.3 | 6.23 | 46 | 16.3 | 29.6 | 93.3 |
| MogaNet-S | 27 | 6.5 | **2.24** | 27 | 5.0 | **6.08** | 47 | 16.5 | **25.6** | **94.3** |

Table 7: **3D human pose estimation** and **video prediction** with ExPose and SimVP on Stirling/ESRC 3D, FreiHAND, and MMNIST datasets. FLOPs of the face and hand tasks are measured at $3 \times 256^2$ and $3 \times 224^2$ while using 10 frames at $1 \times 64^2$ resolutions for video prediction.

### 5.3 ABLATION AND ANALYSIS

We first ablate the spatial aggregation module and the channel aggregation module **CA**$(\cdot)$ in Table 1 and Fig. 7 (left). Spatial modules include **FD**$(\cdot)$ and **Moga**$(\cdot)$, containing the **gating branch** and the context branch with multi-order DWConv layers **Multi-DW**$(\cdot)$. We found that all proposed modules yield improvements with favorable costs. Appendix C provides more ablation studies. Furthermore, Fig. 7 (right) empirically shows design modules can learn more middle-order interactions, and Fig. 8 visualizes class activation maps by Grad-CAM (Selvaraju et al., 2017) compared to existing models.

## 6 CONCLUSION

This paper introduces a new modern ConvNet architecture, named MogaNet, through the lens of multi-order game-theoretic interaction. Built upon the modern ConvNet framework, we present a compact Moga Block and channel aggregation module to force the network to emphasize the expressive but inherently overlooked interactions across spatial and channel perspectives. Extensive experiments verify the consistent superiority of MogaNet in terms of both performance and efficiency compared to popular ConvNets, ViTs, and hybrid architectures on various vision benchmarks.

ACKNOWLEDGEMENT

This work was supported by the National Key R&D Program of China (No. 2022ZD0115100), the National Natural Science Foundation of China Project (No. U21A20427), and Project (No. WU2022A009) from the Center of Synthetic Biology and Integrated Bioengineering of Westlake University. This work was done when Zedong Wang and Zhiyuan Chen interned at Westlake University. We thank the AI Station of Westlake University for the support of GPUs. We also thank Mengzhao Chen, Zhangyang Gao, Jianzhu Guo, Fang Wu, and all anonymous reviewers for polishing the writing of the manuscript.

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

# A  IMPLEMENTATION DETAILS

## A.1  ARCHITECTURE DETAILS

The detailed architecture specifications of Mo-gaNet are shown in Table A1 and Fig. 2, where an input image of $224^2$ resolutions is assumed for all architectures. We rescale the groups of embedding dimensions the number of Moga Blocks for each stage corresponding to different models of vary-ing magnitudes: i) MogaNet-X-Tiny and MogaNet-Tiny with embedding dimensions of $\{32, 64, 96, 192\}$ and $\{32, 64, 128, 256\}$ ex-hibit competitive parameter numbers and com-putational overload as recently proposed light-weight architectures (Mehta & Rastegari, 2022; Chen et al., 2022; Zhang et al., 2022b); ii) MogaNet-Small adopts embedding dimensions

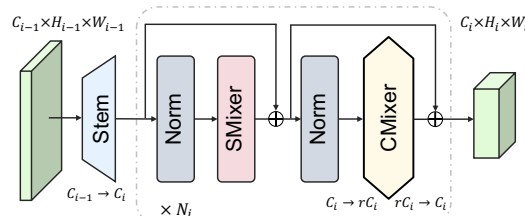

Figure A1: **Modern ConvNet architecture.** It has 4 stages in hierarchical, and $i$-th stage contains an embedding stem and $N_i$ blocks of SMixer$(\cdot)$ and CMixer$(\cdot)$ with PreNorm (Wang et al., 2019) and identical connection (He et al., 2016). The features within the $i$-th stage are in the same shape, except that CMixer$(\cdot)$ will increase the di-mension to $rC_i$ with an expand ratio $r$ as an in-verted bottleneck (Sandler et al., 2018).

of $\{64, 128, 320, 512\}$ in comparison to other prevailing small-scale architectures (Liu et al., 2021; 2022b); iii) MogaNet-Base with embedding dimensions of $\{64, 160, 320, 512\}$ in comparison to medium size architectures; iv) MogaNet-Large with embedding dimensions of $\{64, 160, 320, 640\}$ is designed for large-scale computer vision tasks. v) MogaNet-X-Large with embedding dimensions of $\{96, 192, 480, 960\}$ is a scaling-up version (around 200M parameters) for large-scale tasks. The FLOPs are measured for image classification on ImageNet (Deng et al., 2009) at resolution $224^2$, where a global average pooling (GAP) layer is applied to the output feature map of the last stage, followed by a linear classifier.

| Stage | Output Size | Layer Settings | MogaNet | | | | | |
|---|---|---|---|---|---|---|---|---|
| | | | XTiny | Tiny | Small | Base | Large | XLarge |
| S1 | $\frac{H \times W}{4 \times 4}$ | Stem | Conv$_{3\times3}$, stride 2, C/2 | | | | | |
| | | | Conv$_{3\times3}$, stride 2, C | | | | | |
| | | Embed. Dim. | 32 | 32 | 64 | 64 | 64 | 96 |
| | | # Moga Block | 3 | 3 | 2 | 4 | 4 | 6 |
| | | MLP Ratio | 8 | | | | | |
| S2 | $\frac{H \times W}{8 \times 8}$ | Stem | Conv$_{3\times3}$, stride 2 | | | | | |
| | | Embed. Dim. | 64 | 64 | 128 | 160 | 160 | 192 |
| | | # Moga Block | 3 | 3 | 3 | 6 | 6 | 6 |
| | | MLP Ratio | 8 | | | | | |
| S3 | $\frac{H \times W}{16 \times 16}$ | Stem | Conv$_{3\times3}$, stride 2 | | | | | |
| | | Embed. Dim. | 96 | 128 | 320 | 320 | 320 | 480 |
| | | # Moga Block | 10 | 12 | 12 | 22 | 44 | 44 |
| | | MLP Ratio | 4 | | | | | |
| S4 | $\frac{H \times W}{32 \times 32}$ | Stem | Conv$_{3\times3}$, stride 2 | | | | | |
| | | Embed. Dim. | 192 | 256 | 512 | 512 | 640 | 960 |
| | | # Moga Block | 2 | 2 | 2 | 3 | 4 | 4 |
| | | MLP Ratio | 4 | | | | | |
| Classifier | | | Global Average Pooling, Linear | | | | | |
| Parameters (M) | | | 2.97 | 5.20 | 25.3 | 43.8 | 82.5 | 180.8 |
| FLOPs (G) | | | 0.80 | 1.10 | 4.97 | 9.93 | 15.9 | 34.5 |

Table A1: Architecture configurations of MogaNet variants.

## A.2  EXPERIMENTAL SETTINGS FOR IMAGENET

We conduct image classification experiments on ImageNet (Deng et al., 2009) datasets. All ex-periments are implemented on OpenMixup (Li et al., 2022b) and timm (Wightman et al., 2021) codebases running on 8 NVIDIA A100 GPUs. View more results in Appendix D.1.

**ImageNet-1K.** We perform regular ImageNet-1K training mostly following the training settings of DeiT (Touvron et al., 2021a) and RSB A2 (Wightman et al., 2021) in Table A2, which are widely

| Configuration | DeiT | RSB A2 | MogaNet | | | | | |
|---|---|---|---|---|---|---|---|---|
| | | | XT | T | S | B | L | XL |
| Input resolution | $224^2$ | $224^2$ | $224^2$ | | | | | |
| Epochs | 300 | 300 | 300 | | | | | |
| Batch size | 1024 | 2048 | 1024 | | | | | |
| Optimizer | AdamW | LAMB | AdamW | | | | | |
| AdamW ($\beta_1, \beta_2$) | 0.9, 0.999 | - | 0.9, 0.999 | | | | | |
| Learning rate | 0.001 | 0.005 | 0.001 | | | | | |
| Learning rate decay | Cosine | Cosine | Cosine | | | | | |
| Weight decay | 0.05 | 0.02 | 0.03 | 0.04 | 0.05 | 0.05 | 0.05 | 0.05 |
| Warmup epochs | 5 | 5 | 5 | | | | | |
| Label smoothing $\epsilon$ | 0.1 | 0.1 | 0.1 | | | | | |
| Stochastic Depth | ✓ | ✓ | 0.05 | 0.1 | 0.1 | 0.2 | 0.3 | 0.4 |
| Rand Augment | 9/0.5 | 7/0.5 | 7/0.5 | 7/0.5 | 9/0.5 | 9/0.5 | 9/0.5 | 9/0.5 |
| Repeated Augment | ✓ | ✓ | ✗ | | | | | |
| Mixup $\alpha$ | 0.8 | 0.1 | 0.1 | 0.1 | 0.8 | 0.8 | 0.8 | 0.8 |
| CutMix $\alpha$ | 1.0 | 1.0 | 1.0 | | | | | |
| Erasing prob. | 0.25 | ✗ | 0.25 | | | | | |
| ColorJitter | ✗ | ✗ | ✗ | ✗ | 0.4 | 0.4 | 0.4 | 0.4 |
| Gradient Clipping | ✓ | ✗ | ✗ | | | | | |
| EMA decay | ✓ | ✗ | ✗ | ✗ | ✓ | ✓ | ✓ | ✓ |
| Test crop ratio | 0.875 | 0.95 | 0.90 | | | | | |

Table A2: Hyper-parameters for ImageNet-1K training of DeiT, RSB A2, and MogaNet. We use a similar setting as RSB for XL and T versions of MogaNet and DeiT for the other versions.

| Configuration | IN-21K PT | | | | IN-1K FT | | | |
|---|---|---|---|---|---|---|---|---|
| | S | B | L | XL | S | B | L | XL |
| Input resolution | $224^2$ | | | | $384^2$ | | | |
| Epochs | 90 | | | | 30 | | | |
| Batch size | 1024 | | | | 512 | | | |
| Optimizer | AdamW | | | | AdamW | | | |
| AdamW ($\beta_1, \beta_2$) | 0.9, 0.999 | | | | 0.9, 0.999 | | | |
| Learning rate | $1 \times 10^{-3}$ | | | | $5 \times 10^{-5}$ | | | |
| Learning rate decay | Cosine | | | | Cosine | | | |
| Weight decay | 0.05 | | | | 0.05 | | | |
| Warmup epochs | 5 | | | | 0 | | | |
| Label smoothing $\epsilon$ | 0.2 | | | | 0.1 | 0.1 | 0.2 | 0.2 |
| Stochastic Depth | 0 | 0.1 | 0.1 | 0.1 | 0.4 | 0.6 | 0.7 | 0.8 |
| Rand Augment | 9/0.5 | | | | 9/0.5 | | | |
| Repeated Augment | ✗ | | | | ✗ | | | |
| Mixup $\alpha$ | 0.8 | | | | ✗ | | | |
| CutMix $\alpha$ | 1.0 | | | | ✗ | | | |
| Erasing prob. | 0.25 | | | | 0.25 | | | |
| ColorJitter | 0.4 | | | | 0.4 | | | |
| Gradient Clipping | ✗ | | | | ✗ | | | |
| EMA decay | ✗ | | | | ✓ | | | |
| Test crop ratio | 0.90 | | | | 1.0 | | | |

Table A3: Detailed training recipe for ImageNet-21K pre-training (IN-21K PT) and ImageNet-1K fine-tuning (IN-1K FT) in high resolutions for MogaNet.

adopted for Transformer and ConvNet architectures. For all models, the default input image resolution is $224^2$ for training from scratch. We adopt $256^2$ resolutions for lightweight experiments according to MobileViT (Mehta & Rastegari, 2022). Taking training settings for the model with 25M or more parameters as the default, we train all MogaNet models for 300 epochs by AdamW (Loshchilov & Hutter, 2019) optimizer using a batch size of 1024, a basic learning rate of $1 \times 10^{-3}$, a weight decay of 0.05, and a Cosine learning rate scheduler (Loshchilov & Hutter, 2016) with 5 epochs of linear warmup (Devlin et al., 2018). As for augmentation and regularization techniques, we adopt most of the data augmentation and regularization strategies applied in DeiT training settings, including Random Resized Crop (RRC) and Horizontal flip (Szegedy et al., 2015), RandAugment (Cubuk et al., 2020), Mixup (Zhang et al., 2018), CutMix (Yun et al., 2019), random erasing (Zhong et al., 2020), ColorJitter (He et al., 2016), stochastic depth (Huang et al., 2016), and label smoothing (Szegedy et al., 2016). Similar to ConvNeXt (Liu et al., 2022b), we do not apply Repeated augmentation (Hoffer et al., 2020) and gradient clipping, which are designed for Transformers but do not enhance the performances of ConvNets while using Exponential Moving Average (EMA) (Polyak & Juditsky, 1992) with the decay rate of 0.9999 by default. We also remove additional augmentation strategies (Cubuk et al., 2019; Liu et al., 2022d; Li et al., 2021; Liu et al., 2022c), *e.g.,* PCA lighting (Krizhevsky et al., 2012) and AutoAugment (Cubuk et al., 2019). Since lightweight architectures (3∼10M parameters) tend to get under-fitted with strong augmentations and regularization, we adjust the training configurations for MogaNet-XT/T following (Mehta & Rastegari, 2022; Chen et al., 2022; Zhang et al., 2022b), including employing the weight decay of 0.03 and 0.04, Mixup with $\alpha$ of 0.1, and RandAugment of 7/0.5 for MogaNet-XT/T. Since EMA is proposed to stabilize the training process of large models, we also remove it for MogaNet-XT/T as a fair comparison. An increasing degree of stochastic depth path augmentation is employed for larger models. In evaluation, the top-1 accuracy using a single crop with a test crop ratio of 0.9 is reported as (Yuan et al., 2021b; Yu et al., 2022; Guo et al., 2023).

**ImageNet-21K.** Following ConvNeXt, we further provide the training recipe for ImageNet-21K (Deng et al., 2009) pre-training and ImageNet-1K fine-tuning with high resolutions in Table A3. EMA is removed in pre-training, while CutMix and Mixup are removed for fine-tuning.

## A.3 OBJECT DETECTION AND SEGMENTATION ON COCO

Following Swin (Liu et al., 2021) and PoolFormer (Yu et al., 2022), we evaluate objection detection and instance segmentation tasks on COCO (Lin et al., 2014) benchmark, which include 118K training images (*train2017*) and 5K validation images (*val2017*). We adopt RetinaNet (Lin et al., 2017b), Mask R-CNN (He et al., 2017), and Cascade Mask R-CNN (Cai & Vasconcelos, 2019) as the standard detectors and use ImageNet-1K pre-trained weights as the initialization of the backbones. As

for RetinaNet and Mask R-CNN, we employ AdamW (Loshchilov & Hutter, 2019) optimizer for training $1\times$ scheduler (12 epochs) with a basic learning rate of $1 \times 10^{-4}$ and a batch size of 16. As for Cascade Mask R-CNN, the $3\times$ training scheduler and multi-scale training resolutions (MS) are adopted. The pre-trained weights on ImageNet-1K and ImageNet-21K are used accordingly to initialize backbones. The shorter side of training images is resized to 800 pixels, and the longer side is resized to not more than 1333 pixels. We calculate the FLOPs of compared models at $800 \times 1280$ resolutions. Experiments of COCO detection are implemented on `MMDetection` (Chen et al., 2019) codebase and run on 8 NVIDIA A100 GPUs. View detailed results in Appendix D.2.

### A.4 SEMANTIC SEGMENTATION ON ADE20K

We evaluate semantic segmentation on ADE20K (Zhou et al., 2018) benchmark, which contains 20K training images and 2K validation images, covering 150 fine-grained semantic categories. We first adopt Semantic FPN (Kirillov et al., 2019) following PoolFormer (Yu et al., 2022) and Uniformer (Li et al., 2022a), which train models for 80K iterations by AdamW (Loshchilov & Hutter, 2019) optimizer with a basic learning rate of $2 \times 10^{-4}$, a batch size of 16, and a poly learning rate scheduler. Then, we utilize UperNet (Xiao et al., 2018b) following Swin (Liu et al., 2021), which employs AdamW optimizer using a basic learning rate of $6 \times 10^{-5}$, a weight decay of 0.01, a poly scheduler with a linear warmup of 1,500 iterations. We use ImageNet-1K and ImageNet-21K pre-trained weights to initialize the backbones accordingly. The training images are resized to $512^2$ resolutions, and the shorter side of testing images is resized to 512 pixels. We calculate the FLOPs of models at $800 \times 2048$ resolutions. Experiments of ADE20K segmentation are implemented on `MMSegmentation` (Contributors, 2020b) codebase and run on 8 NVIDIA A100 GPUs. View full comparison results in Appendix D.3.

### A.5 2D HUMAN POSE ESTIMATION ON COCO

We evaluate 2D human keypoints estimation tasks on COCO (Lin et al., 2014) benchmark based on Top-Down SimpleBaseline (Xiao et al., 2018a) (adding a Top-Down estimation head after the backbone) following PVT (Wang et al., 2021b) and UniFormer (Li et al., 2022a). We fine-tune all models for 210 epochs with Adam optimizer (Kingma & Ba, 2014) using a basic learning rate selected in $\{1 \times 10^{-3}, 5 \times 10^{-4}\}$, a multi-step learning rate scheduler decay at 170 and 200 epochs. ImageNet-1K pre-trained weights are used as the initialization of the backbones. The training and testing images are resized to $256 \times 192$ or $384 \times 288$ resolutions, and the FLOPs of models are calculated at both resolutions. COCO pose estimation experiments are implemented on `MMPose` (Contributors, 2020a) codebase and run on 8 NVIDIA A100 GPUs. View full experiment results in Appendix D.4.

### A.6 3D HUMAN POSE ESTIMATION

We evaluate MogaNet and popular architectures with 3D human pose estimation tasks with a single monocular image based on ExPose (Choutas et al., 2020). We first benchmark widely-used ConvNets with the 3D face mesh surface estimation task based on ExPose. All models are trained for 100 epochs on Flickr-Faces-HQ Dataset (FFHQ) (Karras et al., 2019) and tested on Stirling/ESRC 3D dataset (Feng et al., 2018), which consists of facial RGB images with ground-truth 3D face scans. 3D Root Mean Square Error (3DRMSE) measures errors between the predicted and ground-truth face scans. Following ExPose, the Adam optimizer is employed with a batch size of 256, a basic learning rate selected in $\{2 \times 10^{-4}, 1 \times 10^{-4}\}$, a multi-step learning rate scheduler decay at 60 and 100 epochs. ImageNet-1K pre-trained weights are adopted as the backbone initialization. The training and testing images are resized to $256 \times 256$ resolutions. Then, we evaluate ConvNets with the hand 3D pose estimation tasks. FreiHAND dataset (Zimmermann et al., 2019), which contains multi-view RGB hand images, 3D MANO hand pose, and shape annotations, is adopted for training and testing. Mean Per-Joint Position Error (PA-MPJPE) is used to evaluate 3D skeletons. Notice that a "PA" prefix denotes that the metric measures error after solving rotation, scaling, and translation transforms using Procrustes Alignment. Refer to ExPose for more implementation details. All models use the same training settings as the 3D face task, and the training and testing resolutions are $224 \times 224$. Experiments of 3D pose estimation are implemented on `MMHuman3D` (Contributors, 2021) codebase and run on 4 NVIDIA A100 GPUs. View full results in Appendix D.5.

## A.7 Video Prediction on Moving MNIST

We evaluate various Metaformer architectures (Yu et al., 2022) and MogaNet with video prediction tasks on Moving MNIST (MMNIST) (Lin et al., 2014) based on SimVP (Gao et al., 2022). Notice that the hidden translator of SimVP is a 2D network module to learn spatio-temporal representation, which any 2D architecture can replace. Therefore, we can benchmark various architectures based on the SimVP framework. In MMNIST (Srivastava et al., 2015), each video is randomly generated with 20 frames containing two digits in $64 \times 64$ resolutions, and the model takes 10 frames as the input to predict the next 10 frames. Video predictions are evaluated by Mean Square Error (MSE), Mean Absolute Error (MAE), and Structural Similarity Index (SSIM). All models are trained on MMNIST from scratch for 200 or 2000 epochs with Adam optimizer, a batch size of 16, a OneCycle learning rate scheduler, an initial learning rate selected in $\{1 \times 10^{-2}, 5 \times 10^{-3}, 1 \times 10^{-3}, 5 \times 10^{-4}\}$. Experiments of video prediction are implemented on `OpenSTL`[1] codebase (Tan et al., 2023) and run on a single NVIDIA Tesla V100 GPU. View full benchmark results in Appendix D.6.

# B Empirical Experiment Results

## B.1 Representation Bottleneck of DNNs from the View of Multi-order Interaction

**Multi-order game-theoretic interaction.** In Sec. 3, we interpret the learned representation of DNNs through the lens of multi-order game-theoretic interaction (Zhang et al., 2020; Deng et al., 2022), which disentangles inter-variable communication effects in a DNN into diverse game-theoretic components of different interaction orders. The order here denotes the *scale of context* involved in the whole computation process of game-theoretic interaction.

For computer vision, the $m$-th order interaction $I^{(m)}(i, j)$ measures the average game-theoretic interaction effects between image patches $i$ and $j$ on all $m$ image patch contexts. Take face recognition as an example, we can consider patches $i$ and $j$ as *two eyes* on this face. Besides, we regard other $m$ visible image patches included on the face. The interaction effect and contribution between the eye's patches $i$ and $j$ toward the task depend on such $m$ visible patches as the context, which is measured as the aforementioned $I^{(m)}(i, j)$. If $I^{(m)}(i, j) > 0$, patches $i$ and $j$ show a positive effect under $m$ context. Accordingly, if $I^{(m)}(i, j) < 0$, we consider $i$ and $j$ have a negative effect under $m$ context. More importantly, interactions of low-order mainly reflect **widely-shared local texture** and **common** visual concepts. The middle-order interactions are primarily responsible for encoding **discriminative high-level** representations. However, the high-order ones are inclined to let DNNs memorize the pattern of **rare outliers** and large-scale shape with **intensive global interactions**, which can presumably over-fit our deep models (Deng et al., 2022; Cheng et al., 2021). Consequently, the occurrence of *excessively low- or high-order* game-theoretic interaction in a deep architecture may therefore be undesirable.

Formally, given an input image $x$ with a set of $n$ patches $N = \{1, \ldots, n\}$ (*e.g.*, an image with $n$ pixels in total), the multi-order interaction $I^{(m)}(i, j)$ can be calculated as:

$$I^{(m)}(i, j) = \mathbb{E}_{S \subseteq N \setminus \{i,j\}, |S|=m}[\Delta f(i, j, S)], \tag{11}$$

where $\Delta f(i, j, S) = f(S \cup \{i, j\}) - f(S \cup \{i\}) - f(S \cup \{j\}) + f(S)$. $f(S)$ indicates the score of output with patches in $N \setminus S$ kept unchanged but replaced with the baseline value (Ancona et al., 2019a), For example, a low-order interaction (*e.g.*, $m = 0.05n$) means the relatively simple collaboration between variables $i, j$ under a small range of context, while a high-order interaction (*e.g.*, $m = 0.95n$) corresponds to the complex collaboration under a large range of context. Then, we can measure the overall interaction complexity of deep neural networks (DNNs) by the relative interaction strength $J^{(m)}$ of the encoded $m$-th order interaction:

$$J^{(m)} = \frac{\mathbb{E}_{x \in \Omega} \mathbb{E}_{i,j} |I^{(m)}(i, j|x)|}{\mathbb{E}_{m'} \mathbb{E}_{x \in \Omega} \mathbb{E}_{i,j} |I^{(m')}(i, j|x)|}, \tag{12}$$

where $\Omega$ is the set of all samples and $0 \leq m \geq n - 2$. Note that $J^{(m)}$ is the average interaction strength over all possible patch pairs of the input samples and indicates the distribution (area under

---

[1] https://github.com/chengtan9907/OpenSTL

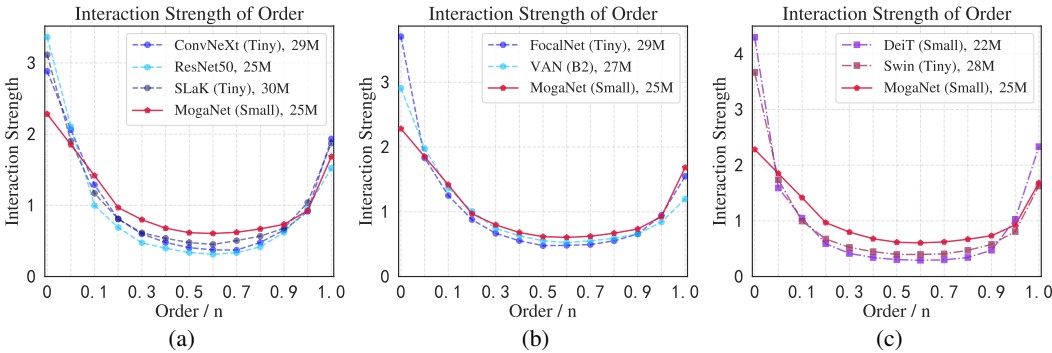

Figure A2: **Distributions of the interaction strength** $J^{(m)}$ for (a) ConvNets with different convolution kernel sizes, (b) ConvNets with gating aggregations, and (c) Transformers on ImageNet-1K with $224^2$ resolutions. Middle-order strengths mean the middle-complex interaction, where a medium number of patches (*e.g.,* 0.2∼0.8n) participate.

curve sums up to one) of the order of interactions of DNNs. In Fig. A2, we calculate the interaction strength $J^{(m)}$ with Eq. 12 for the models trained on ImageNet-1K using the official implementation[2] provided by (Deng et al., 2022). Specially, we use the image of $224 \times 224$ resolution as the input and calculate $J^{(m)}$ on $14 \times 14$ grids, *i.e.,* $n = 14 \times 14$. And we set the model output as $f(x_S) = \log \frac{P(\hat{y}=y|x_S)}{1-P(\hat{y}=y|x_S)}$ given the masked sample $x_S$, where $y$ denotes the ground-truth label and $P(\hat{y} = y|x_S)$ denotes the probability of classifying the masked sample $x_S$ to the true category. Fig. A2a and Fig. A2b compare existing ConvNets with large kernels or gating designs and demonstrate that MogaNet can model middle-order interactions better to learn more informative representations.

**Relationship of explaining works of ViTs.** Since the thriving of ViTs in a wide range of computer vision tasks, recent studies mainly investigate the *why* ViTs work from two directions: (a) Evaluation of robustness against noises finds that self-attentions (Naseer et al., 2021; Park & Kim, 2022; Zhou et al., 2021; Li et al., 2023a) or gating mechanisms (Zhou et al., 2022) in ViTs are more robust than classical convolutional operations (Simonyan & Zisserman, 2014; Szegedy et al., 2016). For example, ViTs can still recognize the target object with large occlusion ratios (*e.g.,* only 10∼20% visible patches) or corruption noises. This phenomenon might stem from the inherent redundancy of images and the competition property of self-attention mechanisms (Wang et al., 2021a; Wu et al., 2022b). Several recently proposed works (Yin et al., 2022; Graham et al., 2021) show that ViTs can work with some essential tokens (*e.g.,* 5∼50%) that are selected according to the complexity of input images by dynamic sampling strategies, which also utilize the feature selection properties of self-attentions. From the perspective of multi-order interactions, convolutions with local inductive bias (using small kernel sizes) prefer low-order interactions, while self-attentions without any inductive bias tend to learn low-order and high-order interactions. (b) Evaluation of out-of-distribution samples reveals that both self-attention mechanisms and depth-wise convolution (DWConv) with large kernel designs share similar shape-bias tendency as human vision (Tuli et al., 2021; Geirhos et al., 2021; Ding et al., 2022b), while canonical ConvNets (using convolutions with small kernel sizes) exhibit strong bias on local texture (Geirhos et al., 2019; Hermann et al., 2020). Current works (Ding et al., 2022b) attribute shape or texture-bias tendency to the receptive field of self-attention or convolution operations, *i.e.,* an operation with the larger receptive field or more long-range dependency is more likely to be shape-bias. However, there are still gaps between shape-bias operations and human vision. Human brains (Treisman & Gelade, 1980; Deng et al., 2022) attain visual patterns and clues and conduct middle-complexity interactions to recognize objects, while a self-attention or convolution operation can only encode global or local features to conduct high or low-complexity interactions. As the existing design of DNNs only stacks regionality perception or context aggregation operations in a cascaded way, it is inevitable to encounter the representation bottleneck.

## B.2 VISUALIZATION OF CAM

We further visualize more examples of Grad-CAM (Selvaraju et al., 2017) activation maps of MogaNet-S in comparison to Transformers, including DeiT-S (Touvron et al., 2021a), T2T-ViT-

---

[2]https://github.com/Nebularaid2000/bottleneck

S (Yuan et al., 2021b), Twins-S (Chu et al., 2021), and Swin (Liu et al., 2021), and ConvNets, including ResNet-50 (He et al., 2016) and ConvNeXt-T (Liu et al., 2022b), on ImageNet-1K in Fig. A4. Due to the self-attention mechanism, the pure Transformers architectures (DeiT-S and T2T-ViT-S) show more refined activation maps than ConvNets, but they also activate some irrelevant parts. Combined with the design of local windows, local attention architectures (Twins-S and Swin-T) can locate the full semantic objects. Results of previous ConvNets can roughly localize the semantic target but might contain some background regions. The activation parts of our proposed MogaNet-S are more similar to local attention architectures than previous ConvNets, which are more gathered on the semantic objects.

## C   MORE ABLATION AND ANALYSIS RESULTS

In addition to Sec. 5.3, we further conduct more ablation and analysis of our proposed MogaNet on ImageNet-1K. We adopt the same experimental settings as Sec. 1.

### C.1   ABLATION OF ACTIVATION FUNCTIONS

We conduct the ablation of activation functions used in the proposed multi-order gated aggregation module on ImageNet-1K. Table A4 shows that using SiLU (Elfwing et al., 2018) activation for both branches achieves the best performance. Similar results were also found in Transformers, *e.g.,* GLU variants with SiLU or GELU (Hendrycks & Gimpel, 2016) yield better performances than using Sigmoid or Tanh activation functions (Shazeer, 2020; Hua et al., 2022). We assume that SiLU is the most suitable activation because it owns both the property of Sigmoid (gating effects) and GELU (training friendly), which is defined as $x \cdot \mathrm{Sigmoid}(x)$.

| Top-1 | | Context branch | | |
|---|---|---|---|---|
| | Acc (%) | None | GELU | SiLU |
| | None | 76.3 | 76.7 | 76.7 |
| Gating | Sigmoid | 76.8 | 77.0 | 76.9 |
| branch | GELU | 76.7 | 76.8 | 77.0 |
| | SiLU | 76.9 | 77.1 | **77.2** |

Table A4:   Ablation of various activation functions for the gating and context branches in the proposed $\mathrm{Moga}(\cdot)$ module, which SiLU achieves the best performance in two branches.

| Modules | Top-1 Acc (%) | Params. (M) | FLOPs (G) |
|---|---|---|---|
| Baseline (+Gating branch) | 77.2 | 5.09 | 1.070 |
| $\mathrm{DW}_{7\times7}$ | 77.4 | 5.14 | 1.094 |
| $\mathrm{DW}_{5\times5,d=1} + \mathrm{DW}_{7\times7,d=3}$ | 77.5 | 5.15 | 1.112 |
| $\mathrm{DW}_{5\times5,d=1} + \mathrm{DW}_{5\times5,d=2} + \mathrm{DW}_{7\times7,d=3}$ | 77.5 | 5.17 | 1.185 |
| +Multi-order, $C_l : C_m : C_h = 1 : 0 : 3$ | 77.5 | 5.17 | 1.099 |
| +Multi-order, $C_l : C_m : C_h = 0 : 1 : 1$ | 77.6 | 5.17 | 1.103 |
| +Multi-order, $C_l : C_m : C_h = 1 : 6 : 9$ | 77.7 | 5.17 | 1.104 |
| +Multi-order, $C_l : C_m : C_h = 1 : 3 : 4$ | **77.8** | 5.17 | 1.102 |

Table A5: Ablation of multi-order DWConv layers in the proposed $\mathrm{Moga}(\cdot)$. The baseline adopts the MogaNet framework using the non-linear projection, $\mathrm{DW}_{5\times5}$, and the SiLU gating branch as $\mathrm{SMixer}(\cdot)$ and using the vanilla MLP as $\mathrm{CMixer}(\cdot)$.

### C.2   ABLATION OF MULTI-ORDER DWCONV LAYERS

In addition to Sec. 4.2 and Sec. 5.3, we also analyze the multi-order depth-wise convolution (DW-Conv) layers as the static regionality perception in the multi-order aggregation module $\mathrm{Moga}(\cdot)$ on ImageNet-1K. As shown in Table A5, we analyze the channel configuration of three parallel dilated DWConv layers: $\mathrm{DW}_{5\times5,d=1}$, $\mathrm{DW}_{5\times5,d=2}$, and $\mathrm{DW}_{7\times7,d=3}$ with the channels of $C_l$, $C_m$, $C_h$. we first compare the performance of serial DWConv layers (*e.g.,* $\mathrm{DW}_{5\times5,d=1}+\mathrm{DW}_{7\times7,d=3}$) and parallel DWConv layers. We find that the parallel design can achieve the same performance with fewer computational overloads because the DWConv kernel is equally applied to all channels. When we adopt three DWConv layers, the proposed parallel design reduces $C_l + C_h$ and $C_l + C_m$ times computations of $\mathrm{DW}_{5\times5,d=2}$ and $\mathrm{DW}_{5\times5,d=2}$ in comparison to the serial stack of these DWConv layers. Then, we empirically explore the optimal configuration of the three channels. We find that $C_l : C_m : C_h = 1: 3: 4$ yields the best performance, which well balances the small, medium, and large DWConv kernels to learn low, middle, and high-order contextual representations. We calculate and discuss the FLOPs of the proposed three DWConv layers in the next subsection to verify the efficiency. Similar conclusions are also found in relevant designs (Pan et al., 2022a; Si et al., 2022; Rao et al., 2022), where global context aggregations take the majority (*e.g.,* $\frac{1}{2} \sim \frac{3}{4}$ channels or context components). We also verify the parallel design with the optimal configuration based

on MogaNet-S/B. Therefore, we can conclude that our proposed multi-order DWConv layers can efficiently learn multi-order contextual information for the context branch of $\text{Moga}(\cdot)$.

### C.3 FLOPs and Throughputs of MogaNet

**FLOPs of Multi-order Gated Aggregation Module**   We divide the computation of the proposed multi-order gated aggregation module into two parts of convolution operations and calculate the FLOPs for each part.

- **Conv1×1.** The FLOPs of $1 \times 1$ convolution operation $\phi_{\text{gate}}$ , $\phi_{\text{context}}$ and $\phi_{\text{out}}$ can be derived as:

$$\text{FLOPs}(\phi_{\text{gate}}) = 2HWC^2,$$
$$\text{FLOPs}(\phi_{\text{context}}) = 2HWC^2, \tag{13}$$
$$\text{FLOPs}(\phi_{\text{out}}) = 2HWC^2.$$

- **Depth-wise convolution.** We consider the depth-wise convolution (DW) with dilation ratio $d$. The DWConv is performed for the input $X$, where $X \in \mathbb{R}^{HW \times C_{in}}$. Therefore, the FLOPs for all DW in Moga module are:

$$\text{FLOPs}(\text{DW}_{5\times5, d=1}) = 2HWC_{in}K_{5\times5}^2,$$
$$\text{FLOPs}(\text{DW}_{5\times5, d=2}) = \frac{3}{4}HWC_{in}K_{5\times5}^2, \tag{14}$$
$$\text{FLOPs}(\text{DW}_{7\times7, d=3}) = HWC_{in}K_{7\times7}^2.$$

Overall, the total FLOPs of our Moga module can be derived as follows:

$$\text{FLOPs}(\text{Moga}) = 2HWC_{in}\left[\frac{11}{8}K_{5\times5}^2 + \frac{1}{2}K_{7\times7}^2 + 3C_{in}\right]$$
$$= HWC_{in}\left[\frac{471}{4} + 6C_{in}\right]. \tag{15}$$

**Throughput of MogaNet**   We further analyze throughputs of MogaNet variants on ImageNet-1K. As shown in Fig. A3, MogaNet has similar throughputs as Swin Transformer while producing better performances than Swin and ConvNet. Since we add channel splitting and GAP operations in MogaNet, the throughput of ConvNeXt exceeds MogaNet to some extent.

### C.4 Ablation of Normalization Layers

For most ConvNets, BatchNorm (Ioffe & Szegedy, 2015b) (BN) is considered an essential component to improve the convergence speed and prevent overfitting. However, BN might cause some instability (Wu

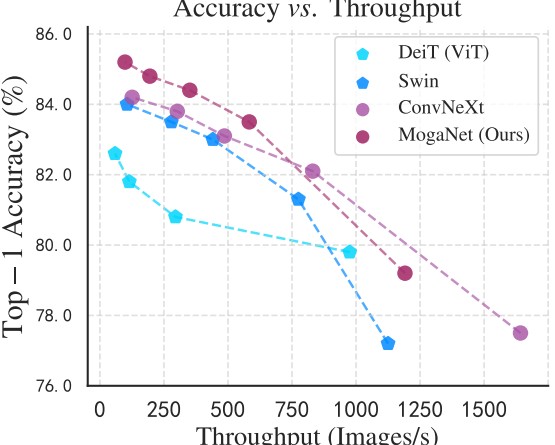

Figure A3: Accuracy-throughput diagram of models on ImageNet-1K measured on an NVIDIA V100 GPU.

& Johnson, 2021) or harm the final performance of models (Brock et al., 2021a;b). Some recently proposed ConvNets (Liu et al., 2022b; Guo et al., 2023) replace BN by LayerNorm (Ba et al., 2016) (LN), which has been widely used in Transformers (Dosovitskiy et al., 2021) and Metaformer architectures (Yu et al., 2022), achieving relatively good performances in various scenarios. Here, we conduct an ablation of normalization (Norm) layers in MogaNet on ImageNet-1K, as shown in Table A6. As discussed in ConvNeXt (Liu et al., 2022b), the Norm layers used in each block (**within**) and after each stage (**after**) have different effects. Thus we study them separately. Table A6 shows that using BN in both places yields better performance than using LN (after) and BN (within), except MogaNet-T with $224^2$ resolutions, while using LN in both places performs the worst. Consequently,

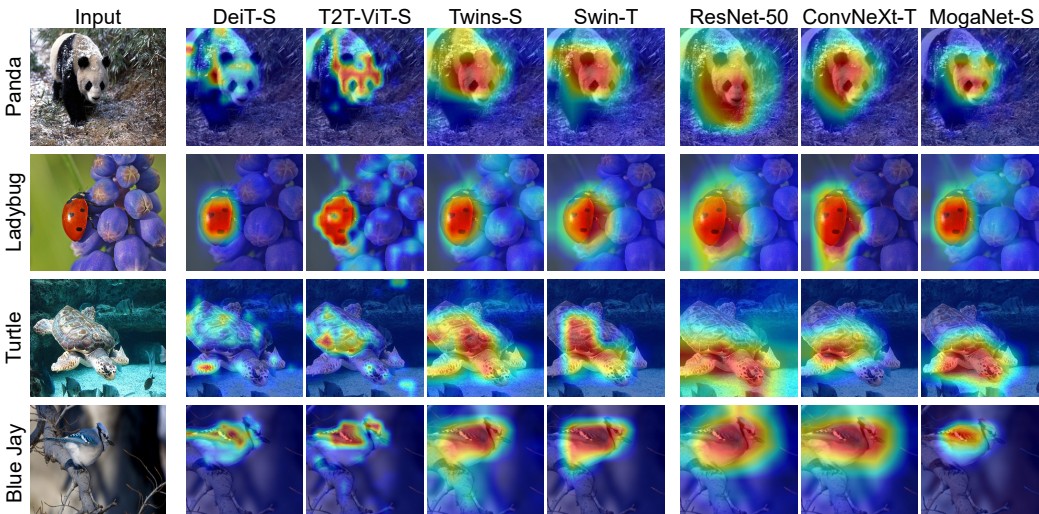

Figure A4: Visualization of Grad-CAM activation maps of the models trained on ImageNet-1K.

we use BN as the default Norm layers in our proposed MogaNet for two reasons: (i) With pure convolution operators, the rule of combining convolution operations with BN within each stage is still useful for modern ConvNets. (ii) Although using LN after each stage might help stabilize the training process of Transformers and hybrid models and might sometimes bring good performance for ConvNets, adopting BN after each stage in pure convolution models still yields better performance. Moreover, we replace BN with precise BN (Wu & Johnson, 2021) (pBN), which is an optimal alternative normalization strategy to BN. We find slight performance improvements (around 0.1%), especially when MogaNet-S/B adopts the EMA strategy (by default), indicating that we can further improve MogaNet with advanced BN. As discussed in ConvNeXt, EMA might severely hurt the performances of models with BN. This phenomenon might be caused by the unstable and inaccurate BN statistics estimated by EMA in the vanilla BN with large models, which will deteriorate when using another EMA of model parameters. We solve this dilemma by exponentially increasing the EMA decay from 0.9 to 0.9999 during training as momentum-based contrastive learning methods (Caron et al., 2021; Bao et al., 2022), *e.g.,* BYOL (Grill et al., 2020). It can also be tackled by advanced BN variants (Hoffer et al., 2017; Wu & Johnson, 2021).

| Norm (after)
Norm (within) | Input
size | LN
LN | LN
BN | BN
BN | pBN
pBN |
|---|---|---|---|---|---|
| MogaNet-T | $224^2$ | 78.4 | **79.1** | 79.0 | **79.1** |
| MogaNet-T | $256^2$ | 78.8 | 79.4 | **79.6** | **79.6** |
| MogaNet-S | $224^2$ | 82.5 | 83.2 | **83.3** | **83.3** |
| MogaNet-S (EMA) | $224^2$ | 82.7 | 83.2 | 83.3 | **83.4** |
| MogaNet-B | $224^2$ | 83.4 | 83.9 | 84.1 | **84.2** |
| MogaNet-B (EMA) | $224^2$ | 83.7 | 83.8 | 84.3 | **84.4** |

Table A6: Ablation of normalization layers in MogaNet.

## C.5 REFINED TRAINING SETTINGS FOR LIGHTWEIGHT MODELS

To explore the full power of lightweight models of our MogaNet, we refined the basic training settings for MogaNet-XT/T according to RSB A2 (Wightman et al., 2021) and DeiT-III (Touvron et al., 2022). Compared to the default setting as provided in Table A2, we only adjust the learning rate and the augmentation strategies for faster convergence while keeping other settings unchanged. As shown in Table A7, MogaNet-XT/T gain +0.4~0.6% when use the large learning rate of $2 \times 10^{-3}$ and 3-Augment (Touvron et al., 2022) without complex designs. Based on the advanced setting, MogaNet with $224^2$ input resolutions yields significant performance improvements against previous methods, *e.g.,* MogaNet-T gains +3.5% over DeiT-T (Touvron et al., 2021a) and +1.2% over Parc-Net-S (Zhang et al., 2022b). Especially, MogaNet-T with $256^2$ resolutions achieves top-1 accuracy of 80.0%, outperforming DeiT-S of 79.8% reported in the original paper, while MogaNet-XT with

| Architecture | Input size | Learning rate | Warmup epochs | Rand Augment | 3-Augment | EMA | Top-1 Acc (%) |
|---|---|---|---|---|---|---|---|
| DeiT-T | $224^2$ | $1 \times 10^{-3}$ | 5 | 9/0.5 | ✗ | ✓ | 72.2 |
| DeiT-T | $224^2$ | $2 \times 10^{-3}$ | 20 | ✗ | ✓ | ✗ | 75.9 |
| ParC-Net-S | $256^2$ | $1 \times 10^{-3}$ | 5 | 9/0.5 | ✗ | ✓ | 78.6 |
| ParC-Net-S | $256^2$ | $2 \times 10^{-3}$ | 20 | ✗ | ✓ | ✗ | 78.8 |
| MogaNet-XT | $224^2$ | $1 \times 10^{-3}$ | 5 | 7/0.5 | ✗ | ✗ | 76.5 |
| MogaNet-XT | $224^2$ | $2 \times 10^{-3}$ | 20 | ✗ | ✓ | ✗ | 77.1 |
| MogaNet-XT | $256^2$ | $1 \times 10^{-3}$ | 5 | 7/0.5 | ✗ | ✗ | 77.2 |
| MogaNet-XT | $256^2$ | $2 \times 10^{-3}$ | 20 | ✗ | ✓ | ✗ | 77.6 |
| MogaNet-T | $224^2$ | $1 \times 10^{-3}$ | 5 | 7/0.5 | ✗ | ✗ | 79.0 |
| MogaNet-T | $224^2$ | $2 \times 10^{-3}$ | 20 | ✗ | ✓ | ✗ | 79.4 |
| MogaNet-T | $256^2$ | $1 \times 10^{-3}$ | 5 | 7/0.5 | ✗ | ✗ | 79.6 |
| MogaNet-T | $256^2$ | $2 \times 10^{-3}$ | 20 | ✗ | ✓ | ✗ | **80.0** |

Table A7: Advanced training recipes for Lightweight models of MogaNet on ImageNet-1K.

| Architecture | Type | #P. (M) | FLOPs (G) | AP | $AP_{50}$ | $AP_{75}$ | $AP^S$ | $AP_M$ | $AP_L$ |
|---|---|---|---|---|---|---|---|---|---|
| | | | | | | RetinaNet 1× | | | |
| RegNet-800M | C | 17 | 168 | 35.6 | 54.7 | 37.7 | 19.7 | 390 | 47.8 |
| PVTV2-B0 | T | 13 | 160 | 37.1 | 57.2 | 39.2 | 23.4 | 40.4 | 49.2 |
| **MogaNet-XT** | C | 12 | 167 | **39.7** | **60.0** | **42.4** | **23.8** | **43.6** | **51.7** |
| ResNet-18 | C | 21 | 189 | 31.8 | 49.6 | 33.6 | 16.3 | 34.3 | 43.2 |
| RegNet-1.6G | C | 20 | 185 | 37.4 | 56.8 | 39.8 | 22.4 | 41.1 | 49.2 |
| RegNet-3.2G | C | 26 | 218 | 39.0 | 58.4 | 41.9 | 22.6 | 43.5 | 50.8 |
| PVT-T | T | 23 | 183 | 36.7 | 56.9 | 38.9 | 22.6 | 38.8 | 50.0 |
| PoolFormer-S12 | T | 22 | 207 | 36.2 | 56.2 | 38.2 | 20.8 | 39.1 | 48.0 |
| PVTV2-B1 | T | 24 | 187 | 41.1 | 61.4 | 43.8 | 26.0 | 44.6 | 54.6 |
| **MogaNet-T** | C | 14 | 173 | **41.4** | **61.5** | **44.4** | **25.1** | **45.7** | **53.6** |
| ResNet-50 | C | 37 | 239 | 36.3 | 55.3 | 38.6 | 19.3 | 40.0 | 48.8 |
| Swin-T | T | 38 | 245 | 41.8 | 62.6 | 44.7 | 25.2 | 45.8 | 54.7 |
| PVT-S | T | 34 | 226 | 40.4 | 61.3 | 43.0 | 25.0 | 42.9 | 55.7 |
| Twins-SVT-S | T | 34 | 209 | 42.3 | 63.4 | 45.2 | 26.0 | 45.5 | 56.5 |
| Focal-T | T | 39 | 265 | 43.7 | - | - | - | - | - |
| PoolFormer-S36 | T | 41 | 272 | 39.5 | 60.5 | 41.8 | 22.5 | 42.9 | 52.4 |
| PVTV2-B2 | T | 35 | 281 | 44.6 | 65.7 | 47.6 | 28.6 | 48.5 | 59.2 |
| CMT-S | H | 45 | 231 | 44.3 | 65.5 | 47.5 | 27.1 | 48.3 | 59.1 |
| **MogaNet-S** | C | 35 | 253 | **45.8** | **66.6** | **49.0** | **29.1** | **50.1** | **59.8** |
| ResNet-101 | C | 57 | 315 | 38.5 | 57.8 | 41.2 | 21.4 | 42.6 | 51.1 |
| PVT-M | T | 54 | 258 | 41.9 | 63.1 | 44.3 | 25.0 | 44.9 | 57.6 |
| Focal-S | T | 62 | 367 | 45.6 | - | - | - | - | - |
| PVTV2-B3 | T | 55 | 263 | 46.0 | 67.0 | 49.5 | 28.2 | 50.0 | 61.3 |
| PVTV2-B4 | T | 73 | 315 | 46.3 | 67.0 | 49.6 | 29.0 | 50.1 | 62.7 |
| **MogaNet-B** | C | 54 | 355 | **47.7** | **68.9** | **51.0** | **30.5** | **52.2** | **61.7** |
| ResNeXt-101-64 | C | 95 | 473 | 41.0 | 60.9 | 44.0 | 23.9 | 45.2 | 54.0 |
| PVTV2-B5 | T | 92 | 335 | 46.1 | 66.6 | 49.5 | 27.8 | 50.2 | 62.0 |
| **MogaNet-L** | C | 92 | 477 | **48.7** | **69.5** | **52.6** | **31.5** | **53.4** | **62.7** |

Table A8: **Object detection** with RetinaNet (1× training schedule) on COCO *val2017*. The FLOPs are measured at resolution $800 \times 1280$.

$224^2$ resolutions outperforms DeiT-T under the refined training scheme by 1.2% with only 3M parameters.

# D  MORE COMPARISON EXPERIMENTS

## D.1  FAST TRAINING ON IMAGENET-1K

In addition to Sec. 5.1, we further provide comparison results for 100 and 300 epochs training on ImageNet-1K. As for 100-epoch training, we adopt the original RSB A3 (Wightman et al., 2021) setting for all methods, which adopts LAMB (You et al., 2020) optimizer and a small training resolution of $160^2$. We search the basic learning in $\{0.006, 0.008\}$ for all architectures and adopt the gradient clipping for Transformer-based networks. As for 300-epoch training, we report results of

| Architecture | Type | #P. (M) | FLOPs (G) | $AP^b$ | $AP^b_{50}$ | $AP^b_{75}$ | $AP^m$ | $AP^m_{50}$ | $AP^m_{75}$ |
|---|---|---|---|---|---|---|---|---|---|
| | | | | | | Mask R-CNN 1× | | | |
| RegNet-800M | C | 27 | 187 | 37.5 | 57.9 | 41.1 | 34.3 | 56.0 | 36.8 |
| **MogaNet-XT** | C | 23 | 185 | **40.7** | **62.3** | **44.4** | **37.6** | **59.6** | **40.2** |
| ResNet-18 | C | 31 | 207 | 34.0 | 54.0 | 36.7 | 31.2 | 51.0 | 32.7 |
| RegNet-1.6G | C | 29 | 204 | 38.9 | 60.5 | 43.1 | 35.7 | 57.4 | 38.9 |
| PVT-T | T | 33 | 208 | 36.7 | 59.2 | 39.3 | 35.1 | 56.7 | 37.3 |
| PoolFormer-S12 | T | 32 | 207 | 37.3 | 59.0 | 40.1 | 34.6 | 55.8 | 36.9 |
| **MogaNet-T** | C | 25 | 192 | **42.6** | **64.0** | **46.4** | **39.1** | **61.3** | **42.0** |
| ResNet-50 | C | 44 | 260 | 38.0 | 58.6 | 41.4 | 34.4 | 55.1 | 36.7 |
| RegNet-6.4G | C | 45 | 307 | 41.1 | 62.3 | 45.2 | 37.1 | 59.2 | 39.6 |
| PVT-S | T | 44 | 245 | 40.4 | 62.9 | 43.8 | 37.8 | 60.1 | 40.3 |
| Swin-T | T | 48 | 264 | 42.2 | 64.6 | 46.2 | 39.1 | 61.6 | 42.0 |
| MViT-T | T | 46 | 326 | 45.9 | **68.7** | 50.5 | 42.1 | **66.0** | 45.4 |
| PoolFormer-S36 | T | 32 | 207 | 41.0 | 63.1 | 44.8 | 37.7 | 60.1 | 40.0 |
| Focal-T | T | 49 | 291 | 44.8 | 67.7 | 49.2 | 41.0 | 64.7 | 44.2 |
| PVTV2-B2 | T | 45 | 309 | 45.3 | 67.1 | 49.6 | 41.2 | 64.2 | 44.4 |
| LITV2-S | T | 47 | 261 | 44.9 | 67.0 | 49.5 | 40.8 | 63.8 | 44.2 |
| CMT-S | H | 45 | 249 | 44.6 | 66.8 | 48.9 | 40.7 | 63.9 | 43.4 |
| Conformer-S/16 | H | 58 | 341 | 43.6 | 65.6 | 47.7 | 39.7 | 62.6 | 42.5 |
| Uniformer-S | H | 41 | 269 | 45.6 | 68.1 | 49.7 | 41.6 | 64.8 | 45.0 |
| ConvNeXt-T | C | 48 | 262 | 44.2 | 66.6 | 48.3 | 40.1 | 63.3 | 42.8 |
| FocalNet-T (SRF) | C | 49 | 267 | 45.9 | 68.3 | 50.1 | 41.3 | 65.0 | 44.3 |
| FocalNet-T (LRF) | C | 49 | 268 | 46.1 | 68.2 | 50.6 | 41.5 | 65.1 | 44.5 |
| **MogaNet-S** | C | 45 | 272 | **46.7** | 68.0 | **51.3** | **42.2** | 65.4 | **45.5** |
| ResNet-101 | C | 63 | 336 | 40.4 | 61.1 | 44.2 | 36.4 | 57.7 | 38.8 |
| RegNet-12G | C | 64 | 423 | 42.2 | 63.7 | 46.1 | 38.0 | 60.5 | 40.5 |
| PVT-M | T | 64 | 302 | 42.0 | 64.4 | 45.6 | 39.0 | 61.6 | 42.1 |
| Swin-S | T | 69 | 354 | 44.8 | 66.6 | 48.9 | 40.9 | 63.4 | 44.2 |
| Focal-S | T | 71 | 401 | 47.4 | 69.8 | 51.9 | 42.8 | 66.6 | 46.1 |
| PVTV2-B3 | T | 65 | 397 | 47.0 | 68.1 | 51.7 | 42.5 | 65.7 | 45.7 |
| LITV2-M | T | 68 | 315 | 46.5 | 68.0 | 50.9 | 42.0 | 65.1 | 45.0 |
| UniFormer-B | H | 69 | 399 | 47.4 | 69.7 | 52.1 | 43.1 | 66.0 | 46.5 |
| ConvNeXt-S | C | 70 | 348 | 45.4 | 67.9 | 50.0 | 41.8 | 65.2 | 45.1 |
| **MogaNet-B** | C | 63 | 373 | **47.9** | **70.0** | **52.7** | **43.2** | **67.0** | **46.6** |
| Swin-B | T | 107 | 496 | 46.9 | 69.6 | 51.2 | 42.3 | 65.9 | 45.6 |
| PVTV2-B5 | T | 102 | 557 | 47.4 | 68.6 | 51.9 | 42.5 | 65.7 | 46.0 |
| ConvNeXt-B | C | 108 | 486 | 47.0 | 69.4 | 51.7 | 42.7 | 66.3 | 46.0 |
| FocalNet-B (SRF) | C | 109 | 496 | 48.8 | 70.7 | 53.5 | 43.3 | 67.5 | 46.5 |
| **MogaNet-L** | C | 102 | 495 | **49.4** | **70.7** | **54.1** | **44.1** | **68.1** | **47.6** |

Table A9: **Object detection and instance segmentation** with Mask R-CNN (1× training schedule) on COCO *val2017*. The FLOPs are measured at resolution $800 \times 1280$.

RSB A2 (Wightman et al., 2021) for classical CNN or the original setting for Transformers or modern ConvNets. In Table A15, when compared with models of similar parameter size, our proposed MogaNet-XT/T/S/B achieves the best performance in both 100 and 300 epochs training. Results of 100-epoch training show that MogaNet has a faster convergence speed than previous architectures of various types. For example, MogaNet-T outperforms EfficientNet-B0 and DeiT-T by 2.4% and 8.7%, MogaNet-S outperforms Swin-T by 3.4%, and MogaNet-B outperforms Swin-S by 2.0%. Notice that ConvNeXt variants have a great convergence speed, *e.g.,* ConvNeXt-S achieves 81.7% surpassing Swin-S by 1.5 and recently proposed ConvNet HorNet-$S_{7\times7}$ by 0.5 with similar parameters. But our proposed MogaNet convergences faster than ConvNet, *e.g.,* MogaNet-S outperforms ConvNeXt-T by 2.3% with similar parameters while MogaNet-B/L reaching competitive performances as ConvNeXt-B/L with only 44∼50% parameters.

## D.2   DETECTION AND SEGMENTATION RESULTS ON COCO

In addition to Sec. 5.2, we provide full results of object detection and instance segmentation tasks with RetinaNet, Mask R-CNN, and Cascade Mask R-CNN on COCO. As shown in Table A8 and Table A9, RetinaNet or Mask R-CNN with MogaNet variants outperforms existing models when training 1× schedule. For example, RetinaNet with MogaNet-T/S/B/L achieve 45.8/47.7/48.7 $AP^b$, outperforming PVT-T/S/M and PVTV2-B1/B2/B3/B5 by 4.7/4.6/5.8 and 0.3/1.2/1.7/2.6 $AP^b$; Nask R-CNN with MogaNet-S/B/L achieve 46.7/47.9/49.4 $AP^b$, exceeding Swin-T/S/B and ConvNeXt-

| Architecture | Type | #P. (M) | FLOPs (G) | $AP^{bb}$ | $AP^b_{50}$ | $AP^b_{75}$ | $AP^m$ | $AP^m_{50}$ | $AP^m_{75}$ |
|---|---|---|---|---|---|---|---|---|---|
| ResNet-50 | C | 77 | 739 | 46.3 | 64.3 | 50.5 | 40.1 | 61.7 | 43.4 |
| Swin-T | T | 86 | 745 | 50.4 | 69.2 | 54.7 | 43.7 | 66.6 | 47.3 |
| Focal-T | T | 87 | 770 | 51.5 | 70.6 | 55.9 | - | - | - |
| ConvNeXt-T | C | 86 | 741 | 50.4 | 69.1 | 54.8 | 43.7 | 66.5 | 47.3 |
| FocalNet-T (SRF) | C | 86 | 746 | 51.5 | 70.1 | 55.8 | 44.6 | 67.7 | 48.4 |
| **MogaNet-S** | C | 78 | 750 | **51.6** | **70.8** | **56.3** | **45.1** | **68.7** | **48.8** |
| ResNet-101-32 | C | 96 | 819 | 48.1 | 66.5 | 52.4 | 41.6 | 63.9 | 45.2 |
| Swin-S | T | 107 | 838 | 51.9 | 70.7 | 56.3 | 45.0 | 68.2 | 48.8 |
| ConvNeXt-S | C | 108 | 827 | 51.9 | 70.8 | 56.5 | 45.0 | 68.4 | 49.1 |
| **MogaNet-B** | C | 101 | 851 | **52.6** | **72.0** | **57.3** | **46.0** | **69.6** | **49.7** |
| Swin-B | T | 145 | 982 | 51.9 | 70.5 | 56.4 | 45.0 | 68.1 | 48.9 |
| ConvNeXt-B | C | 146 | 964 | 52.7 | 71.3 | 57.2 | 45.6 | 68.9 | 49.5 |
| **MogaNet-L** | C | 140 | 974 | **53.3** | **71.8** | **57.8** | **46.1** | **69.2** | **49.8** |
| Swin-L[‡] | T | 253 | 1382 | 53.9 | 72.4 | 58.8 | 46.7 | 70.1 | 50.8 |
| ConvNeXt-L[‡] | C | 255 | 1354 | 54.8 | 73.8 | 59.8 | 47.6 | 71.3 | 51.7 |
| ConvNeXt-XL[‡] | C | 407 | 1898 | 55.2 | 74.2 | 59.9 | 47.7 | 71.6 | 52.2 |
| RepLKNet-31L[‡] | C | 229 | 1321 | 53.9 | 72.5 | 58.6 | 46.5 | 70.0 | 50.6 |
| HorNet-L[‡] | C | 259 | 1399 | 56.0 | - | - | 48.6 | - | - |
| **MogaNet-XL[‡]** | C | 238 | 1355 | **56.2** | **75.0** | **61.2** | **48.8** | **72.6** | **53.3** |

Table A10: **Object detection and instance segmentation** with Cascade Mask R-CNN ($3\times$ training schedule) with multi-scaling training (MS) on COCO *val2017*. [‡] denotes the model is pre-trained on ImageNet-21K. The FLOPs are measured at resolution $800 \times 1280$.

T/S/B by 4.5/3.1/2.5 and 2.5/2.5/2.4 with similar parameters and computational overloads. Noticeably, MogaNet-XT/T can achieve better detection results with fewer parameters and lower FLOPs than lightweight architectures, while MogaNet-T even surpasses some Transformers like Swin-S and PVT-S. For example, Mask R-CNN with MogaNet-T improves Swin-T by 0.4 $AP^b$ and outperforms PVT-S by 1.3 $AP^m$ using only around 2/3 parameters. As shown in Table A10, Cascade Mask R-CNN with MogaNet variants still achieves the state-of-the-art detection and segmentation results when training $3\times$ schedule with multi-scaling (MS) and advanced augmentations. For example, MogaNet-L/XL yield 53.3/56.2 $AP^b$ and 46.1/48.8 $AP^m$, which improves Swin-B/L and ConvNeXt-B/L by 1.4/2.3 and 0.6/1.4 $AP^b$ with similar parameters and FLOPS.

### D.3 Semantic Segmentation Results on ADE20K

In addition to Sec. 5.2, we provide comprehensive comparison results of semantic segmentation based on UperNet on ADE20K. As shown in Table A11, UperNet with MogaNet produces state-of-the-art performances in a wide range of parameter scales compared to famous Transformer, hybrid, and convolution models. As for the lightweight models, MogaNet-XT/T significantly improves ResNet-18/50 with fewer parameters and FLOPs budgets. As for medium-scaling models, MogaNet-S/B achieves 49.2/50.1 $mIoU^{ss}$, which outperforms the recently proposed ConvNets, *e.g.,* +1.1 over HorNet-T using similar parameters and +0.7 over SLaK-S using 17M fewer parameters. As for large models, MogaNet-L/XL surpass Swin-B/L and ConvNeXt-B/L by 1.2/1.9 and 1.8/0.3 $mIoU^{ss}$ while using fewer parameters.

### D.4 2D Human Pose Estimation Results on COCO

In addition to Sec. 5.2, we provide comprehensive experiment results of 2D human key points estimation based on Top-Down SimpleBaseline on COCO. As shown in Table A13, MogaNet variants achieve competitive or state-of-the-art performances compared to popular architectures with two types of resolutions. As for lightweight models, MogaNet-XT/T significantly improves the performances of existing models while using similar parameters and FLOPs. Meanwhile, MogaNet-S/B also produces 74.9/75.3 and 76.4/77.3 AP using $256 \times 192$ and $384 \times 288$ resolutions, outperforming Swin-B/L by 2.0/1.0 and 1.5/1.0 AP with nearly half of the parameters and computation budgets.

| Architecture | Date | Type | Crop size | Param. (M) | FLOPs (G) | mIoU$^{ss}$ (%) |
|---|---|---|---|---|---|---|
| ResNet-18 | CVPR'2016 | C | $512^2$ | 41 | 885 | 39.2 |
| **MogaNet-XT** | Ours | C | $512^2$ | 30 | 856 | **42.2** |
| ResNet-50 | CVPR'2016 | C | $512^2$ | 67 | 952 | 42.1 |
| **MogaNet-T** | Ours | C | $512^2$ | 33 | 862 | **43.7** |
| DeiT-S | ICML'2021 | T | $512^2$ | 52 | 1099 | 44.0 |
| Swin-T | ICCV'2021 | T | $512^2$ | 60 | 945 | 46.1 |
| TwinsP-S | NIPS'2021 | T | $512^2$ | 55 | 919 | 46.2 |
| Twins-S | NIPS'2021 | T | $512^2$ | 54 | 901 | 46.2 |
| Focal-T | NIPS'2021 | T | $512^2$ | 62 | 998 | 45.8 |
| Uniformer-S$_{h32}$ | ICLR'2022 | H | $512^2$ | 52 | 955 | 47.0 |
| UniFormer-S | ICLR'2022 | H | $512^2$ | 52 | 1008 | 47.6 |
| ConvNeXt-T | CVPR'2022 | C | $512^2$ | 60 | 939 | 46.7 |
| FocalNet-T (SRF) | NIPS'2022 | C | $512^2$ | 61 | 944 | 46.5 |
| HorNet-T$_{7\times7}$ | NIPS'2022 | C | $512^2$ | 52 | 926 | 48.1 |
| **MogaNet-S** | Ours | C | $512^2$ | 55 | 946 | **49.2** |
| Swin-S | ICCV'2021 | T | $512^2$ | 81 | 1038 | 48.1 |
| Twins-B | NIPS'2021 | T | $512^2$ | 89 | 1020 | 47.7 |
| Focal-S | NIPS'2021 | T | $512^2$ | 85 | 1130 | 48.0 |
| Uniformer-B$_{h32}$ | ICLR'2022 | H | $512^2$ | 80 | 1106 | 49.5 |
| ConvNeXt-S | CVPR'2022 | C | $512^2$ | 82 | 1027 | 48.7 |
| FocalNet-S (SRF) | NIPS'2022 | C | $512^2$ | 83 | 1035 | 49.3 |
| SLaK-S | ICLR'2023 | C | $512^2$ | 91 | 1028 | 49.4 |
| **MogaNet-B** | Ours | C | $512^2$ | 74 | 1050 | **50.1** |
| Swin-B | ICCV'2021 | T | $512^2$ | 121 | 1188 | 49.7 |
| Focal-B | NIPS'2021 | T | $512^2$ | 126 | 1354 | 49.0 |
| ConvNeXt-B | CVPR'2022 | C | $512^2$ | 122 | 1170 | 49.1 |
| RepLKNet-31B | CVPR'2022 | C | $512^2$ | 112 | 1170 | 49.9 |
| FocalNet-B (SRF) | NIPS'2022 | C | $512^2$ | 124 | 1180 | 50.2 |
| SLaK-B | ICLR'2023 | C | $512^2$ | 135 | 1185 | 50.2 |
| **MogaNet-L** | Ours | C | $512^2$ | 113 | 1176 | **50.9** |
| Swin-L$^{\ddagger}$ | ICCV'2021 | T | $640^2$ | 234 | 2468 | 52.1 |
| ConvNeXt-L$^{\ddagger}$ | CVPR'2022 | C | $640^2$ | 245 | 2458 | 53.7 |
| RepLKNet-31L$^{\ddagger}$ | CVPR'2022 | C | $640^2$ | 207 | 2404 | 52.4 |
| **MogaNet-XL$^{\ddagger}$** | Ours | C | $640^2$ | 214 | 2451 | **54.0** |

Table A11: **Semantic segmentation** with UperNet (160K) on ADE20K validation set. $^{\ddagger}$ indicates using IN-21K pre-trained models. The FLOPs are measured at $512 \times 2048$ or $640 \times 2560$ resolutions.

| Architecture | Type | Hand | | | Face | | |
|---|---|---|---|---|---|---|---|
| | | #P. (M) | FLOPs (G) | PA-MPJPE (mm)↓ | #P. (M) | FLOPs (G) | 3DRMSE ↓ |
| MobileNetV2 | C | 4.8 | 0.3 | 8.33 | 4.9 | 0.4 | 2.64 |
| ResNet-18 | C | 13.0 | 1.8 | 7.51 | 13.1 | 2.4 | 2.40 |
| **MogaNet-T** | C | 6.5 | 1.1 | **6.82** | 6.6 | 1.5 | **2.36** |
| ResNet-50 | C | 26.9 | 4.1 | 6.85 | 27.0 | 5.4 | 2.48 |
| ResNet-101 | C | 45.9 | 7.9 | 6.44 | 46.0 | 10.3 | 2.47 |
| DeiT-S | T | 23.4 | 4.3 | 7.86 | 23.5 | 5.5 | 2.52 |
| Swin-T | T | 30.2 | 4.6 | 6.97 | 30.3 | 6.1 | 2.45 |
| Swin-S | T | 51.0 | 13.8 | 6.50 | 50.9 | 8.5 | 2.48 |
| ConvNeXt-T | C | 29.9 | 4.5 | 6.18 | 30.0 | 5.8 | 2.34 |
| ConvNeXt-S | C | 51.5 | 8.7 | 6.04 | 51.6 | 11.4 | 2.27 |
| HorNet-T | C | 23.7 | 4.3 | 6.46 | 23.8 | 5.6 | 2.39 |
| **MogaNet-S** | C | 26.6 | 5.0 | **6.08** | 26.7 | 6.5 | **2.24** |

Table A12: **3D human pose estimation** with ExPose on FFHQ and FreiHAND datasets. The face and hand tasks use pre-vertex and pre-joint errors as the metric. The FLOPs of the face and hand tasks are measured with input images at $256^2$ and $224^2$ resolutions.

| Architecture | Type | Crop size | #P. (M) | FLOPs (G) | AP (%) | $AP^{50}$ (%) | $AP^{75}$ (%) | AR (%) |
|---|---|---|---|---|---|---|---|---|
| MobileNetV2 | C | $256 \times 192$ | 10 | 1.6 | 64.6 | 87.4 | 72.3 | 70.7 |
| ShuffleNetV2 2× | C | $256 \times 192$ | 8 | 1.4 | 59.9 | 85.4 | 66.3 | 66.4 |
| **MogaNet-XT** | C | $256 \times 192$ | 6 | 1.8 | **72.1** | **89.7** | **80.1** | **77.7** |
| RSN-18 | C | $256 \times 192$ | 9 | 2.3 | 70.4 | 88.7 | 77.9 | 77.1 |
| **MogaNet-T** | C | $256 \times 192$ | 8 | 2.2 | **73.2** | **90.1** | **81.0** | **78.8** |
| ResNet-50 | C | $256 \times 192$ | 34 | 5.5 | 72.1 | 89.9 | 80.2 | 77.6 |
| HRNet-W32 | C | $256 \times 192$ | 29 | 7.1 | 74.4 | 90.5 | 81.9 | 78.9 |
| Swin-T | T | $256 \times 192$ | 33 | 6.1 | 72.4 | 90.1 | 80.6 | 78.2 |
| PVT-S | T | $256 \times 192$ | 28 | 4.1 | 71.4 | 89.6 | 79.4 | 77.3 |
| PVTV2-B2 | T | $256 \times 192$ | 29 | 4.3 | 73.7 | 90.5 | 81.2 | 79.1 |
| Uniformer-S | H | $256 \times 192$ | 25 | 4.7 | 74.0 | 90.3 | 82.2 | 79.5 |
| ConvNeXt-T | C | $256 \times 192$ | 33 | 5.5 | 73.2 | 90.0 | 80.9 | 78.8 |
| **MogaNet-S** | C | $256 \times 192$ | 29 | 6.0 | **74.9** | **90.7** | **82.8** | **80.1** |
| ResNet-101 | C | $256 \times 192$ | 53 | 12.4 | 71.4 | 89.3 | 79.3 | 77.1 |
| ResNet-152 | C | $256 \times 192$ | 69 | 15.7 | 72.0 | 89.3 | 79.8 | 77.8 |
| HRNet-W48 | C | $256 \times 192$ | 64 | 14.6 | 75.1 | 90.6 | 82.2 | 80.4 |
| Swin-B | T | $256 \times 192$ | 93 | 18.6 | 72.9 | 89.9 | 80.8 | 78.6 |
| Swin-L | T | $256 \times 192$ | 203 | 40.3 | 74.3 | 90.6 | 82.1 | 79.8 |
| Uniformer-B | H | $256 \times 192$ | 54 | 9.2 | 75.0 | 90.6 | 83.0 | 80.4 |
| ConvNeXt-S | C | $256 \times 192$ | 55 | 9.7 | 73.7 | 90.3 | 81.9 | 79.3 |
| ConvNeXt-B | C | $256 \times 192$ | 94 | 16.4 | 74.0 | 90.7 | 82.1 | 79.5 |
| **MogaNet-B** | C | $256 \times 192$ | 47 | 10.9 | **75.3** | **90.9** | **83.3** | **80.7** |
| MobileNetV2 | C | $384 \times 288$ | 10 | 3.6 | 67.3 | 87.9 | 74.3 | 72.9 |
| ShuffleNetV2 2× | C | $384 \times 288$ | 8 | 3.1 | 63.6 | 86.5 | 70.5 | 69.7 |
| **MogaNet-XT** | C | $384 \times 288$ | 6 | 4.2 | **74.7** | **90.1** | **81.3** | **79.9** |
| RSN-18 | C | $384 \times 288$ | 9 | 5.1 | 72.1 | 89.5 | 79.8 | 78.6 |
| **MogaNet-T** | C | $384 \times 288$ | 8 | 4.9 | **75.7** | **90.6** | **82.6** | **80.9** |
| HRNet-W32 | C | $384 \times 288$ | 29 | 16.0 | 75.8 | 90.6 | 82.7 | 81.0 |
| Uniformer-S | H | $384 \times 288$ | 25 | 11.1 | 75.9 | 90.6 | 83.4 | 81.4 |
| ConvNeXt-T | C | $384 \times 288$ | 33 | 33.1 | 75.3 | 90.4 | 82.1 | 80.5 |
| **MogaNet-S** | C | $384 \times 288$ | 29 | 13.5 | **76.4** | **91.0** | **83.3** | **81.4** |
| ResNet-152 | C | $384 \times 288$ | 69 | 35.6 | 74.3 | 89.6 | 81.1 | 79.7 |
| HRNet-W48 | C | $384 \times 288$ | 64 | 32.9 | 76.3 | 90.8 | 82.0 | 81.2 |
| Swin-B | T | $384 \times 288$ | 93 | 39.2 | 74.9 | 90.5 | 81.8 | 80.3 |
| Swin-L | T | $384 \times 288$ | 203 | 86.9 | 76.3 | 91.2 | 83.0 | 814 |
| HRFormer-B | T | $384 \times 288$ | 54 | 30.7 | 77.2 | 91.0 | 83.6 | 82.0 |
| ConvNeXt-S | C | $384 \times 288$ | 55 | 21.8 | 75.8 | 90.7 | 83.1 | 81.0 |
| ConvNeXt-B | C | $384 \times 288$ | 94 | 36.6 | 75.9 | 90.6 | 83.1 | 81.1 |
| Uniformer-B | C | $384 \times 288$ | 54 | 14.8 | 76.7 | 90.8 | 84.0 | 81.4 |
| **MogaNet-B** | C | $384 \times 288$ | 47 | 24.4 | **77.3** | **91.4** | **84.0** | **82.2** |

Table A13: **2D human pose estimation** with Top-Down SimpleBaseline on COCO *val2017*. The FLOPs are measured at $256 \times 192$ or $384 \times 288$ resolutions.

## D.5 3D HUMAN POSE ESTIMATION RESULTS

In addition to Sec. 5.2, we evaluate popular ConvNets and MogaNet for 3D human pose estimation tasks based on ExPose (Choutas et al., 2020). As shown in Table A12, MogaNet achieves lower regression errors with efficient usage of parameters and computational overheads. Compared to lightweight architectures, MogaNet-T achieves 6.82 MPJPE and 2.36 3DRMSE on hand and face reconstruction tasks, improving ResNet-18 and MobileNetV2 1× by 1.29/0.04 and 1.51/0.28. Compared to models around 25∼50M parameters, MogaNet-S surpasses ResNet-101 and ConvNeXt-T, achieving competitive results as ConvNeXt-S with relatively smaller parameters and FLOPs (*e.g.,* 27M/6.5G *vs* 52M/11.4G on FFHP). Notice that some backbones with more parameters produce worse results than their lightweight variants on the face estimation tasks (*e.g.,* ResNet-50 and Swin-S), while MogaNet-S still yields the better performance of 2.24 3DRMSE.

## D.6 VIDEO PREDICTION RESULTS ON MOVING MNIST

In addition to Sec. 5.2, We verify video prediction performances of various architectures by replacing the hidden translator in SimVP with the architecture blocks. All models use the same number of network blocks and have similar parameters and FLOPs. As shown in Table A14, Compared to Transformer-based and Metaformer-based architectures, pure ConvNets usually achieve lower prediction errors. When training 200 epochs, it is worth noticing that using MogaNet blocks in SimVP significantly improves the SimVP baseline by 6.58/13.86 MSE/MAE and outperforms ConvNeXt and HorNet by 1.37 and 4.07 MSE. MogaNet also holds the best performances in the extended 2000-epoch training setting.

| Architecture | #P. (M) | FLOPs (G) | FPS (s) | 200 epochs | | | 2000 epochs | | |
|---|---|---|---|---|---|---|---|---|---|
| | | | | MSE↓ | MAE↓ | SSIM↑ | MSE↓ | MAE↓ | SSIM↑ |
| ViT | 46.1 | 16.9 | 290 | 35.15 | 95.87 | 0.9139 | 19.74 | 61.65 | 0.9539 |
| Swin | 46.1 | 16.4 | 294 | 29.70 | 84.05 | 0.9331 | 19.11 | 59.84 | 0.9584 |
| Uniformer | 44.8 | 16.5 | 296 | 30.38 | 85.87 | 0.9308 | 18.01 | 57.52 | 0.9609 |
| MLP-Mixer | 38.2 | 14.7 | 334 | 29.52 | 83.36 | 0.9338 | 18.85 | 59.86 | 0.9589 |
| ConvMixer | 3.9 | 5.5 | 658 | 32.09 | 88.93 | 0.9259 | 22.30 | 67.37 | 0.9507 |
| Poolformer | 37.1 | 14.1 | 341 | 31.79 | 88.48 | 0.9271 | 20.96 | 64.31 | 0.9539 |
| SimVP | 58.0 | 19.4 | 209 | 32.15 | 89.05 | 0.9268 | 21.15 | 64.15 | 0.9536 |
| ConvNeXt | 37.3 | 14.1 | 344 | 26.94 | 77.23 | 0.9397 | 17.58 | 55.76 | 0.9617 |
| VAN | 44.5 | 16.0 | 288 | 26.10 | 76.11 | 0.9417 | 16.21 | 53.57 | 0.9646 |
| HorNet | 45.7 | 16.3 | 287 | 29.64 | 83.26 | 0.9331 | 17.40 | 55.70 | 0.9624 |
| **MogaNet** | 46.8 | 16.5 | 255 | **25.57** | **75.19** | **0.9429** | **15.67** | **51.84** | **0.9661** |

Table A14: **Video prediction** with SimVP on Moving MNIST. The FLOPs and FPS are measured at the input tensor of $10 \times 1 \times 64 \times 64$ on an NVIDIA Tesla V100 GPU.

## E EXTENSIVE RELATED WORK

**Convolutional Neural Networks** ConvNets (LeCun et al., 1998; Krizhevsky et al., 2012; He et al., 2016) have dominated a wide range of computer vision (CV) tasks for decades. VGG (Simonyan & Zisserman, 2014) proposes a modular network design strategy, stacking the same type of blocks repeatedly, which simplifies both the design workflow and transfer learning for downstream tasks. ResNet (He et al., 2016) introduces identity skip connections and bottleneck modules that alleviate training difficulties (*e.g.,* vanishing gradient). With the desired properties, ResNet and its variants (Zagoruyko & Komodakis, 2016; Xie et al., 2017; Hu et al., 2018; Zhang et al., 2022a) have become the most widely adopted ConvNet architectures in numerous CV applications. For practical usage, efficient models (Ma et al., 2018; Howard et al., 2017; Sandler et al., 2018; Howard et al., 2019; Tan & Le, 2019; Radosavovic et al., 2020) are designed for a complexity-accuracy trade-off and hardware devices. Since the limited reception fields, spatial and temporal convolutions struggle to capture global dependency (Luo et al., 2016). Various spatial-wise or channel-wise attention strategies (Dai et al., 2017; Hu et al., 2018; Wang et al., 2018; Woo et al., 2018; Cao et al., 2019) are introduced. Recently, taking the merits of Transformer-like macro design (Dosovitskiy et al., 2021), modern ConvNets (Trockman & Kolter, 2022; Ding et al., 2022b; Liu et al., 2023; Rao et al., 2022; Kirchmeyer & Deng, 2023) show thrilling performance with large depth-wise convolutions (Han et al., 2021b) for global contextual features. Among them, VAN (Guo et al., 2023), FocalNet (Yang et al., 2022), HorNet (Rao et al., 2022), and Conv2Former (Hou et al., 2022) exploit multi-scale convolutional kernels with gating operations. However, these methods fail to ensure the networks learn the inherently overlooked features (Deng et al., 2022) and achieve ideal contextual aggregation. Unlike the previous works, we first design three groups of multi-order depth-wise convolutions in parallel followed by a double-branch activated gating operation, and then propose a channel aggregation module to enforce the network to learn informative features of various interaction scales.

**Vision Transformers** Transformer (Vaswani et al., 2017) with self-attention mechanism has become the mainstream choice in natural language processing (NLP) community (Devlin et al., 2018; Brown et al., 2020). Considering that global information is also essential for CV tasks, Vision Transformer (ViT) (Dosovitskiy et al., 2021) is proposed and has achieved promising results on ImageNet (Deng et al., 2009). In particular, ViT splits raw images into non-overlapping fixed-

| Architecture | Date | Type | Param. (M) | 100-epoch | | | 300-epoch | | |
|---|---|---|---|---|---|---|---|---|---|
| | | | | Train | Test | Acc (%) | Train | Test | Acc (%) |
| ResNet-18 (He et al., 2016) | CVPR'2016 | C | 12 | $160^2$ | $224^2$ | 68.2 | $224^2$ | $224^2$ | 70.6 |
| ResNet-34 (He et al., 2016) | CVPR'2016 | C | 22 | $160^2$ | $224^2$ | 73.0 | $224^2$ | $224^2$ | 75.5 |
| ResNet-50 (He et al., 2016) | CVPR'2016 | C | 26 | $160^2$ | $224^2$ | 78.1 | $224^2$ | $224^2$ | 79.8 |
| ResNet-101 (He et al., 2016) | CVPR'2016 | C | 45 | $160^2$ | $224^2$ | 79.9 | $224^2$ | $224^2$ | 81.3 |
| ResNet-152 (He et al., 2016) | CVPR'2016 | C | 60 | $160^2$ | $224^2$ | 80.7 | $224^2$ | $224^2$ | 82.0 |
| ResNet-200 (He et al., 2016) | CVPR'2016 | C | 65 | $160^2$ | $224^2$ | 80.9 | $224^2$ | $224^2$ | 82.1 |
| ResNeXt-50 (Xie et al., 2017) | CVPR'2017 | C | 25 | $160^2$ | $224^2$ | 79.2 | $224^2$ | $224^2$ | 80.4 |
| SE-ResNet-50 (Hu et al., 2018) | CVPR'2018 | C | 28 | $160^2$ | $224^2$ | 77.0 | $224^2$ | $224^2$ | 80.1 |
| EfficientNet-B0 (Tan & Le, 2019) | ICML'2019 | C | 5 | $160^2$ | $224^2$ | 73.0 | $224^2$ | $224^2$ | 77.1 |
| EfficientNet-B1 (Tan & Le, 2019) | ICML'2019 | C | 8 | $160^2$ | $224^2$ | 74.9 | $240^2$ | $240^2$ | 79.4 |
| EfficientNet-B2 (Tan & Le, 2019) | ICML'2019 | C | 9 | $192^2$ | $256^2$ | 77.5 | $260^2$ | $260^2$ | 80.1 |
| EfficientNet-B3 (Tan & Le, 2019) | ICML'2019 | C | 12 | $224^2$ | $288^2$ | 79.2 | $300^2$ | $300^2$ | 81.4 |
| EfficientNet-B4 (Tan & Le, 2019) | ICML'2019 | C | 19 | $320^2$ | $380^2$ | 81.2 | $380^2$ | $380^2$ | 82.4 |
| RegNetY-800MF (Radosavovic et al., 2020) | CVPR'2020 | C | 6 | $160^2$ | $224^2$ | 73.8 | $224^2$ | $224^2$ | 76.3 |
| RegNetY-4GF (Radosavovic et al., 2020) | CVPR'2020 | C | 21 | $160^2$ | $224^2$ | 79.0 | $224^2$ | $224^2$ | 79.4 |
| RegNetY-8GF (Radosavovic et al., 2020) | CVPR'2020 | C | 39 | $160^2$ | $224^2$ | 81.1 | $224^2$ | $224^2$ | 79.9 |
| RegNetY-16GF (Radosavovic et al., 2020) | CVPR'2020 | C | 84 | $160^2$ | $224^2$ | 81.7 | $224^2$ | $224^2$ | 80.4 |
| EfficientNetV2-rw-S (Tan & Le, 2021) | ICML'2021 | C | 24 | $224^2$ | $288^2$ | 80.9 | $288^2$ | $384^2$ | 82.9 |
| EfficientNetV2-rw-M (Tan & Le, 2021) | ICML'2021 | C | 53 | $256^2$ | $384^2$ | 82.3 | $320^2$ | $384^2$ | 81.9 |
| ViT-T (Dosovitskiy et al., 2021) | ICLR'2021 | T | 6 | $160^2$ | $224^2$ | 66.7 | $224^2$ | $224^2$ | 72.2 |
| ViT-S (Dosovitskiy et al., 2021) | ICLR'2021 | T | 22 | $160^2$ | $224^2$ | 73.8 | $224^2$ | $224^2$ | 79.8 |
| ViT-B (Dosovitskiy et al., 2021) | ICLR'2021 | T | 86 | $160^2$ | $224^2$ | 76.0 | $224^2$ | $224^2$ | 81.8 |
| PVT-T (Wang et al., 2021b) | ICCV'2021 | T | 13 | $160^2$ | $224^2$ | 71.5 | $224^2$ | $224^2$ | 75.1 |
| PVT-S (Wang et al., 2021b) | ICCV'2021 | T | 25 | $160^2$ | $224^2$ | 72.1 | $224^2$ | $224^2$ | 79.8 |
| Swin-T (Liu et al., 2021) | ICCV'2021 | T | 28 | $160^2$ | $224^2$ | 77.7 | $224^2$ | $224^2$ | 81.3 |
| Swin-S (Liu et al., 2021) | ICCV'2021 | T | 50 | $160^2$ | $224^2$ | 80.2 | $224^2$ | $224^2$ | 83.0 |
| Swin-S (Liu et al., 2021) | ICCV'2021 | T | 50 | $160^2$ | $224^2$ | 80.5 | $224^2$ | $224^2$ | 83.5 |
| LITV2-T (Pan et al., 2022a) | NIPS'2022 | T | 28 | $160^2$ | $224^2$ | 79.7 | $224^2$ | $224^2$ | 82.0 |
| LITV2-M (Pan et al., 2022a) | NIPS'2022 | T | 49 | $160^2$ | $224^2$ | 80.5 | $224^2$ | $224^2$ | 83.3 |
| LITV2-B (Pan et al., 2022a) | NIPS'2022 | T | 87 | $160^2$ | $224^2$ | 81.3 | $224^2$ | $224^2$ | 83.6 |
| ConvMixer-768-d32 (Trockman & Kolter, 2022) | arXiv'2022 | T | 21 | $160^2$ | $224^2$ | 77.6 | $224^2$ | $224^2$ | 80.2 |
| PoolFormer-S12 (Yu et al., 2022) | CVPR'2022 | T | 12 | $160^2$ | $224^2$ | 69.3 | $224^2$ | $224^2$ | 77.2 |
| PoolFormer-S24 (Yu et al., 2022) | CVPR'2022 | T | 21 | $160^2$ | $224^2$ | 74.1 | $224^2$ | $224^2$ | 80.3 |
| PoolFormer-S36 (Yu et al., 2022) | CVPR'2022 | T | 31 | $160^2$ | $224^2$ | 74.6 | $224^2$ | $224^2$ | 81.4 |
| PoolFormer-M36 (Yu et al., 2022) | CVPR'2022 | T | 56 | $160^2$ | $224^2$ | 80.7 | $224^2$ | $224^2$ | 82.1 |
| PoolFormer-M48 (Yu et al., 2022) | CVPR'2022 | T | 73 | $160^2$ | $224^2$ | 81.2 | $224^2$ | $224^2$ | 82.5 |
| ConvNeXt-T (Liu et al., 2022b) | CVPR'2022 | C | 29 | $160^2$ | $224^2$ | 78.8 | $224^2$ | $224^2$ | 82.1 |
| ConvNeXt-S (Liu et al., 2022b) | CVPR'2022 | C | 50 | $160^2$ | $224^2$ | 81.7 | $224^2$ | $224^2$ | 83.1 |
| ConvNeXt-B (Liu et al., 2022b) | CVPR'2022 | C | 89 | $160^2$ | $224^2$ | 82.1 | $224^2$ | $224^2$ | 83.8 |
| ConvNeXt-L (Liu et al., 2022b) | CVPR'2022 | C | 189 | $160^2$ | $224^2$ | 82.8 | $224^2$ | $224^2$ | 84.3 |
| ConvNeXt-XL (Liu et al., 2022b) | CVPR'2022 | C | 350 | $160^2$ | $224^2$ | 82.9 | $224^2$ | $224^2$ | 84.5 |
| HorNet-T$_{7\times7}$ (Rao et al., 2022) | NIPS'2022 | C | 22 | $160^2$ | $224^2$ | 80.1 | $224^2$ | $224^2$ | 82.8 |
| HorNet-S$_{7\times7}$ (Rao et al., 2022) | NIPS'2022 | C | 50 | $160^2$ | $224^2$ | 81.2 | $224^2$ | $224^2$ | 84.0 |
| VAN-B0 (Guo et al., 2023) | CVMJ'2023 | C | 4 | $160^2$ | $224^2$ | 72.6 | $224^2$ | $224^2$ | 75.8 |
| VAN-B2 (Guo et al., 2023) | CVMJ'2023 | C | 27 | $160^2$ | $224^2$ | 81.0 | $224^2$ | $224^2$ | 82.8 |
| VAN-B3 (Guo et al., 2023) | CVMJ'2023 | C | 45 | $160^2$ | $224^2$ | 81.9 | $224^2$ | $224^2$ | 83.9 |
| **MogaNet-XT** | Ours | C | 3 | $160^2$ | $224^2$ | 72.8 | $224^2$ | $224^2$ | 76.5 |
| **MogaNet-T** | Ours | C | 5 | $160^2$ | $224^2$ | 75.4 | $224^2$ | $224^2$ | 79.0 |
| **MogaNet-S** | Ours | C | 25 | $160^2$ | $224^2$ | 81.1 | $224^2$ | $224^2$ | 83.4 |
| **MogaNet-B** | Ours | C | 44 | $160^2$ | $224^2$ | 82.2 | $224^2$ | $224^2$ | 84.3 |
| **MogaNet-L** | Ours | C | 83 | $160^2$ | $224^2$ | 83.2 | $224^2$ | $224^2$ | 84.7 |

Table A15: ImageNet-1K classification performance of tiny to medium size models (5∼50M) training 100 and 300 epochs. RSB A3 (Wightman et al., 2021) setting is used for 100-epoch training of all methods. As for 300-epoch results, the RSB A2 (Wightman et al., 2021) setting is used for ResNet, ResNeXt, SE-ResNet, EfficientNet, and EfficientNetV2 as reproduced in timm (Wightman et al., 2021), while other methods adopt settings in their original paper.

size patches as visual tokens to capture long-range feature interactions among these tokens by self-attention. By introducing regional inductive bias, ViT and its variants have been extended to various vision tasks Carion et al. (2020); Zhu et al. (2021); Chen et al. (2021); Parmar et al. (2018); Jiang et al. (2021a); Arnab et al. (2021). Equipped with advanced training strategies (Touvron et al., 2021a; 2022) or extra knowledge (Jiang et al., 2021b; Lin et al., 2022; Wu et al., 2022c), pure ViTs can achieve competitive performance as ConvNets in CV tasks. In the literature of Yu et al. (2022), the MetaFormer architecture substantially influenced the design of vision backbones, and all Transformer-like models (Touvron et al., 2021a; Trockman & Kolter, 2022; Wang et al., 2022a) are

classified by how they treat the token-mixing approaches, such as relative position encoding (Wu et al., 2021b), local window shifting (Liu et al., 2021) and MLP layer (Tolstikhin et al., 2021), *etc.* Beyond the aspect of macro design, Touvron et al. (2021b); Yuan et al. (2021a) introduced knowledge distillation and progressive tokenization to boost training data efficiency. Compared to ConvNets banking on the inherent inductive biases (*e.g.,* locality and translation equivariance), the pure ViTs are more over-parameterized and rely on large-scale pre-training (Dosovitskiy et al., 2021; Li et al., 2023b) by contrastive learning (He et al., 2020; Zang et al., 2022; Li et al., 2023c) or masked image modeling (Bao et al., 2022; He et al., 2022; Li et al., 2023a; Woo et al., 2023) to a great extent. Targeting this problem, one branch of researchers proposes lightweight ViTs (Xiao et al., 2021; Mehta & Rastegari, 2022; Li et al., 2022c; Chen et al., 2023) with more efficient self-attentions variants (Wang et al., 2021a). Meanwhile, the incorporation of self-attention and convolution as a hybrid backbone has been vigorously studied (Guo et al., 2022; Wu et al., 2021a; Dai et al., 2021; d'Ascoli et al., 2021; Li et al., 2022a; Pan et al., 2022b; Si et al., 2022) for imparting regional priors to ViTs.

