# OpenReview forum: "MogaNet: Multi-order Gated Aggregation Network"
_ICLR.cc/2024/Conference — ICLR 2024 poster_

### Official Review · Reviewer_XNzj · 2023-10-18

**Soundness:** 2 fair
**Presentation:** 1 poor
**Contribution:** 3 good
**Rating:** 6
**Confidence:** 5

**Summary:**

The authors introduce a new family of pure CNN architectures, dubbed MogaNet, by analyzing the effects of different-order interactions in deep neural networks. They assert that MogaNet with two new modules for spatial and channel mixing improves middle-order interaction while suppressing extreme-order interaction, yielding promising results for visual recognition. Comparisons on image classification and downstream tasks show the effectiveness.

**Strengths:**

- The impact of multi-order interaction in neural networks provided by previous work is a fascinating starting point for network architecture design.
- Both quantitative and qualitative experiments demonstrate adequate research work.
- Specific training parameters are provided, which are rather important for reproduction by the community.

**Weaknesses:**

The core idea of this work is hard to follow. Why the proposed network design is "an appropriate composition of convolutional locality priors and global context aggregations" is not entirely evident. In addition to the quantitative depiction in Fig. 3, qualitative analysis of the reasons why earlier network designs failed at this point would be extremely helpful in explaining this central argument. Please refer to the QUESTIONS section for details.

**Questions:**

Primary:
- The perspective on multi-order interactions that serves as the foundation and impetus for this work is presented in [r1]. Analysis of the factors  why existing networks results in an inappropriate composition of local and global context is still lacking.
- It is thought that the self-attention mechanism is more adaptable than typical gate attention. Why does ViT fail to learn middle-order interactions when self-attention is present?
- For the purpose of capturing low-, middle-, and high-order interactions, Moga Block uses three parallel branches. To confirm that the model concentrates more on the middle-order as anticipated, it would be preferable to give the gate values (or attention map).
- The substration $Y-GAP(Y)$, which "forces the network against its implicitly preferred extreme-order interactions," is the main function of FD. Why does the CA block not include this operation?
- Wrong markup in Fig.5, i.e., two $1:0:0$.

Others:
- Will MogaNet be less robust because low-order interation "represents common and widely shared local patterns with great robustness"?
- It would be better if figures following the order they appear in the main-body text.
- It would be interesting to know how the losses (presented in [r1]) and MogaNet pitted against (or collaborate with) each other, because both supervision signal and structure design matter for deep neural networks.
- The two expressions before and after Equation 1 contradict each other, i.e., "in the same shape" and "downsampled features".

I would be glad to raise my rating if thoughtful responses are presented.

[r1] H. Deng, Q. Ren, H. Zhang, and Q. Zhang, “Discovering and Explaining the Representation Bottleneck of DNNs,” in International Conference on Learning Representations (ICLR), 2022.

---

> ### Author Response · Authors · 2023-11-20
> **Response to Reviewer XNzj (PART 1/2)**
>
> We first express our sincere gratitude for your time and effort in reviewing our submission. We have made related adjustments in our revision and provided a general response. We invite you to go through the general response because they will probably answer most of your questions.
>
> ### **1. Some Hard-to-Follow Statements.**
>
> As stated in the first point above, our work is inspired by an ICLR 2022 work [1], which is the theoretical foundation of this work. Due to page limitations, we cannot put the entire theoretical derivation of multi-order interaction in the paper. The issues and theoretical foundations referred to in this work are not unstandardized concepts but rather issues derived through solid theoretical studies. We agree that we did not fully explore this aspect in our original submission. Thus, we have endeavored to condense and integrate the necessary understanding of it in the revision. Please refer to our general response for the clear illustration and concepts of MogaNet. For definitions and derivations regarding the interaction, please refer to [1].
>
> ### **2. Why Do Previous Models Lack Middle-order Interactions?**
>
> To let you better grasp the theoretical starting point of our paper, we put aside the lengthy theoretical derivation, and based on the existing related research, I would like to make some top-down elaboration on the understanding of the interaction in MogaNet. The study in [1] empirically found that current deep neural networks commonly suffer from a middle-order deficient representation bottleneck.
>
> To the best of our knowledge, we suggest that the interaction in practice is mainly influenced by three factors: **the data distribution, the loss function, and the network architecture design.** [2] found that in computer vision tasks, the ideal interaction curve tends to be the strongest in the low-order, and the middle-order is comparable to the high-order one. This is related to the data distribution property of computer vision tasks. [1] not only provided the theoretical foundation but also presented a loss function to solve this issue. [3] affected the interaction curve by dealing with the data distribution.
>
> As for ViT,  we believe that the main reason for the lack of middle-order, despite the use of adaptive data-dependent feature extraction, is the absence of a targeted module design. Global self-attention fails to promote middle-order interaction, which is easy to recognize by its definition, properties, and the findings of prior works.
>
> There is no research on affecting the interactions by network architecture design. Therefore, combining the conclusions of [1] and our empirical analysis in this work, we argue that there is an absence of middle-order interaction in the current deep models. MogaNet is the first attempt to address this issue from the novel perspective of network architecture design.
>
> ### **3. How Does the Design of MogaNet Affect Interaction?**
>
> Overall, our main idea is to force the network to encode the interactions that it would originally ignore and, at the same time, to design a feature extraction method that covers all three interactions as much as possible based on the current experience in computer vision.
> Specifically, the GAP operation in FD(.) extracts the most coarse-grained but primary feature (encoded by the inherent interaction), and the subtract operation extracts the subsidiary feature (encoded by the originally ignored interaction) that the network may have overlooked and then finally supplements the feature to the network. This is the main idea of FD(.), which is also applied to our channel-wise CA(.) module.
>
> As for the Multi-DW(.), we aim to divide the channels into different groups and perform different feature extraction operations in parallel (to characterize different interactions as much as possible). After merging the channels, the context information is extracted through a gating operation to enhance the originally missing high-order interaction as much as possible in an adaptive data-dependent way. The CA(.) is the same idea as FD(.) but performs in the channel space.
>
> ### Reference
>
> [1] Discovering and Explaining the Representation Bottleneck of DNNs. ICLR 2022.
>
> [2] Architecture-Agnostic Masked Image Modeling--From ViT back to CNN. ICML 2023.
>
> [3] Discovering and Explaining the Representation Bottleneck of Graph Neural Networks from Multi-order Interactions. ArXiv 2022.

---

> ### Author Response · Authors · 2023-11-20
> **Response to Reviewer XNzj (PART 2/2)**
>
> ### **4. Why Does CA(.) Not Include FD(.) Operation?**
>
> We are delighted that you have noticed our FD(.) and CA(.) designs. In fact, the FD(.) and CA(.) are, as you said, built on the same concept and purpose. You may have recognized that they go hand in hand by going through the general response and other responses on novelty. CA(.) is the first channel-wise operation that reduces the channel dimension to 1 and conducts feature complementation to force the network against its implicitly preferred interactions.
>
> ### **5. Wrong Markup in Fig.5.**
>
> We appreciate you pointing out this issue. We have modified this in the revision. Many thanks again.
>
> ### **6. Visualization of Multi-order Interpretation.**
>
> Thank you for your constructive feedback. We have provided a visualization comparison and visualization ablation study in Fig.5, Fig.8, and Fig. A4 in our manuscript. You can recognize the effect of MogaNet through these visualization results. Intuitively, the encoded extremely low- or high-order interactions focus on local textures or extremely discriminative parts, not localizing precise semantic parts. The gating operation effectively eliminates the disturbing contextual noise and makes the model effectively locate the semantic patterns.
>
> ### **7. Will MogaNet Be Less Robust?**
>
> MogaNet would not lose robustness. the texture portrayed by low-order interaction lacks high-level semantic features. Concretely, taking computer vision as an example, the model needs to face real-world complex scenes. Too much low-order interaction would make the model lose its discriminative capacity, reducing its robustness in real-world applications. Thus, MogaNet would not become less robust. Extensive experiments on various downstream tasks have empirically verified this point.
>
> ### **8. How Does the [1] Losses and MogaNet Pitted Against or Collaborated With Each Other?**
>
> Special thanks for your extremely valuable suggestions! We conducted the experiments you designated on CIFAR-100 with 224$\times$224 resolutions and 200 epochs training. The results show that simply combining MogaNet with the loss function in [1] (forcing the network to learn the specific interactions) does not yield performance gains, while Swin-T or ConvNeXt-T have little gains using the low-order or the high-order interaction loss. After analysis, we conclude that the loss function constraint of [1] forcing up a certain interaction might exert a negative effect on MogaNet. There might be two reasons. On the one hand, MogaNet expects to complement the interaction components that the network is originally prone to neglect rather than deliberately elevating a certain part. On the other hand, MogaNet has achieved high performance on CIFAR-100 compared to Swin-T and ConvNeXt, which might be too saturated to optimize by the loss functions. Based on these interesting findings, we plan to conduct more in-depth research based on this interesting result in our future work. We really appreciate your constructive suggestions.
>
> | Loss Function      |   Swin-T  | ConvNeXt-T | MogaNet-S |
> |--------------------|:---------:|:----------:|:---------:|
> | Normal Training    |   78.25   |    82.74   | **85.31** |
> | Low Interaction    | **78.33** |    82.75   |   84.60   |
> | Middle Interaction |   77.97   |    82.46   |   85.28   |
> | High Interaction   |   78.12   |  **82.80** |   84.98   |
>
> ### **9. Contradict Expressions.**
>
> We sincerely appreciate you pointing out this issue. We have modified this in the revision.
>
> In conclusion, we sincerely appreciate all your effort and valuable feedback. If you are satisfied with our response and effort, please consider updating your score. If you need any clarification, please feel free to contact us. We are more than pleased to discuss with you and look forward to hearing back from you!
>
> ### Reference
>
> [1] Discovering and Explaining the Representation Bottleneck of DNNs. ICLR 2022.

---

> ### Comment · Reviewer_XNzj · 2023-11-23
>
> I appreciate that you have addressed the majority of my concerns with the revised manuscript, explanation, and supplemented experimental results.
> - Regarding R1/R2, we could, of course, refer the reader to the pertinent literature; nonetheless, in my opinion, a succinct and accurate analysis is still required for a speedy understanding of the main idea of this work. And it is good to see the revision has taken this further.
> - It's encouraging to see the additional relevant experimental findings about the loss as well. This section (comparisons and related qualitative analysis) might motivate readers in the future.
>
> Based on this, I would like to raise my rating to 6.

---

> ### Author Response · Authors · 2023-11-23
>
> Dear Reviewer XNzj,
>
> Thanks for your timely reply! We sincerely appreciate your thorough review of our revision, additional experiment results, and analysis. We are delighted that you have found our rebuttal addressed the majority of your concerns and raised your rating to 6.
>
> - With your thoughtful and constructive suggestions, this paper has clearly improved and even enlightened us for our future work. Many thanks for your time and effort in helping us make our MogaNet better. Your recognition of the “fascinating starting point”, “important for reproduction”,  & experiments of our research is greatly appreciated.
> - In addition, we have also provided the supplemented experimental results (please refer to “Response to Reviewer Koyy (PART 2/2)” for details if you are interested) on ablation studies of how the internal module design of MogaNet affects the interactions. The results show that the proposed modules and their internal design can effectively improve the expressive middle-order interactions (0.3n~0.5n) as well as the classification performance. We believe these will benefit the community as a kit-start to consider network architecture design from new perspectives.
> - We respectfully believe that this work is particularly  **attuned to the ICLR community**, and we hope that MogaNet **can be recognized by more researchers in the community**. Thus, if there might be any additional opportunities for us to further discuss and improve our manuscript, we would like to discuss this with you and would be happy to provide more information based on your feedback or further questions.
>
> Again, we express our sincere appreciation for your constructive feedback and your time and efforts in reviewing our work. We would eagerly welcome any further guidance at your convenience!
>
> Best regards,
>
> Authors

---

### Official Review · Reviewer_tkKT · 2023-10-27

**Soundness:** 2 fair
**Presentation:** 2 fair
**Contribution:** 2 fair
**Rating:** 6
**Confidence:** 5

**Summary:**

This paper introduces a multi-order gated aggregation network, aiming to encode expressive interactions into ConvNets and increase the representation ability of ConvNets. Particularly, a multi-order spatial gated aggregation module is introduced to encode multi-order interactions of features, while a multi-order channel reallocation module is introduced to reduce information redundancy, which can enhance middle-order game-theoretic interactions. The experiments are conducted on several vision tasks, including image classification, object detection, instance and semantic segmentation, 2D and 3D pose estimation, and video prediction.

**Strengths:**

+: The experiments are conducted on several vision tasks, and the results show the proposed networks are competitive to existing popular architectures. In my opinion, extensive experiments are main strength of this work.

+: The overall architectures of this work are clearly descried, and seems to be easy implement.

**Weaknesses:**

-: The analysis on multi-order game-theoretic interaction encourage to propose the multi-order gated aggregation network. However, in my opinion, relationship between Sec. 3 (i.e., analysis) and Sec. 4 (implementation) seems a bit loose. Specifically, I have a bit doubt on why fine-grained local texture (low-order) and complex global shape (middle-order) can be instantiated by Conv1×1(·) and GAP(·) respectively. And why three different DWConv layers with dilation ratios can capture low, middle, and high-order interactions? What are close relationship between multi-order game-theoretic interaction and multi-order channel reallocation module? Therefore, could the authors give more detailed and rigorous correspondings between analysis in Sec. 3 and module design in Sec. 4?

-: From the architecture perspective, the proposed MogaNet exploit multiple depth-wise convolutions for token mixing and channel attention-based FFN for channel mixing. The idea on multiple depth-wise convolutions for token mixing was used in RepLKNet, while channel attention-based FFN was explored in LocalViT and DHVT. So I have a bit doubt on technological novelty of core ideas on designing overall architectures of MogaNet. I suggest the authors can give more detailed analysis on novelty of MogaNet from the architecture perspective.

[RepLKNet]: Scaling up your kernels to 31x31: Revisiting large kernel design in CNNs. In CVPR, 2022

[LocalViT]: LocalViT: Bringing Locality to Vision Transformers. arXiv, 2021.

[DHVT]: Bridging the Gap Between Vision Transformers and Convolutional Neural Networks on Small Datasets. NeurIPS, 2022.

**Questions:**

Additional comments:

-: The compared results show that performance gains of the proposed MogaNet over existing popular architectures is not significant. So could the authors show more advantages of the  MogaNet?

-: Besides parameters and FLOPs, latency is more important for practice applications. So could the authors show some results on the latency (e.g., training time and inference time) of the MogaNet?

-: There exist much more efficient and effective channel attention, besides SE. Therefore, the authors would better compare more efficient channel attention methods to verify the effectiveness of CA module.

---

> ### Author Response · Authors · 2023-11-20
> **Response to Reviewer tkKT (PART 1/2)**
>
> We sincerely appreciate all your valuable feedback, and we are delighted about your comprehensive assessment of our paper. We have made numerous adjustments to our submission and provide a comprehensive general response. We invite you to review our revision and general response because they will probably answer most of your questions. All revised information in our revision is highlighted for your convenience.
>
> ### **1. Presentation Clarity for Sec.3.**
>
> You have expressed concerns about the presentation of Sec.3 in our paper. We acknowledge that there may have been sections in our paper that were not as clear as they could have been. Your feedback heightened our awareness of the importance of clear communication in research. During the revision process, we realized that the lack of logical clarity seriously undermined the expression of what we were contributing. Therefore, we have revised the paper based on your valuable suggestions. Please refer to the revised content **in magenta color** for more details.
>
> ### **2. Technological Novelty.**
>
> Specifically, MogaNet's network architecture can be categorized into FD(.), Multi-DW(.) and CA(.). The first two together form the Spatial Moga(.) block. You have mentioned the following counterparts: VAN [1], FocalNet [2], LocalViT [3], Conv2Former [4], and PVT-V2 [5]. First, please forgive me for reiterating that MogaNet is from a completely unique starting point compared to these works, and we are not solving the same problem. Second, just at the level of network architecture design, we are also quite apart, and MogaNet is still novel.
>
> **(a) For the Spatial Operation:** the VAN [1] uses large DW convolution kernels in series to perform convolution operations. FocalNet [2] utilizes DW convolution kernels to perform convolution and gating operations in series for hierarchical feature maps. Conv2Former [4] applies a so-called “convolutional modulation” operation for feature extraction. Different from all of them, MogaNet first uses a DWConv, then performs channel-wise splitting operations, and performs different operations (different sizes of DWConvs and identity connections) for different groups of channels **in parallel**, followed by a double-branch activated gating operation. This is a completely different design from the aforementioned works. More importantly, we also have different starting points. We have also performed extensive ablation studies and empirical analysis, and our results show that our design achieves exceptional performance gains while being closely aligned with our aforementioned main purpose.
>
> **(b) For the Channel Operation:** Firstly, starting from LocalViT [3] and PVT-V2 [5], most of the ViT or modern ConvNet models utilize “CFFN” (Convolutional FFN, i.e., 1$\times$1 Conv → 3$\times$3 DWConv → 1$\times$1 Conv) as the CMixer. In other words, this is a consensus now for the community and should not be taken as a reason for the “limited novelty” of MogaNet. Secondly, MogaNet's CA(.) is designed to address the aforementioned macro-objectives, and its design concept is the same as that of FD(.). Meanwhile, to the best of our knowledge, CA(.) is the first channel-wise operation that reduces the channel dimension to 1 and conducts feature complementation to **force the network to learn the inherently overlooked features**. Combined with the outstanding performance of CA(.) and the provided empirical analysis, we believe that it is certainly evident to verify the novelty of CA(.).
>
> ### Reference
>
> [1] Visual Attention Network, CVMJ 2023.
>
> [2] Focal Modulation Networks, NeurIPS 2022.
>
> [3] LocalViT: Bringing Locality to Vision Transformers, ArXiv 2021.
>
> [4] A Simple Transformer-Style ConvNet for Visual Recognition, ArXiv 2022.
>
> [5] PVT v2: Improved Baselines with Pyramid Vision Transformer, CVMJ 2022.

---

> ### Author Response · Authors · 2023-11-20
> **Response to Reviewer tkKT (PART 2/2)**
>
> ### **3.  About Performance Gains.**
>
> We fully respect your assessment that the performance gains are not significant. However, we would like to emphasize two points in this regard:
>
> **(a) Consistent gains across a broad range of tasks.**
>
> We have conducted extensive experiments in mainstream image benchmarks and various downstream tasks. The results show that MogaNet achieves the consistent best results. To be clear, many related prior works have not done that many experiments, and they are also accepted by top conferences. In contrast, MogaNet is rather the more empirically testable work with sound experiments.
>
> **(b) Comparison with prior SOTA models.**
>
> Taking the small CNN models as an example, we rank the SOTA models by **chronological order** and count their performance gains on ImageNet-1K. Note that we reproduce RepLKNet-T$^\dagger$ according to the layer setting in SLaK. We observe that, with the exception of the earlier "low-hanging fruit" work, all the SOTAs provide a gain of **0.2~0.7% top-1 accuracy** over the previous work (ConvNeXt as the baseline), and some of them even suffer from performance **degradation**. In contrast, MogaNet-S shows a performance gain of **1.3% accuracy** over ConvNeXt-T, which exceeds the results of prior works accepted by top conferences. Meanwhile, we also add the self-supervised pre-training result of MogaNet (using A2MIM [5]) in comparison to ConvNeXt.V2 [6] (using CMAE [6]), indicating that MogaNet can further benefit from self-supervised learning.
>
> | Architecture           | Date      | \# Param. | Pre-train |   Top-1 (Gain)  |
> |------------------------|:---------:|:---------:|:---------:|:---------------:|
> | ConvNeXt-T             | CVPR'2022 |    29M    |     -     |   82.1 (+0.0)   |
> | RepLKNet-T$^\dagger$      | CVPR'2022 |    29M    |     -     |   82.4 (+0.3)   |
> | HorNet-T$_{7\times 7}$ | NIPS'2022 |    22M    |     -     |   82.8 (+0.7)   |
> | FocalNet-T (LRF)       | NIPS'2022 |    29M    |     -     |   82.3 (+0.2)   |
> | VAN-B2                 | CVMJ'2023 |    27M    |     -     |   82.8 (+0.7)   |
> | SLaK-T                 | ICLR'2023 |    30M    |     -     |   82.5 (+0.4)   |
> | ConvNeXt.V2-T          | CVPR'2023 |    28M    |    CMAE   |   83.0 (+0.9)   |
> | **MogaNet-S**          |    Ours   |    25M    |     -     | **83.4 (+1.3)** |
> | **MogaNet-S**          |    Ours   |    25M    |   A2MIM   | **83.5 (+1.4)** |
>
> Thus, we believe that the gains brought by MogaNet are perfectly adequate and not trivial.
>
> ### **4. Latency Results.**
>
> Due to the limited space, we have not shown the latency results in the main test. However, we provide results for the throughput in Appendix C.3, and MogaNet demonstrates a competitive throughput performance. It proves that MogaNet can take into account the throughput efficiency, in addition to the parameter efficiency and FLOPs. Please view Table A3 in Appendix C.3 for details.
>
> ### **5. CA Counterpart Comparison.**
>
> In the CMixer, compared to popular channel aggregation modules (SE, GLU, SwiGLU [1]) and specially-designed FFN variants (CMT [2] and RepFFN [3]) on ImageNet-1K based on MogaNet-S. CFFN (1$\times$1 Conv → 3$\times$3 DWConv → 1$\times$1 Conv) served as the baseline in PVTv2 [4]. Note that RepFFN utilizes 4 group convolution kernels (gConv) to reduce the parameters and apply a structural re-parameterization to merge the kernels into the FC layer for inference.
>
> | Method           |    Data    | \# Param. | Additional Modules      |   Top-1  |
> |------------------|:----------:|:---------:|---------------------------|:--------:|
> | CFFN (PVTv2)     |  CVMJ'2022 |   24.5M   | -                          |   82.9   |
> | +SE              |  ICCV'2019 |  +0.71M   | GAP, 2xFC, ReLU      |   83.1   |
> | +GLU             | arXiv'2020 |   +0.0M   | Split, Sigmoid             |   82.9   |
> | +SwiGLU (DINOv2) | arXiv'2023 |   +0.0M   | Split, SiLU         |   83.0   |
> | +IRFFN (CMT)     |  CVPR'2022 |  +0.05M   | BN, GELU          |   83.0   |
> | +RepMLP          |  CVPR'2022 |  +1.75M   | 4xgConv (k=1, 3, 5, 7), 4xBN |   83.2   |
> | **+CA**          |    Ours    |  +0.52M   | FC, Scale, GELU           | **83.4** |
>
> Overall, we sincerely appreciate your effort and valuable feedback. If you are satisfied with our response and effort, please consider updating your score. If you need any clarification, please feel free to contact us. We are more than pleased and are looking forward to hearing back from you!
>
> ### Reference
>
> [1] DINOv2: Learning Robust Visual Features without Supervision. ArXiv, 2023.
>
> [2] CMT: Convolutional neural networks meet vision transformers. CVPR, 2022.
>
> [3] RepMLP: Re-parameterizing Convolutions into Fully-connected Layers for Image Recognition. CVPR, 2022.
>
> [4] PVTv2: Improved Baselines with Pyramid Vision Transformer. CVMJ, 2022.
>
> [5] Architecture-Agnostic Masked Image Modeling -- From ViT back to CNN. ICML, 2023.
>
> [6] ConvNeXt V2: Co-designing and Scaling ConvNets with Masked Autoencoders. CVPR, 2023.

---

> ### Author Response · Authors · 2023-11-23
> **Summarized Response to the Unaddressed Concerns**
>
> Dear Reviewer tkKT,
>
> We first greatly appreciate your thorough review and all your valuable feedback. Many thanks for your time and effort in helping us make our MogaNet better. With your constructive suggestions, this work has been clearly improved and even enlightened us for our future work.
>
> Herein, we **sincerely invite you to view our response** and provide us with your comments. If you are satisfied with our response and effort, please consider updating your score. If you need any clarification, please feel free to contact us. We would be happy to provide more information based on your feedback or further questions.
>
> - In our response, we carefully illustrated and answered all your questions with experiments or detailed explanations. As you have requested, we have conducted additional experiments on MogaNet to show the **performance gains** and **more CA counterparts comparisons**. In addition, your concerns about the **technical novelty of MogaNet** are meticulously explained in our response. Please refer to our tables and explanations for details. Meanwhile, we have incorporated all changes into the revised manuscript for your consideration. We hope your concerns have been addressed.
> - Besides, we have also provided the supplemented experimental results (please refer to “Response to Reviewer Koyy (PART 2/2)” for details if you are interested) on ablation studies of how the **internal module design** of MogaNet affects the interactions. The results show that the proposed modules and their internal design can effectively improve the expressive middle-order interactions (0.3n~0.5n) along with the classification performance gains.
>
> We respectfully believe that this work is **attuned to the ICLR community**, and we hope that our work **can be seen by more researchers** in the community. Thus, **your rating is tremendously valuable to us.** If there might be any additional opportunities for us to further improve our manuscript and potentially increase your rating, we would like to discuss this with you and would be happy to provide more information based on your feedback or further questions.
>
> Best regards,
>
> Authors

---

### Official Review · Reviewer_Koyy · 2023-10-27

**Soundness:** 3 good
**Presentation:** 4 excellent
**Contribution:** 3 good
**Rating:** 8
**Confidence:** 4

**Summary:**

This paper proposes MogaNet, a new form of attentional aggregation mechanism across spatial dimension and channel dimension. The main motivation for this new design is to force the network against learning its implicitly preferred extreme-order interactions, and instead to learn the mid-order interactions more easily. The paper presents empirical evidence that with the proposed spatial and channel aggregation modules, the network can score higher in learning the mid-order interactions as well as achieve state-of-the-art results on multiple computer vision tasks and benchmarks.

**Strengths:**

## originality
This paper presents a **novel perspective** that we should design neural networks such that it can efficiently learn **multi-order** interactions, esp. the mid-order ones. Guided by this perspective, this paper proposes a new form of **attention** mechanism (Moga Block) for both spatial and channel aggregation. While the proposed Moga Block is **not** exactly of strong novelty, the lens through which the new design is investigated and measured is very **interesting and novel**.

## quality & clarity
This paper is **excellently presented** and backed up with extensive experiments in both the main paper and the supplementary materials. The writing is precise and concise. The figure and table layout is well thought out.

## significance
While the claim on the benefit of learning multi-order interactions still need to be verified with time, I believe the **strong empirical performance** achieved by the new design is of strong significance already.

**Weaknesses:**

There lacks a **theoretical understanding** on why the proposed Moga Block can help facilitate the learning of more mid-order interactions. There also lacks a **theoretical understanding** on why more mid-order interactions is better for the computer vision tasks. What should the **best curve** for "interaction strength of order" look like? Should it be a horizontal line across all the interaction orders? (If not, why should we automatically believe that more mid-order interactions will be better?)

Figure 7 shows the proposed "Moga(.)" module and "CA(.)" module are helping the model to learn more mid-order interactions. But it would be also very helpful to show how the **internal design** of "Moga(.)" and "CA(.)" modules affect the curve for "Interaction Strength of Order".  For example, why do we choose the "Cl : Cm : Ch = 1:3:4" (section 4.4)? Would different mix move the curve differently? Same question for the design in Figure 4(a) and 4(b), which sub-design is the most effective component in moving the curve?

**Questions:**

see questions raised above

---

> ### Author Response · Authors · 2023-11-20
> **Response to Reviewer Koyy (PART 1/2)**
>
> We first express our sincere gratitude for your time and effort in reviewing our submission. We have made related adjustments to our submission and provided a comprehensive general response to discuss the novelty and contributions of this work. We invite you to go through our revision and general response first.
>
> Upon reflection, we agree that there are opportunities to strengthen the novelty and clarify the contributions of our proposed method. Here are our responses to the key points raised:
>
> ### **1. Theoretical Understanding of Multi-order Interaction for CV.**
>
> Due to the limited length of the paper, we cannot give all the theoretical derivations in the main text. We provide part of the theoretical derivation in the supplementary material. For more details, please refer to the ICLR 2022 work [1] and Appendix B.
>
> As stated in the first point above, our work is inspired by [1], which is the theoretical foundation of our work. Due to page limitations, we cannot comprehensively give the entire theoretical derivation of multi-order interaction in the paper. We have endeavored to condense and integrate the necessary understanding of it in the revision and provide part of the theoretical derivation in the supplementary material (or refer to [1]). Herein, to let you better grasp the theoretical starting point of our paper, we put aside the lengthy theoretical derivation, and based on the existing related research, I would like to make some top-down elaboration on the understanding of the interaction in MogaNet.
>
> Detailed theoretical derivation and empirical analysis are carried out in [1]. The results empirically show that low-order interaction has the representation capacity for local texture, which is robust but deficient in high-level semantic information. High-order interaction shows a strong representation ability for global information, which requires complex global feature interactions, but the portion of images in datasets is not that large. Therefore, high-order interaction has been empirically shown to reduce the model's robustness and is vulnerable to attacks. Middle-order has strong robustness while exhibiting better semantic feature extraction capability than the low-order one.
>
> ### **2. What Should An Ideal Interaction Curve Look Like?**
>
> To the best of our knowledge, the ideal interaction curve should cover all the low, middle, and high levels of interactions and be comparable to all these three levels. However, the situation is different in real-world practice. Because real-world scenarios are complex and changeable, different encoded interactions are adept at dealing with different scenarios. Based on existing research, we suggest that the interaction curve in real practice is mainly influenced by three factors: **the inductive bias of data distribution, the loss function, and the design of network architectures.** [2] found that in computer vision tasks, the ideal interaction curve tends to be the strongest in the low-order, and the middle-order is comparable to the high-order one, which is related to the data distribution property of natural images. But in other data modalities like graphs, the ideal interaction curve should be different, e.g., the middle-order interactions tend to be the highest because of the sub-graph structures. [1] not only first proposed a theoretical explanation but also presented a targeted loss function. [3] affected the interaction curve by dealing with the data distribution. MogaNet is the first attempt to address this issue from the perspective of network architecture design that fits the natural interaction distribution adaptively.
>
> ### Reference
>
> [1] Discovering and Explaining the Representation Bottleneck of DNNs, ICLR 2022.
>
> [2] Discovering and Explaining the Representation Bottleneck of Graph Neural Networks from Multi-order Interactions, Arxiv 2022.
>
> [3] Architecture-Agnostic Masked Image Modeling--From ViT back to CNN, ICML 2023.

---

> ### Author Response · Authors · 2023-11-20
> **Response to Reviewer Koyy (PART 2/2)**
>
> ### **3. How Does the Design of MogaNet Affect the Interaction Curve?**
>
> Overall, our main idea is to force the network to encode the interactions that it would originally ignore and, simultaneously, to design a feature extraction method that covers all three interactions as much as possible based on the current experience in computer vision.
> Specifically, the GAP operation in FD(.) extracts the most coarse-grained but primary feature (encoded by the inherent interaction), and the subtract operation extracts the subsidiary feature (encoded by the originally ignored interaction) that the network may have overlooked and then finally supplements the feature to the network. This is the main idea of FD(.), which is also applied to our channel-wise CA(.) module. They can help the network to learn **the inherently overlooked features encoded by the originally overlooked interactions**. As shown in the following table, we report top-1 accuracy and the multi-order interaction strength for the ablation studies of the proposed modules based on MogaNet-S. The proposed Multi-order DW(.) and FD(.) for SMixer and CA(.) for CMixer can improve the middle-order interactions (0.3n~0.5n) and the classification performance.
>
> | Modules            |  Parts |   Top-1  |  0.0n | 0.05n |  0.1n |  0.3n |  0.5n |  0.7n |  0.9n | 0.95n |  1.0n |
> |--------------------|:------:|:--------:|:-----:|:-----:|:-----:|:-----:|:-----:|:-----:|:-----:|:-----:|:-----:|
> | Baseline           |    -   |   82.2   |  2.98 |  2.11 |  1.19 |  0.60 |  0.44 |  0.42 |  0.60 |  1.02 |  2.03 |
> | +Multi-order DW(.) | SMixer |   +0.5   | -0.35 | -0.22 | +0.15 | +0.10 | +0.12 | +0.11 | +0.09 | -0.13 | -0.20 |
> | +FD(.)             | SMixer |   +0.7   | -0.47 | -0.24 | +0.15 | +0.13 | +0.17 | +0.15 | +0.10 | -0.10 | -0.28 |
> | +CA(.)             | CMixer | **+1.2** | -0.70 | -0.26 | +0.23 | **+0.19** | **+0.22** | **+0.20** | +0.13 | -0.09 | -0.35 |
>
> As for the internal designs in Multi-order DW(.), we aim to divide the channels into different groups and perform different feature extraction operations in parallel (to characterize different interactions as much as possible). After merging the channels, the context information is extracted through a gating operation to enhance the originally missing high-order interaction as much as possible in an adaptive data-dependent way. We also report top-1 accuracy and the distribution of interaction strengths for the channel ablation of Multi-order DW(.) as follows. We find that low-order DWConv (1:0:0) prefers low-order interactions while combining the middle- and high-order DWConv (0:1:1) learns more middle- and high-order interactions. Therefore, we can find an optimal channel combination in DWConv layers that reconciles multi-order interactions, i.e., using $C_{l}: C_{m}: C_{h} = 1:3:4$.
>
> | Modules           | $C_l: C_m: C_h$ |   Top-1  |  0.0n | 0.05n |  0.1n |    0.3n   |    0.5n   |    0.7n   |  0.9n | 0.95n |  1.0n |
> |-------------------|:---------------:|:--------:|:-----:|:-----:|:-----:|:---------:|:---------:|:---------:|:-----:|:-----:|:-----:|
> | Multi-order DW(.) |      1:0:0      |   82.9   |  2.61 |  1.98 |  1.25 |    0.66   |    0.50   |    0.51   |  0.65 |  0.85 |  1.87 |
> | Multi-order DW(.) |      0:1:1      |   83.1   | -0.36 | -0.17 | +0.04 |   +0.10   |   +0.06   |   +0.09   | +0.05 | +0.07 | +0.03 |
> | Multi-order DW(.) |      1:6:9      |   83.2   | -0.25 | -0.15 | +0.10 |   +0.13   |   +0.08   |   +0.12   | +0.08 | +0.10 | -0.09 |
> | Multi-order DW(.) |    **1:3:4**    | **83.4** | -0.33 | -0.13 | +0.17 | **+0.14** | **+0.11** | **+0.11** | +0.07 | +0.12 | -0.20 |
>
> In conclusion, thanks again for your constructive feedback, and we have considered your valuable comments for the revision and future work. If you are satisfied with our response and effort, please consider updating your score. If you need any clarification, please feel free to contact us. We are more than pleased and looking forward to hearing back from you!

---

> > ### Comment · Reviewer_Koyy · 2023-11-22
> > **Thank you for the additional analysis**
> >
> > I appreciate the revised manuscript, particularly the additional experiments and detailed analysis. These additions are valuable resources for future readers and researchers pursuing similar lines of inquiry. This is to confirm I have reviewed your responses and incorporated them into my final recommendation as a reviewer.

---

> ### Author Response · Authors · 2023-11-22
>
> Dear Reviewer Koyy,
>
> Thanks for your timely reply! We are delighted that you have found our rebuttal feedback valuable and maintain your rating of 6. Your recognition of the originality, quality & clarity, and significance of our research is greatly appreciated. We would respectfully inquire if there might be any additional opportunities for us to further improve our manuscript and potentially increase the rating. Once again, thanks for your constructive feedback, and we would eagerly welcome any further guidance at your convenience!
>
> Best regards,
>
> Authors

---

### Official Review · Reviewer_6yMg · 2023-11-04

**Soundness:** 3 good
**Presentation:** 3 good
**Contribution:** 2 fair
**Rating:** 6
**Confidence:** 5

**Summary:**

This paper introduces a new convolutional neural network for varies of computer vision tasks.
Specifically, it is motivated by the theory of  multi-order game-theoretic interaction in deep neural networks.
This paper finds that popular transformer-based and CNN-based networks have limited interactions on the middle-order interactions. So it introduces the MogaNet with multi-order gated aggregation to solve this problem.
MogaNet uses convolutions with different kernel size as well as the gated aggregation, which can adatively conduct the multi-order aggregation.

Experiments are perform on several popular benchmarks, such image classification, semantic segmentation, object detection, instance segmentation, and pose estimation. Results show that MogaNet achieves the SOTA performance on several popular benchmarks.

**Strengths:**

+ The paper is well-written, comprehensively introducing the motivation and method details.

+ The experiments are comprehensive, covering several popular vision tasks as well as varies of network scales.

+ The experimental and visualized analysis is good, helping the reviewer better understand the method.

+ Code has been released, so the reproducibility can be ensured.

**Weaknesses:**

- Despite good experiments and visualizations, I think the novelty is limited.
As described in the introduction, the low-order interactions are modeling the local features, such as edge and texture. The high-order on the other hand models high-level semantic features. So multi-order feature aggreation indicates the multiscale aggregation with low and high level features. This paper implements it via depth-wise convolution with different kernel size and further adds gated operation, introducing multi-order gated aggregation.
However, FocalNet exibits similar behavior, proposing hierachical gated aggregation with locality perception and gated aggregation. So I think the proposed MogaNet has similar motivation and mechanism with FocalNet [1].

- Moreover,
The proposed method is a variant of convolutional modulation, but lacks an in-depth discussion on differences with recent CNNs based on convolutional modulations, such as VAN [2], FocalNet [1], and Conv2Former [3].
Besides, VAN [2] and FocalNet [1] should be added in Figure 3 for a comprehensive analysis on the interaction strength.

- Regarding to the Figure 7 of ablation study, I am confused that the main improvement is not from multi-order convolution or gate, which are claimed as major contributions of this work.
Instead, the main improvement is from the CA, which is embedded in the feed-forward network. Note that other networks do not have CA in their feed-forward network, introducing somewhat unfair comparison.
Therefore, I think the authors should better clarify the mechanism of the CA and claim it as the major contribution, not only emphasizing the proposed multi-order gated aggregation.


[1] Focal Modulation Networks, NeurIPS 2022

[2] Visual Attention Network, CVMJ 2023

[3] A Simple Transformer-Style ConvNet for Visual Recognition, ArXiv 2022

**Questions:**

Refer to the weakness.

The major problem is the limited novelty. Besides, there lacks a comprehensive discussion on convolutional modulations.
The major improvement is from the CA, not the modules claimed as major contributions.

---

> ### Author Response · Authors · 2023-11-20
> **Response to Reviewer 6yMg (PART 1/3)**
>
> We express our gratitude for your valuable review and constructive feedback. We have adjusted our revision and provided a general response to some common issues. We invite you to go through the general response first, as they may have answered some of your confusion, and you will have a new understanding of the novelty and contributions of MogaNet.
>
> Upon reflection, we agree that there are opportunities to strengthen the novelty and clarify the contributions of our proposed method.  Specifically, MogaNet's network architecture can be categorized into FD(.), Multi-DW(.) and CA(.). The first two together form the Spatial Moga(.) block. Here, we will address your concern about the novelty from the following three aspects:
>
> ### **1. On Novelty**
>
> You raise important points regarding the novelty and differentiation of our method from related work. We agree that aggregating multi-scale features is a common theme in modern ConvNets, or the so-called “convolutional modulations”. While related work explores multi-scale aggregation, we believe our MogaNet is quite novel from both its purpose and the design itself.
>
> Our MogaNet work was inspired by an ICLR22 work [1], which revealed for the first time the problem that existing deep neural networks suffer from a representation bottleneck under a hierarchical theoretical analysis (Multi-order Game-theoretic Interaction) and thereby proposed a loss function to solve this problem. This work was recognized by ICLR2022, but no further attempts have been made to solve this problem. Thus, we are inspired to address this issue from the novel perspective of network architecture design (instead of loss function).
>
> Specifically, deep neural networks show a natural proclivity towards multi-order interactions - the most expressive middle-order interactions are not well encoded in the current learning process, and we attempt to address this issue with network architectures that can **force the network to learn to encode more expressive interactions that would have otherwise been inherently overlooked.** To facilitate understanding, from a computer vision standpoint, you can roughly, but not quite accurately, interpret this as forcing the network to capture features that were discriminative but inherently ignored.
>
> To the best of our knowledge, **we are the first to address this issue [1] with a specific network architecture design and are also the first to approach network architecture design from the multi-order interaction [1] perspective.** Therefore, it is clear that we have a completely different starting point compared to other related works. Again, briefly speaking, we are making a new attempt to solve the problem that was first raised by the ICLR 2022 work [1] rather than a merely "borrowed" network design without a top-down guideline. As such, we believe that MogaNet is obviously interesting and novel from an academic research point of view. Moreover, we summarize the performance gains of MogaNet and related ConvNets on ImageNet-1K as follows, indicating the effectiveness and importance of our work in the line of improving ConvNets.
>
> ### Reference
>
> [1] Discovering and Explaining the Representation Bottleneck of DNNs, ICLR 2022.

---

> ### Author Response · Authors · 2023-11-20
> **Response to Reviewer 6yMg (PART 2/3)**
>
> ### **2. On Discussion of Other Convolutional Modulations.**
>
> You have mentioned the following similar works: VAN [1], FocalNet [2], and Conv2Former [3] (an occurrent work as MogaNet in late 2022). While related literature exists, the **intention and starting points** behind our methods are actually **far apart**. Firstly, MogaNet is from a completely unique starting point compared to these works, and we are not solving the same problem. We aim to solve this obstacle within DNNs by architecture design. We actually own completely different motivations and targets, so we don’t discuss them in this work as the available pages are limited. Second, just at the level of network architecture design, we are also quite apart, and MogaNet is still novel and important:
>
> **(a) For the Spatial Operation:** the VAN [1] uses large DW convolution kernels in series to perform convolution operations. FocalNet [2] utilizes DW convolution kernels to perform convolution and gating operations in series for hierarchical feature maps. Conv2Former [3] applies a so-called “convolutional modulation” operation for feature extraction. Different from all of them, MogaNet first uses a DWConv, then performs channel-wise splitting operations, and performs different operations (different sizes of DWConvs and identity connections) for different groups of channels **in parallel**, followed by a double-branch activated gating operation. This is a completely different design from the aforementioned works. More importantly, we also have different starting points. We have also performed extensive ablation studies and empirical analysis, and our results show that our design achieves exceptional performance gains while being closely aligned with our aforementioned main purpose. In addition, MogaNet's performance is far above them.
>
> **(b) For the Channel Operation:** First, starting from LocalViT [4] and PVT-V2 [5], most of the ViT or modern ConvNet models utilize CFFN (Convolutional FFN, i.e., 1$\times$1 Conv → 3$\times$3 DWConv → 1$\times$1 Conv) as the channel feature extractor. In other words, this is a consensus now for the community and should not be taken as a reason for the “limited novelty” of MogaNet. Second, MogaNet's CA(.) is designed to address the aforementioned macro-objectives, and its design concept is the same as that of FD(.), which is **forcing the network to learn to encode more expressive interactions that would have been inherently (originally) overlooked.** Meanwhile, to the best of our knowledge, CA(.) is the first channel-wise operation that reduces the channel dimension to 1 and conducts feature complementation. Combined with the outstanding performance of CA(.) and the provided empirical analysis, we believe that it is certainly evident to demonstrate the novelty of CA(.).
>
> We appreciate your helpful feedback and will add a related description in our revision. You rightly note that we could have better situated our work within the broader context of convolutional modulations. We have revised our Sec.1, Sec.2, and Sec.3 in our revision. As shown in the following table, we conduct interaction analysis experiments, which show that MogaNet not only surpasses the popular modern ConvNets you mentioned but exhibits better interaction performance, where the expressive middle-order interactions (0.3n~0.5n) are effectively enhanced. View Appendix B.1 and Fig. A2 for details.
>
> | Modules                |    Date    |   Top-1  |  0.0n | 0.05n |  0.1n |    0.3n   |    0.5n   |    0.7n   |  0.9n | 0.95n |  1.0n |
> |------------------------|:----------:|:--------:|:-----:|:-----:|:-----:|:---------:|:---------:|:---------:|:-----:|:-----:|:-----:|
> | ConvNeXt-T             |  CVPR'2022 |   82.1   |  2.98 |  2.11 |  1.19 |    0.60   |    0.44   |    0.42   |  0.60 |  1.02 |  2.03 |
> | HorNet-T$_{7\times 7}$ |  NIPS'2022 |   82.8   | +0.62 | -0.16 | +0.02 |   +0.01   |   +0.02   |   +0.02   | -0.02 | -0.06 | -0.55 |
> | FocalNet-T (LRF)       |  NIPS'2022 |   82.3   | +0.73 | -0.28 | +0.06 |   +0.07   |   +0.04   |   +0.07   | +0.06 | -0.07 | -0.47 |
> | VAN-B2                 |  CVMJ'2023 |   82.8   | -0.08 | -0.13 | +0.18 |   +0.14   |   +0.11   |   +0.13   | +0.05 | -0.18 | -0.77 |
> | SLaK                   |  ICLR'2023 |   82.5   | +0.13 | -0.21 | -0.02 |   +0.01   |   +0.04   |   +0.08   | +0.07 | +0.02 | -0.16 |
> | Conv2Former-T          | arXiv'2022 |   83.2   | -0.32 | -0.29 | +0.20 |   +0.16   |   +0.12   |   +0.15   | +0.08 | -0.10 | -0.75 |
> | **MogaNet**            |    Ours    | **83.4** | -0.70 | -0.26 | +0.23 | **+0.19** | **+0.22** | **+0.20** | +0.13 | -0.09 | -0.35 |
>
> ### Reference
>
> [1] Visual Attention Network, CVMJ 2023.
>
> [2] Focal Modulation Networks, NeurIPS 2022.
>
> [3] A Simple Transformer-Style ConvNet for Visual Recognition, ArXiv 2022.
>
> [4] LocalViT: Bringing Locality to Vision Transformers, ArXiv 2021.
>
> [5] PVT v2: Improved Baselines with Pyramid Vision Transformer, CVMJ 2022.

---

> > ### Comment · Reviewer_6yMg · 2023-11-23
> >
> > Thanks a lot for the feedback.
> >
> > It is good to present an experimental analysis on existing convolutional modulation networks. It will be better to add this analysis to final version.
> >
> > While this work has significant difference compare with VAN and the concurrent work Conv2Former, this work shares similar insights with FocalNet, since both of them conduct multi-order aggregation with gated convolutions.
> >
> > Also, it is important to discuss the above works in the related work, all of which are based on convolutional modulations.
> >
> > At last, the largest improvement is not from the major contribution. Authors' new table still seems confusing, as the minor module CA provides 0.5% improvement while each module of major contributions only have 0.2~0.3% improvement.
> >
> > Based on the current status, I would like to keep my rating.

---

> ### Author Response · Authors · 2023-11-20
> **Response to Reviewer 6yMg (PART 3/3)**
>
> ### **3. On the Main Improvement in MogaNet.**
>
> The reviewer makes a fair point. Our ablation study could have better disambiguated the contributions of multi-order aggregation (Moga) vs. CA. To clarify, the spatial Moga(.) module and channel CA(.) module collaborate **as an integration** to achieve our goal, where **the expressive but originally overlooked interactions encoded in DNNs can be forced to enhance.** As shown in the following table, we ablate the proposed modules by adding them one by one to the baseline (similar to Table 1) and calculate the distributions of interaction strengths. Note that 0.3n~0.5n reflects the favorite middle-order interactions. In particular, CA(.) is embedded to allow aggregation to dynamically attend to useful features. CA(.) is designed to address the aforementioned macro-objectives, and its design concept is the same as that of FD(.). As provided in the new ablation study of our revision, the CA(.) can only perform its best coupled with Moga(.). It empirically indicates that the Moga(.) and CA(.) collaborate as a unified design for the same goal but are designed from different feature spaces (spatial-wise and channel-wise, respectively).
>
> | Modules            |  Parts |   Top-1  |  0.0n | 0.05n |  0.1n |  0.3n |  0.5n |  0.7n |  0.9n | 0.95n |  1.0n |
> |--------------------|:------:|:--------:|:-----:|:-----:|:-----:|:-----:|:-----:|:-----:|:-----:|:-----:|:-----:|
> | Baseline           |    -   |   82.2   |  2.98 |  2.11 |  1.19 |  0.60 |  0.44 |  0.42 |  0.60 |  1.02 |  2.03 |
> | +Gating branch     | SMixer |   +0.2   | -0.06 | -0.03 | +0.01 | +0.03 | +0.01 | +0.02 | -0.01 | +0.02 | +0.01 |
> | +Multi-order DW(.) | SMixer |   +0.5   | -0.35 | -0.22 | +0.15 | +0.10 | +0.12 | +0.11 | +0.09 | -0.13 | -0.20 |
> | +FD(.)             | SMixer |   +0.7   | -0.47 | -0.24 | +0.15 | +0.13 | +0.17 | +0.15 | +0.10 | -0.10 | -0.28 |
> | +CA(.)             | CMixer | **+1.2** | -0.70 | -0.26 | +0.23 | +0.19 | +0.22 | +0.20 | +0.13 | -0.09 | -0.35 |
>
> In conclusion, we thank you for pushing us to better position our work with respect to the rising landscape of modern ConvNets. We sincerely appreciate all your valuable feedback to help us improve our work.  We believe that the issues raised by the reviewers have been addressed in the revision. We hope that you will reconsider our submission with this response and the revision. If you are satisfied with our response and effort, please consider updating your score. If you need any clarification, please feel free to contact us. We are more than pleased and looking forward to hearing back from you!

---

> ### Author Response · Authors · 2023-11-23
> **Additional Response to Concerns**
>
> Thanks for your timely and detailed reply! We appreciate your thorough review of all our revised manuscript, explanation, and experimental results.
>
> ### **1. About Convolutional Modulation Network.**
>
> We are delighted that you have found that our rebuttal analysis is good. Based on your suggestion, we have cited and discussed the works you mentioned in the related content to our latest revision (Appendix E).
>
> ### **2. The Significant Difference between MogaNet and FocalNet.**
>
> First, we leave aside the point that the two share completely different research motivations and novelties, albeit we believe this is truly at the heart of the ICLR community. Here, we just consider the network architecture details of both.
>
> - As for FocalNet, it mainly consists of two parts: (i) Hierarchical contextualization, implemented using a stack of **seriously connected** depth-wise convolutions, to gradually encode **multi-scale** long-range contexts. (ii) Gated aggregation, which is also designed **in series** and is implemented as the **vanilla gating operation.** Last, all the outputs are aligned to add up to the final output. Its distinctive point is that it is designed only from the **classical CV perspective**, aiming at capturing the **long-range feature relationship** under the pure ConvNet architecture. Also, it is crucial to emphasize that the FocalNet work **never mentions the word "ORDER" in the whole paper, nor the fact that ORDER has completely different meanings in different contexts and research areas.** Thus, we objectively believe the fact is that FocalNet is a fantastic work that has nothing to do with MogaNet.
> - As for our MogaNet, it primarily comprises the spatial Moga block and CA module.
>
>     **(i)** For the **spatial Moga** block, we first employ the **FD module** to **force the network to learn the inherently overlooked features (different from FocalNet).** Next, a DWConv followed a **channel-wise splitting** operation is designed **(different from FocalNet)**. For different groups of channels, we design different operations comprising different scales of DWConvs and identity connections **in parallel (unlike that of FocalNet).** Along this line, a double-branch SiLU activated gating **(unlike that of FocalNet)**.
>
>     **(ii)** For the **CA(.)**, it is designed with the same purpose as the **FD module** but is performed within the channel space. To the best of our knowledge, CA(.) is **the first** channel-wise operation that reduces the channel dimension to 1 and conducts feature complementation to force the network to learn the inherently overlooked features. **Objectively, this makes CA(.) technically novel and efficient.** Combined with CA(.) 's outstanding performance, clear visualization, and empirical analysis, we believe that it is certainly evident to distinguish CA(.) from other counterparts.
>
> ### **3. On the Major Contribution.**
>
> We first apologize that our expression has caused you confusion. We would like to address your question from two perspectives.
>
> - **The New Table Results.**
> First, as you certainly know, modern ConvNets can be interpreted as mainly comprising two parts: spatial mixing and channel mixing. Our modules, Moga(.) and CA(.), are designed as these two components, respectively.  Among them, Moga(.) is composed of FD(.), Multi-order DW(.), and the gating branch. As shown in our new table, in terms of **accuracy,** Moga(.) or the SMixer brings **a performance gain of +0.7, slightly above that of the last row CA(.)**. Besides, considering the **interaction effects**, we can see from the 3rd line Multi-order DW(.) 's interaction result, which clearly shows that the middle-order interactions (0.3n~0.5n) are effectively improved by the spatial module, along with the performance gains. Thus, we indeed have not misrepresented the main contribution of MogaNet.
> - **Re-clarification of the Main Contribution and Concepts of MogaNet.**
>
>     Holistically, our main purpose is not to propose a specific network module, but to explore visual network architecture design that can **force the network to learn to encode more expressive interactions that would have been inherently overlooked.** As we have stated, this issue was first revealed by the ICLR 2022 work, but no research tried to explore network design from this angle. Based on this concept, we design MogaNet to instantiate it on both spatial Moga(.) and channel CA(.) spaces, respectively. Therefore, it is a new kick-start for network design from an angle that has not been explored. We share a different starting point with other works. In short, the macro concept with related instantiations is our main contribution. We have conducted extensive experiments and analysis to empirically prove all the instantiations are effective toward our goals.
>
> We express our gratitude for your time and effort. We believe that the more the truth is debated, the clearer it becomes. Hope your issues have been explained and resolved.

---

> ### Author Response · Authors · 2023-11-23
> **Summarized Response to the Unaddressed Concerns**
>
> Dear Reviewer 6yMg,
>
> Thanks again for your feedback! We hope you have noticed our latest reply. We would like to emphasize that we highly treasure your valuable responses and suggestions, and found the issues you raised are crucial for us to better scrutinize and improve our work. **We have revised our work and answered all your questions.** Thus we sincerely invite you to view our response again at the end of the rebuttal period. We hope that after you check this out, you will remove some of the misconceptions about this work. Here, we would like to summarize our response above to give you an overview and make our final statement.
>
> * As you requested, we have added analysis and descriptions of convolutional modulation networks to our final revision (e.g., Appendix E).
> * Your concerns about the technical difference between our MogaNet and you have mentioned FocalNet are meticulously explained in our response. We have disassembled all the components of these two models to compare them one by one in our response. We hope your concerns have been addressed.
> * We have re-interpreted the implications of the results provided in the new table. You may have incorrectly understood what we were trying to convey before. Our spatial Moga(.) brings a performance gain of +0.7, which is slightly above that of CA(.) for the CMixer. Meanwhile, we have conveyed the main idea of this work in simple terms. View our "Additional Response to Concerns" for details.
>
> At the current version, we have been particularly sincere and diligent in responding to your valuable comments. **We trust that our response has clearly answered your questions, and we invite you to review it.** We respectfully believe that this work is attuned to the ICLR community, and we hope that our work can be seen by more researchers in the community. Thus, your rating is extremely valuable to us. If there might be any additional opportunities for us to further improve our manuscript and potentially increase your rating, we would like to discuss this with you and would be happy to provide more information based on your feedback or further questions. Thanks again for your efforts!
>
> Best regards,
>
> Authors

---

### Author Response · Authors · 2023-11-20
**Official General Response by Authors**

Dear Reviewers,

Greetings!

We would like to first express our sincere appreciation for the time and effort all 4 reviewers dedicated to reviewing our submission. We have individually responded to each reviewer's comments, and we found them all to be particularly helpful in making improvements to our revision. The key points of revision are highlighted in $\color{magenta}{magenta}$ color.  We have noticed that there seem to be specific questions in common. To give you a better picture of the novelty and contribution we have made, we thereby provide further clarification as follows:

**1. Presentation Clarity**

We acknowledge that there may have been sections in our paper that were not as clear as they could have been. Your feedback heightened our awareness of the importance of clear communication in research. During the revision process, we realized that the lack of clarity seriously undermined the expression of what we were contributing. Therefore, we have revised the paper to improve the clarity based on your valuable suggestions. Please refer to the revised content **in magenta color** for more details.

**2. Main Contributions and Novelty**

Several reviewers expressed concerns about the academic novelty of our MogaNet. We have a deep appreciation for the reviewers' comments, and we are now aware of exactly where we went wrong. We have emphasized our key motivation, the main challenges we have tackled, and the reason why this network architecture design is of significance in our revision. Please refer to the revised **abstract, Sec.1, Sec.3, Sec.4,** and all the magenta **highlighted content** for details. Here, we would like to present an overview of the motivation, our contributions, and novelties, taking into account the reviewers’ comments and the improvements made in our revision:

**(2.1) The Distinctive Purpose of MogaNet.** Our MogaNet work was inspired by an ICLR22 work, which revealed for the first time the problem that existing deep neural networks suffer from a representation bottleneck under a hierarchical theoretical analysis (Multi-order Game-theoretic Interaction) and thereby proposed a loss function to solve this problem. This work was recognized by ICLR2022, but no further attempts have been made in the community to solve this problem. Thus, we are inspired to address this problem from a new perspective of network architecture design (instead of the loss function).

Specifically, deep neural networks show a natural proclivity towards multi-order interactions - the most expressive middle-order interactions are not well encoded in the current learning process, and we attempt to address this issue with network architectures that can **force the network to learn to encode more expressive interactions that would have otherwise been inherently overlooked.** To facilitate understanding, from a computer vision standpoint, you can roughly, but not quite accurately, interpret this as forcing the network to capture features that were discriminative but originally ignored.

To the best of our knowledge, **we are the first to address this issue with specific network architecture design and are also the first to approach network architecture design from the multi-order interaction perspective.** Therefore, it is clear that we have a completely different starting point compared to other works. Again, briefly speaking, we are making a new attempt to solve the problem that was first raised by the ICLR2022 work rather than a merely "borrowed" network design without a top-down guideline. As such, we believe that MogaNet is obviously interesting and novel from an academic research point of view.

**(2.2) The Network Architecture Design of MogaNet.**

We believe that MogaNet's network architecture itself is novel rather than "limited novelty", as some reviewers have claimed. Specifically, the proposed modules in MogaNet include FD(.), Multi-DW(.) (with gating), and CA(.). The first two together form the Spatial Moga(.) block as SMixer. Reviewers mainly mention the following relevant works: VAN, FocalNet, LocalViT, Conv2Former, and PVT-V2. Firstly, please forgive us for reiterating that MogaNet is from a completely unique starting point compared to these works, and we are not solving the same problem. Secondly, at the level of network architecture design, we are also quite apart. We included tables in the individual responses to illustrate the novelty of MogaNet.

In a nutshell, we sincerely appreciate all your effort and valuable feedback. Since the discussion period can only last until November 22 and we are approaching this deadline, we would like to discuss this with you during this time and would be happy to provide more information based on your feedback or further questions. If you are satisfied with our response and effort, please consider updating your score. If you need any clarification, please feel free to contact us. We are looking forward to hearing back from you!

Sincerely,

Authors

---

### Author Response · Authors · 2023-11-22
**Official Comment by Authors**

Dear Reviewers,

We greatly appreciate your effort and valuable feedback. In our response, we carefully illustrated and answered all your questions in detail. Additional experiments and analysis results are also provided as you had requested. In addition, we further clarified several unclear statements in the paper. We have incorporated all changes into the revised manuscript for your consideration. We hope your concerns have been addressed.

As you may know, unlike previous years, the discussion period this year can only last until November 22, and we are gradually approaching this deadline. We take it serious and would like to discuss this with you during this time. We would be happy to provide more information based on your feedback or further questions.

If you are satisfied with our response, please consider updating your score. If you need any clarification, please feel free to contact us.

Best regards,

Authors

---

### Meta-Review · Area_Chair_FrBd · 2023-12-08

**Metareview:**

The paper proposes MogaNet, a new convolutional neural network architecture that utilizes multi-order gated aggregation to increase the representation ability of ConvNets. The network is composed of two modules for spatial and channel mixing. This network achieves state-of-the-art results on various computer vision tasks such as image classification, object detection, semantic segmentation, and pose estimation. The paper is well executed and shows extensive quantitative results. The rebuttal successfully answered some of the reviewer’s questions, making this paper a weak accept (6).

**Justification For Why Not Higher Score:**

While the rebuttal answered some concerns, the reviewers pointed out limited discrimination from FocalNet as well as the importance of the presented results. To what extend would MogaNet become a reference architecture in the future is unclear.

**Justification For Why Not Lower Score:**

Solid and thorough experimental evaluation of a reasonable architectural improvement.

---

### Decision · Program_Chairs · 2024-01-16

Accept (poster)